# BROOD: Bilevel and robust optimization and outlier detection for efficient tuning of high-energy physics event generators

Wenjing Wang[1], Mohan Krishnamoorthy[2], Juliane Müller[1]★, Stephen Mrenna[3]†,
Holger Schulz[4], Xiangyang Ju[1], Sven Leyffer[2] and Zachary Marshall[1]

**1** Lawrence Berkeley National Laboratory, Berkeley, CA 94720
**2** Argonne National Laboratory, Lemont, IL 60439
**3** Fermi National Accelerator Laboratory, Batavia, IL 60510
**4** Department of Computer Science, Durham University, South Road, Durham DH1 3LE, UK

★ julianemueller@lbl.gov, † mrenna@fnal.gov

## Abstract

The parameters in Monte Carlo (MC) event generators are tuned on experimental measurements by evaluating the goodness of fit between the data and the MC predictions. The relative importance of each measurement is adjusted manually in an often time-consuming, iterative process to meet different experimental needs. In this work, we introduce several optimization formulations and algorithms with new decision criteria for streamlining and automating this process. These algorithms are designed for two formulations: bilevel optimization and robust optimization. Both formulations are applied to the datasets used in the ATLAS A14 tune and to the dedicated hadronization datasets generated by the SHERPA generator, respectively. The corresponding tuned generator parameters are compared using three metrics. We compare the quality of our automatic tunes to the published ATLAS A14 tune. Moreover, we analyze the impact of a pre-processing step that excludes data that cannot be described by the physics models used in the MC event generators.

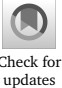

# 1 Introduction and Motivation

Monte Carlo (MC) event generators are simulation tools that predict the properties of high-energy particle collisions. Event generators are built from theoretical formulae and models that describe the probabilities for various sub-event phenomena that occur in a high-energy collision. They are developed by physicists as a bridge between particle physics perturbation theory, which is defined at very high energy scales, and the observed sub-atomic particles, which are low-energy states of the strongly-interacting full theory. This bridge is essential for interpreting event collision data in terms of the fundamental quantities of the underlying theory. See [1] for an overview of the event generators used for physics analysis at the Large Hadron Collider (LHC).

The description of particle collisions requires an understanding of phenomena at many different energy scales. At high energy scales (much larger than the masses of the sub-atomic particles), first principle predictions can be made in a perturbative framework based on a few universal parameters. At intermediate energy scales, an approximate perturbation theory can be established that introduces less universal parameters. At low energy, motivated, but subjective, models are introduced to describe sub-atomic particle production. These low-energy models introduce a large number of narrowly defined parameters. To make predictions or inferences, one must have a handle on the preferred models and the values of the parameters needed to describe the data. This process of adjusting the parameters of the MC simulations to match data is called *tuning*.

This tuning task is complicated by the fact that the phenomenological models cannot claim to be complete or scale-invariant. When compared to a large set of collider data collected in different energy regimes, the MC-models do not describe the full range of event properties equally well. Typically, the physicists demand a tune that describes a subset of the data very well, another subset moderately well, and a remainder that must only be described qualitatively. This distribution of subsets may well vary from one group of physicists to another and has led to the education of experts who subjectively select and weigh data to achieve some

physics goal. Two such exercises are the Monash tune [2] and the A14 tune [3], though others exist in the literature. Both of these tunes are successful, in the sense that they have been useful in understanding a wide range of phenomena observed at particle colliders. However, the current approach to tuning remains inefficient and biased and, given the nature of the problem with physicists having different tuning objectives, mathematical rigor is lacking.

This work introduces a framework that, once agreed upon, greatly reduces the subjective element of the tuning process and replaces it with an automated way to select the data for parameter tuning.

## 1.1 Notation and terminology

The data used in the tuning process are in the form of observables, denoted by $\mathcal{O}$, and the set of observables is denoted by $\mathcal{S}_{\mathcal{O}}$. Observables are quantities constructed from the (directly or indirectly) measured sub-atomic particles produced in an event. In this case, each observable is presented as a *histogram* that shows the frequency that the observable is measured over a range of possible values (see Figure 8 for example histograms). The range can be one or many divisions of the interval from the minimum to the maximum value that the observable can obtain. These divisions are called *bins*. In practice, the size of a bin is set by how well an observable can be measured. The number of bins of an observable $\mathcal{O}$ is denoted as $|\mathcal{O}|$. We use $\mathcal{R}$ to denote the reference data in the histograms, a subscript $b$ to denote a bin, $\mathcal{R}_b$ to denote the data value in a bin, and $\Delta\mathcal{R}_b$ to denote the corresponding 1-$\sigma$ measurement uncertainty which is interpreted as the standard deviation of a Gaussian random variable.

The MC-model has parameters $\mathbf{p}$, a $d$-dimensional vector in the space $\Omega$, $\mathbf{p} \in \Omega \subset \mathbb{R}^d$. The MC-based simulations are denoted by MC($\mathbf{p}$) to emphasize that they depend on the physics parameters $\mathbf{p}$. The histograms computed from the MC simulation have the same structure as the histograms obtained from the measurement data $\mathcal{R}$, with a prediction per bin $\mathrm{MC}_b(\mathbf{p})$ and an uncertainty associated with each bin $\Delta\mathrm{MC}_b(\mathbf{p})$. The uncertainty on the MC simulation is the numerical precision of the prediction, which typically scales as the inverse of the square root of the number of simulated events in a particular bin. Theoretical and model uncertainties are not currently included, but are discussed later.

## 1.2 Mathematical formulation of the tuning problem

Our goal is to find a set of physics parameters, $\mathbf{p}^*$, that minimizes the difference between the experimental data and the simulated data from an MC event generator. This difference is defined as follows:

$$\chi^2_{\mathrm{MC}}(\mathbf{p}, \mathbf{w}) = \sum_{\mathcal{O} \in \mathcal{S}_{\mathcal{O}}} w_{\mathcal{O}} \sum_{b \in \mathcal{O}} \frac{(\mathrm{MC}(\mathbf{p}) - \mathcal{R}_b)^2}{\Delta\mathrm{MC}_b(\mathbf{p})^2 + \Delta\mathcal{R}_b^2}, \tag{1}$$

where $w_{\mathcal{O}}$ is the weight for an observable $\mathcal{O}$ and $\mathbf{w}$ is a vector of weights, $\mathbf{w} = [w_1, \ldots, w_{|\mathcal{S}_{\mathcal{O}}|}]^T$. In general, the number of bins can be different for different observables. The weights $w_{\mathcal{O}} \geq 0$ reflect how much an observable contributes to the tune, i.e., if $w_{\mathcal{O}} = 0$ for some $\mathcal{O}$, then this observable will not influence the tuning of $\mathbf{p}$. Since (1) is likely multimodal, several local optima exist (see [4, page 13] for the definition of local optimality) and our goal with using numerical optimization is to find at least a locally optimal solution, which is not guaranteed to be found by hand-tuning methods. Note that (1) treats the observables independently without correlations. Currently, the majority of collider data available for tuning are provided without these correlations. When such information becomes readily available, (1) will need to be modified in a non-trivial way to include them.

The MC simulation is computationally expensive (the generation of 1 million events for a given set of parameters consumes about 800 CPU minutes on a typical computing cluster),

severely limiting the number of parameter choices **p** that can be used in the tuning. To overcome these issues, we construct a parameterization of the MC simulation following the work in [5] and advancing the method to new approximation models. Our new implementation, named APPRENTICE, is available at https://github.com/HEPonHPC/apprentice. The function in Eq. (1) is not minimized directly. Instead, during the optimization over **p**, the MC simulation is replaced by a surrogate model (here, a polynomial (see [5]) or a rational approximation to a number of MC simulations). For each bin $b$ of each histogram, the central value and the corresponding uncertainty of the model prediction are parameterized independently as functions of the model parameters **p** yielding analytic expressions $f_b(\mathbf{p})$ and $\Delta f_b(\mathbf{p})$, respectively, that can be evaluated in milliseconds. Thus, instead of Eq. (1), we minimize

$$\chi^2(\mathbf{p}, \mathbf{w}) = \sum_{\mathcal{O} \in \mathcal{S}_{\mathcal{O}}} w_{\mathcal{O}} \sum_{b \in \mathcal{O}} \frac{(f_b(\mathbf{p}) - \mathcal{R}_b)^2}{\Delta f_b(\mathbf{p})^2 + \Delta \mathcal{R}_b^2}, \tag{2}$$

which can be done efficiently using numerical methods. Eq. (2) implicitly assumes that each bin $b$ is completely independent of all other bins. Note that the choice of surrogate model introduces an uncertainty whose quantification is outside of the scope of this paper.

In practice, the weights $w_{\mathcal{O}}$ in Eq. (2) are adjusted manually, based on experience and physics intuition. The selection of weights is time-consuming and different experts may have different opinions about how well each observable is approximated by the model. Our goal is to automate the weight adjustment, yielding a less subjective and less time-consuming process to find the optimal physics parameters **p** that will then be used in the actual MC simulation. This problem was also considered in [6], where weights are assigned based on correlations between parameters and observables without any reference to measured data values. Also related to this work is that of [7], which treats tuning as a black-box optimization problem within the framework of Bayesian optimization, but with no weighting of data.

For convenience, we summarize our notation in Table 1.

Table 1: Notation.

| Notation | Definition |
| --- | --- |
| $\mathcal{O}$ | observables that are constructed from data and MC-based simulations in the form of histograms |
| $|\mathcal{O}|$ | the number of bins in an observable $\mathcal{O}$ |
| $\mathcal{S}_{\mathcal{O}}$ | the set of observables used in the tune |
| $|\mathcal{S}_{\mathcal{O}}|$ | the number of observables |
| $\mathcal{R}$ | the data in the histograms |
| $b$ | a bin of a histogram $\mathcal{O}$ |
| $\mathcal{R}_b$ | the data value in a bin |
| $\Delta \mathcal{R}_b$ | data uncertainty corresponding to the data value in a bin |
| **p** | a $d$-dimensional vector of real-valued parameters |
| MC(**p**) | an MC simulation that depends on the physics parameters **p** |
| MC$_b$(**p**) | the MC simulation in a bin $b$ |
| $\Delta$MC$_b$(**p**) | an uncertainty associated with the MC simulation in a bin $b$ |
| $f_b(\mathbf{p})$ | central value of the model prediction parameterized independently as a function of the model parameters **p** |
| $\Delta f_b(\mathbf{p})$ | the uncertainty of the model prediction parameterized independently as a function of the model parameters **p** |
| $r_b(\mathbf{p})$ | the variance associated with bin b as a function of model parameter **p** |
| **w** | an $|\mathcal{S}_{\mathcal{O}}|$-dimensional vector of real-valued weights |

| Notation | Definition |
|---|---|
| $w_{\mathcal{O}}$ | the weight given to a histogram in constructing a tune (if $w_{\mathcal{O}} = 0$ for some $\mathcal{O}$, then this observable will not influence the tuning of $\mathbf{p}$). |
| $\widehat{\mathbf{p}}_{\mathbf{w}}$ | optimal physics parameters for a given choice for the weights |
| $\mathbf{w}^*$ | an optimal set of weights for the observables |
| $\widehat{\mathbf{p}}_{\mathbf{w}^*}$ | the optimal set of simulation parameters corresponding to an optimal set of weights $\mathbf{w}^*$ for the observables |
| $g$ | the outer objective function of $\mathbb{R}^{|\mathcal{S}_{\mathcal{O}}| \times d} \mapsto \mathbb{R}$ used in the bilevel optimization |
| $\mu$ | a hyperparameter that specifies the percentage of the observables used in the robust optimization |
| $\chi^2_{\mathcal{O}}(\mathbf{p})$ | the per-observable error averaged over all bins in the observable $\mathcal{O}$ |
| $\mathbf{p}^{\mathcal{O}}_{\text{ideal}}$ | the *ideal* tune for an observable $\mathcal{O}$, i.e., the parameters that minimize Eq. (12) when using only observable $\mathcal{O}$ for the tune |

## 2 Finding the Optimal Weights for Each Observable

In this section, we describe two mathematical formulations for finding the optimal weights in Eq. (2), namely bilevel and robust optimization.

### 2.1 Bilevel optimization formulation

We formulate a bilevel optimization problem as follows:

$$\min_{\mathbf{w} \in [0,1]^{|\mathcal{S}_{\mathcal{O}}|}, \hat{\mathbf{p}}_{\mathbf{w}} \in \Omega} g(\mathbf{w}, \widehat{\mathbf{p}}_{\mathbf{w}}), \tag{3a}$$

$$\text{subject to} \sum_{\mathcal{O} \in \mathcal{S}_{\mathcal{O}}} w_{\mathcal{O}} = 1, \tag{3b}$$

$$\widehat{\mathbf{p}}_{\mathbf{w}} \in \arg\min_{\mathbf{p} \in \Omega} \chi^2(\mathbf{p}, \mathbf{w}), \tag{3c}$$

where the upper-level function $g : \mathbb{R}^{|\mathcal{S}_{\mathcal{O}}| \times d} \mapsto \mathbb{R}$ describes a merit function to determine the goodness of weights (see below for the definitions we use in this work). The lower-level Eq. (3c) (same as Eq. (2)) corresponds to finding optimal parameters $\widehat{\mathbf{p}}_{\mathbf{w}}$ for a given set of weights $\mathbf{w}$ (note that, in general, multiple local minimizers may exist). The weights are normalized to sum to unity, see Eq. (3b), in order to prevent the trivial solution where all weights are 0. Bilevel optimization problems have been studied extensively in the literature, see, e.g., [8–12].

In the following, we discuss two definitions of the outer objective function $g(\mathbf{w}, \widehat{\mathbf{p}}_{\mathbf{w}})$. Other formulations are possible and our selection is driven by the goal to achieve reasonably good agreement between the simulated and the observed data for all observables (rather than fitting a few observables extremely well and others poorly).

#### 2.1.1 Formulation 1: Portfolio to balance mean and variance of errors

The portfolio objective function is motivated by portfolio optimization in finance [13], where the goal is to maximize the expected return while minimizing the risk. Translated to our problem, we want to minimize the expected error over all observables while also minimizing the variance over these errors.

For a given set of weights $\mathbf{w}$, we obtain the "$\mathbf{w}$-optimal" parameters $\widehat{\mathbf{p}}_{\mathbf{w}} = \hat{\mathbf{p}}(\mathbf{w})$. For each observable $\mathcal{O}$, an error term is averaged over the number of bins in the observable ($|\mathcal{O}|$):

$$e_{\mathcal{O}}(\widehat{\mathbf{p}}_{\mathbf{w}}) = \frac{1}{|\mathcal{O}|} \sum_{b \in \mathcal{O}} \frac{(f_b(\widehat{\mathbf{p}}_{\mathbf{w}}) - \mathcal{R}_b)^2}{\Delta f_b(\widehat{\mathbf{p}}_{\mathbf{w}})^2 + \Delta \mathcal{R}_b^2}, \quad \mathcal{O} \in \mathcal{S}_{\mathcal{O}}, \tag{4}$$

where the error $e_{\mathcal{O}}(\widehat{\mathbf{p}}_{\mathbf{w}})$ for each observable depends on the choice of the weights $\mathbf{w}$. Thus, we obtain a set of $|\mathcal{S}_{\mathcal{O}}|$ average error values from which we compute the following statistics:

$$\mu(\hat{\mathbf{p}}(\mathbf{w})) = \mu(\widehat{\mathbf{p}}_{\mathbf{w}}) = \frac{1}{|\mathcal{S}_{\mathcal{O}}|} \sum_{\mathcal{O} \in \mathcal{S}_{\mathcal{O}}} e_{\mathcal{O}}(\widehat{\mathbf{p}}_{\mathbf{w}}): \text{ average error over all observables}, \tag{5a}$$

$$\sigma^2(\hat{\mathbf{p}}(\mathbf{w})) = \sigma^2(\widehat{\mathbf{p}}_{\mathbf{w}}) = \frac{1}{|\mathcal{S}_{\mathcal{O}}|} \sum_{\mathcal{O} \in \mathcal{S}_{\mathcal{O}}} [e_{\mathcal{O}}(\widehat{\mathbf{p}}_{\mathbf{w}}) - \mu(\widehat{\mathbf{p}}_{\mathbf{w}})]^2 \tag{5b}$$

$$: \text{ empirical variance of errors over all observables}.$$

The portfolio objective function for the outer optimization then becomes

$$g(\mathbf{w}, \widehat{\mathbf{p}}_{\mathbf{w}}) = \mu(\widehat{\mathbf{p}}_{\mathbf{w}}) + \sigma^2(\widehat{\mathbf{p}}_{\mathbf{w}}), \tag{6}$$

which aims at simultaneously minimizing the expected error *and* the variance of the errors of all observables. Thus, instead of minimizing only an expected value and potentially obtaining a solution that allows for some observables having large errors and others small errors, we aim to find a solution that provides a good tradeoff between both metrics. For problems in which minimizing the variance is of higher priority, one can introduce a multiplier $\lambda$ before the variance term that reflects "risk aversion". In that case, if $\lambda$ is large, we are more risk-averse, since reducing the variance associated with the errors will drive the minimization. If $\lambda$ is small, we are less risk-averse, and minimizing the mean of the errors is emphasized.

### 2.1.2 Formulation 2: Scoring of model fit and data uncertainty

We consider a second outer objective function formulation based on scoring schemes ([14, Eq. (27)]). The performance of a generic predictive model $P$ at a point $x$ is defined by a scoring rule, $S(P, x) = -\left(\frac{x - \mu_P}{\sigma_P}\right)^2 - \log \sigma_P^2$, where $P$ has mean performance $\mu_P$ and variance $\sigma_P^2$. A larger value for $S(P, x)$ signifies better model performance. Thus, we minimize the negative of $S(P, x)$:

$$s(P, x) = -S(P, x) = \left(\frac{x - \mu_P}{\sigma_P}\right)^2 + \log \sigma_P^2. \tag{7}$$

The intuition behind this scoring scheme is that it takes both the model fit and data uncertainty into consideration. For our application, $x$ corresponds to the simulation prediction $f_b(\mathbf{p})$, $\mu_P$ to our observation data $\mathcal{R}_b$, and the variance $\sigma_P^2$ to our data uncertainty $\Delta \mathcal{R}_b$. For each bin $b$ in an observable, we calculate the score based on Eq. (7). Then, we compute the median (and mean) of the scores over all bins to obtain the median (average) performance for each observable. In order to form the upper-level objective function, we sum up the median (mean) scores over all observables:

- Outer objective based on median score

$$g(\mathbf{w}, \widehat{\mathbf{p}}_{\mathbf{w}}) = \sum_{\mathcal{O} \in \mathcal{S}_{\mathcal{O}}} \tilde{s}_{\mathcal{O}}(\widehat{\mathbf{p}}_{\mathbf{w}}), \tag{8a}$$

$$\tilde{s}_{\mathcal{O}}(\widehat{\mathbf{p}}_{\mathbf{w}}) = \text{median of } \left\{ \left(\frac{f_b(\widehat{\mathbf{p}}_{\mathbf{w}}) - \mathcal{R}_b}{\Delta \mathcal{R}_b}\right)^2 + \log(\Delta \mathcal{R}_b^2), \forall b \in \mathcal{O} \right\}. \tag{8b}$$

• Outer objective based on mean score

$$g(\mathbf{w}, \widehat{\mathbf{p}}_\mathbf{w}) = \sum_{\mathcal{O} \in \mathcal{S}_\mathcal{O}} \bar{s}_\mathcal{O}(\widehat{\mathbf{p}}_\mathbf{w}), \tag{9a}$$

$$\bar{s}_\mathcal{O}(\widehat{\mathbf{p}}_\mathbf{w}) = \frac{1}{|\mathcal{O}|} \sum_{b \in \mathcal{O}} \left\{ \left( \frac{f_b(\widehat{\mathbf{p}}_\mathbf{w}) - \mathcal{R}_b}{\Delta \mathcal{R}_b} \right)^2 + \log(\Delta \mathcal{R}_b^2) \right\}. \tag{9b}$$

Both the median and the mean score outer objective functions take into account the deviation of the prediction of $f_b(\widehat{\mathbf{p}}_\mathbf{w})$ from $\mathcal{R}_b$ and the uncertainty in the data $\Delta \mathcal{R}_b$. Thus, if an observable has large uncertainties in the data or the model $f_b(\widehat{\mathbf{p}}_\mathbf{w})$ does not approximate the data $\mathcal{R}_b$ well, the score for this observable deteriorates. Ideally, both terms $\left( \frac{f_b(\widehat{\mathbf{p}}_\mathbf{w}) - \mathcal{R}_b}{\Delta \mathcal{R}_b} \right)^2$ and $\log(\Delta \mathcal{R}_b^2)$ will be small.

### 2.1.3 Solving the bilevel optimization problem using surrogate models

Solving the inner optimization problem (3c) for each weight vector $\mathbf{w}$ is generally computationally non-trivial and its computational demand increases with the number of physics parameters $\mathbf{p}$ that have to be optimized and the number of observables present. Here, we use APPRENTICE to obtain a set of optimal physics parameters $\widehat{\mathbf{p}}_\mathbf{w}$. The goal is to try as few weights $\mathbf{w}$ as possible. We interpret the solution of the inner optimization problem as a black-box function evaluation of $g(\mathbf{w}, \hat{\mathbf{p}}_\mathbf{w})$ for $\mathbf{w}$. Given an initial set of input-output data pairs $\left\{ (\mathbf{w}_i, g(\mathbf{w}_i, \hat{\mathbf{p}}_{\mathbf{w}_i})) \right\}_{i=1}^I$, we fit a surrogate model[1] (here a radial basis function [15]) that allows us to predict the values of $g(\mathbf{w}, \hat{\mathbf{p}}_\mathbf{w})$ at untried $\mathbf{w}$. In each iteration of the optimization algorithm, these predictions are used to select the most promising weight vector for which the inner optimization problem should be solved next. Promising weight vectors have either low predicted values of $g(\cdot)$ or are far away from already evaluated points [16,17]. Each time a new weight vector has been evaluated, the surrogate model is updated. This iterative process repeats until a stopping criterion has been met, e.g., a maximal number of weight vectors has been evaluated or a maximal CPU time has been reached. Details about the surrogate model algorithm are given in the online supplement Section A.1.

Note that the surrogate model based optimizer balances local and global searches in order to enable an escape from local optima. However, our algorithm cannot guarantee to converge to the globally optimal solution because the optimization problem is highly multi-modal and blackbox.

## 2.2 Robust optimization formulation

As an alternative to the bilevel formulation, we developed a single-level robust optimization formulation for finding the optimal weights for Eq. (2). Robust optimization estimates the parameters $\mathbf{p}$ that minimize the largest deviation $(f_b(\mathbf{p}) - \mathcal{R}_b)^2$ over all bins in an uncertainty set $\mathcal{U}_b$ of bin $b$:

$$\underset{\mathbf{w} \in [0,1], \mathbf{p} \in \Omega}{\text{minimize}} \sum_{\mathcal{O} \in \mathcal{S}_\mathcal{O}} \frac{w_\mathcal{O}}{|\mathcal{O}|} \sum_{b \in \mathcal{O}} \underset{\mathcal{R}_b \in \mathcal{U}_b}{\text{maximize}} (f_b(\mathbf{p}) - \mathcal{R}_b)^2 . \tag{10}$$

The uncertainty set $\mathcal{U}_b$ contains uncertain data and the goal of the optimization is to choose the best (most robust) solution among candidates that remain feasible for all realizations of the data in $\mathcal{U}_b$. Furthermore, $\mathcal{U}_b$ is not a probability distribution since it is a bound set and we only consider feasible data within the set. Assuming that the experiment and the MC simulation

---

[1]This surrogate model for the weights is independent of the one used to evaluate the MC-based predictions.

are described using independent random variables with mean $\mathcal{R}_b$, the uncertainty set $\mathcal{U}_b$ for each bin $b$ is described by the interval $[\mathcal{R}_b - \Delta\mathcal{R}_b - \Delta f_b(\mathbf{p}), \mathcal{R}_b + \Delta\mathcal{R}_b + \Delta f_b(\mathbf{p})]$.

Introducing slack variables $\boldsymbol{t} = \begin{bmatrix} t_1, t_2, \ldots, t_{|\mathcal{O}|} \end{bmatrix}$, we rewrite (10) as:

$$\underset{\boldsymbol{t},\mathbf{w}\in[0,1],\mathbf{p}\in\Omega}{\text{minimize}} \sum_{\mathcal{O}\in\mathcal{S}_{\mathcal{O}}} \frac{w_{\mathcal{O}}}{|\mathcal{O}|} \sum_{b\in\mathcal{O}} t_b, \tag{11a}$$

subject to

$$t_b \geq (f_b(\mathbf{p}) - (\mathcal{R}_b - \Delta\mathcal{R}_b - \Delta f_b(\mathbf{p})))^2 \quad \forall b \in \mathcal{O}, \forall \mathcal{O} \in \mathcal{S}_{\mathcal{O}} \tag{11b}$$

$$t_b \geq (f_b(\mathbf{p}) - (\mathcal{R}_b + \Delta\mathcal{R}_b + \Delta f_b(\mathbf{p})))^2 \quad \forall b \in \mathcal{O}, \forall \mathcal{O} \in \mathcal{S}_{\mathcal{O}}$$

$$\sum_{\mathcal{O}\in\mathcal{S}_{\mathcal{O}}} \frac{w_{\mathcal{O}}}{|\mathcal{O}|} \geq \frac{\mu}{100} \sum_{\mathcal{O}\in\mathcal{S}_{\mathcal{O}}} \frac{1}{|\mathcal{O}|}, \tag{11c}$$

where the constraint (11c) is enforced to avoid the trivial solution of all weights being zero. In this constraint, we bound the sum of the weights away from zero by a hyperparameter $\mu$ that specifies the percentage of the observables that should be used in the optimization. Problem (11) is attractive because it formulates the problem of finding optimal weights as a single-level optimization problem, which is easier to solve than the bilevel problem Eq. (3). However, like the bilevel algorithms, this approach cannot guarantee to converge to the globally optimal solution due to the nonlinear constraints (11b).

Selecting the best $\mu$ among all the 100 runs of robust optimization is determined using a cumulative density curve of the number of observables satisfying $\dfrac{\chi^2_{\mathcal{O}}(\mathbf{p}^*, \mathbf{w})}{|\mathcal{O}|} \leq \tau$, where $\mathbf{p}^*$ is the optimal parameter obtained from the robust optimization run, $\mathbf{w} = \mathbf{1}$, $\tau \in \mathbb{R}^+$ and $\mathcal{O} \in \mathcal{S}_{\mathcal{O}}$. Hence, in the plot of this curve (e.g., see Figure 11), the number of observables on the y-axis is monotonically increasing as $\tau$ increases on the x-axis. Then, the area between the cumulative density curve for each robust optimization run and the ideal cumulative density curve is computed. To build the ideal cumulative density curve, the $\mathbf{p}^*$ in $\dfrac{\chi^2_{\mathcal{O}}(\mathbf{p}^*, \mathbf{w})}{|\mathcal{O}|} \leq \tau$ is obtained by considering only observable $\mathcal{O}$ in Eq. (2). The best run is then chosen to be the one whose area to the ideal cumulative density curve is the smallest. An example plot of the cumulative density curve and an illustration of the procedure to find the best run is included in Section A.4 of the online supplement.

## 3   Data Pre-processing: Filtering Observables or Bins

We also investigate the question of how to detect and exclude observables or bins whose data $\mathcal{R}_b$ cannot be explained by the MC simulation model. This is driven by a significant discrepancy between the simulation and data. Such discrepancies can arise for at least two reasons: (1) a mistake has been made in the experimental analysis; and/or (2) the observable is out of the domain of predictions that can be made reliably with the simulation. For our studies, we assume that the source of discrepancies is from (2). Because the simulation is a metamodel constructed from many smaller models, it is difficult to make *a priori* statements about the domain of its predictions. If the intrinsic theoretical uncertainty on our models were known quantitatively, then it could be incorporated into the fitting procedure. However, such uncertainties are not known currently except by the brute-force method of choosing extreme values of the input parameters. Important physics may be missing from the metamodel and/or a model can describe the mean behavior but not the rarer fluctuations around the mean. The simulation should be able to describe the physics, but the inclusion of some observables worsen the description. Thus, it is quite reasonable to exclude these observables.

In our discussion to this point, we have assumed that each *observable* has a given weight. However, in those situations where the model can describe the mean behavior, it can be beneficial to filter out individual bins $b$ of the observable. In the observables considered in this study, and typical of the high energy physics phenomenon, the models can have difficulties in describing the complete distribution.

## 3.1 Filtering of observables by outlier detection

Using the surrogate model $f_b(\mathbf{p})$ to approximate the expensive MC simulation, we can efficiently minimize the per-observable-$\chi^2$ function:

$$\chi^2_{\mathcal{O}}(\mathbf{p}) = \frac{1}{|\mathcal{O}|} \sum_{b \in \mathcal{O}} \frac{(f_b(\mathbf{p}) - \mathcal{R}_b)^2}{\Delta f_b(\mathbf{p})^2 + \Delta \mathcal{R}_b^2}, \mathcal{O} \in \mathcal{S}_{\mathcal{O}}, \tag{12}$$

for each observable $\mathcal{O} \in \mathcal{S}_{\mathcal{O}}$ separately. $\chi^2_{\mathcal{O}}(\mathbf{p})$ represents the average per-bin error for the observable and the best possible fit of the model for this single observable. If we used only one observable for the tune, the parameters $\mathbf{p}^{\mathcal{O}}_{\text{ideal}}$ that minimize Eq. (12) would represent the *ideal* tune. The corresponding ideal objective function value $\chi^2_{\mathcal{O}}(\mathbf{p}^{\mathcal{O}}_{\text{ideal}})$ is the best possible result for each individual observable $\mathcal{O}$. Because the ideal parameter values will be different for each observable, we will not be able to obtain one parameter set that minimizes Eq. (12) for all observables simultaneously. Therefore, we obtain a set $\mathcal{X}$ of length $|\mathcal{S}_{\mathcal{O}}|$ of ideal objective function values of Eq. (12): $\mathcal{X} := \{\chi^2_1(\mathbf{p}^1_{\text{ideal}}), \chi^2_2(\mathbf{p}^2_{\text{ideal}}), \dots, \chi^2_{|\mathcal{S}_{\mathcal{O}}|}(\mathbf{p}^{|\mathcal{S}_{\mathcal{O}}|}_{\text{ideal}})\} = \{\chi_i\}_{i=1}^{|\mathcal{S}_{\mathcal{O}}|}$. If the ideal error is large for some observables, it means that the model is not able to fit the data of these observables well at all. Therefore, the inclusion of these data in optimizing Eq. (2) may negatively impact the overall optimization because large errors might bias the optimization.[2]

To address this issue, we use the distribution of the values in $\mathcal{X}$ to identify outliers (observables with values for Eq. (12) "that deviate so much from other observations as to arouse suspicions that it was generated by a different mechanism." [18]). We exclude the outlier observables from the optimization of Eq. (2) by setting their corresponding weights to zero, $w_{\mathcal{O}} = 0$.

Multiple methods can be used for outlier detection, such as scatter plots [19], Z-score [19, Section 1.3.5.17], interquartile range [20], generalized extreme studentized deviate [21], Grubb's test [22,23], Dixon's Q test [24], Thompson tau test [25], Pierce's Criterion [26], and Tietjen-Moore test [27], to name a few. We obtained reasonable results using the Z-score. For the set $\mathcal{X} = \{\chi_i\}_{i=1}^{|\mathcal{S}_{\mathcal{O}}|}$, the Z-score of an observation $\chi_i$ is defined as $z_i = (\chi_i - m)/s$ where $m$ is the mean of the observation set $\mathcal{X}$ and $s$ is the standard deviation. We calculate the Z-score for each data point $i$ in $\mathcal{X}$ and define an outlier as $z_i \geq 3$. In other words, any ideal fit with a residual outside of 3 standard deviations is classified as an outlier. The value 3 was chosen based on the rule of thumb for outlier detection in which almost all of the data (99.7%) should be within three standard deviations from the mean. The benefit of performing the outlier detection is that the computational cost of minimizing Eq. (2) is reduced.

## 3.2 Filtering of bins by hypothesis testing

We explore a second and more refined approach that allows us to identify and exclude bin data that cannot be approximated well by the MC simulator model from the optimization of Eq. (2) instead of eliminating whole observables. The observables themselves are typically chosen to test theoretical or phenomenological models, and the binning is chosen so that it represents the detector resolution [28]. The motivation of excluding bins is that often the physics models

---

[2]We address later the fidelity of the surrogate model.

fail near the boundaries of observables, such as the turn on or tail of a particle production spectrum.

To this end, we use the $\chi^2$ test, which is a hypothesis test performed when the test statistic is $\chi^2$-distributed under the null hypothesis [29]. Note that the $\chi^2$ test statistic is different from the $\chi^2(\mathbf{p}, \mathbf{w})$ objective function introduced earlier. We first compute the $\chi^2$ test statistic for a subset $\mathcal{B}$ of the bins in an observable $\mathcal{O}$ using the computationally cheap approximation model $f_b(\mathbf{p})$:

$$\chi^2_{\mathcal{B}}(\mathbf{p}) = \frac{1}{|\mathcal{B}|} \sum_{b \in \mathcal{B} \subset \mathcal{O}} \frac{(f_b(\mathbf{p}) - \mathcal{R}_b)^2}{\Delta f_b(\mathbf{p})^2 + \Delta \mathcal{R}_b^2} . \tag{13}$$

Since this test statistic is calculated per bin and then summed over a subset of bins $\mathcal{B}$ to get the total test statistic, we believe that the $\chi^2$ hypothesis test is appropriate. For this statistic, we hypothesize that:

Null hypothesis $H_0$: $f_b(\mathbf{p}) = \mathcal{R}_b$,

Alternate hypothesis $H_1$: $H_0$ is rejected, i.e., $f_b(\mathbf{p}) \neq \mathcal{R}_b$ .

In (13), we have a sample of size $|\mathcal{B}|$ based on which we compute the $\chi^2$ test statistic. However, the degrees of freedom of the $\chi^2$ distribution is not $|\mathcal{B}|$ because the samples $f_b(\mathbf{p}), b \in \mathcal{B} \subset \mathcal{O}$ are not independent and they are related to each other through the parameters $\mathbf{p}$. Due to this relationship, the number of degrees of freedom is reduced (see [30] for a similar argument). Hence the resulting degrees of freedom of the $\chi^2$ distribution for the set $\mathcal{B}$ is given by

$$\rho_{\mathcal{B}} = |\mathcal{B}| - d, \tag{14}$$

where $d$ is the dimension of $\mathbf{p}$.

We now choose a value for the significance level $\alpha$. In general, $\alpha$ is chosen by the user and commonly used values are 0.01, 0.05, or 0.1. For the results discussed in Section 4.5, we use 0.05. From a $\chi^2$ distribution table, we then obtain the critical value $\chi^2_{c,\mathcal{B}}$ for bins in $\mathcal{B}$ as a function of the significance level $\alpha$ and degrees of freedom $\rho_{\mathcal{B}}$. More formally, we say that if the probability $P_{H_0}(T \leq \chi^2_{c,\mathcal{B}}) = \alpha$, then under $H_0 : T \sim \chi^2(\rho_{\mathcal{B}})$. Let us assume a random variable $Z \sim \chi^2(\rho_{\mathcal{B}})$, then $P(Z \leq \chi^2_{c,\mathcal{B}}) = \alpha$. Thus, to find $\chi^2_{c,\mathcal{B}}$, we need to compute the inverse of the cumulative distribution function (CDF) of the $\chi^2$ distribution with $\rho_{\mathcal{B}}$ degrees of freedom and at level $\alpha$. Then we compare the test statistic with the critical value to decide whether $H_0$ is accepted or not, i.e., if $\chi^2_{\mathcal{B}} \leq \chi^2_{c,\mathcal{B}}$, we keep the bin subset $\mathcal{B}$; otherwise, we cannot keep this bin subset.

We mainly intend to exclude bins at the extremes of the observables, and hence we require that the bins we keep are contiguous. For some observables all bins may pass the $\chi^2$ test, for others, all bins may be excluded, or a subset of contiguous bins is kept.

The problem is then to find the largest contiguous subset of bins $\mathcal{B}$ such that $\chi^2_{\mathcal{B}} \leq \chi^2_{c,\mathcal{B}}$. This is equivalent to solving the mixed-integer program

$$\max_{s,e \in \{1,2,\ldots,|\mathcal{O}|\}} e - s$$
$$\text{s.t. } \chi^2_{\mathcal{B}} \leq \chi^2_{c,\mathcal{B}}, \quad \mathcal{B} = \{s, \ldots, e\}, \tag{15}$$

where $s$ and $e$ are the start and end indices of contiguous bins in observable $\mathcal{O}$. We want to note here that this optimization problem assumes that the constraint can be evaluated for all subsets $\mathcal{B}$ of the observable $\mathcal{O}$. Thus, the view of the hypothesis test from an optimization standpoint is the data required to check the satisfaction of the constraint, which will either lead to the rejection of the null hypothesis or the failure to reject the null hypothesis for each subset $\mathcal{B}$. Additionally, before starting the optimization, we would need to evaluate the $\chi^2_{\mathcal{B}}$ and

Table 2: Optimization methods used in this study.

| Name | Methodology | Reference |
|---|---|---|
| "Bilevel-portfolio" | bilevel optimization with portfolio outer objective function | Section 2.1.1. |
| "Bilevel-medianscore" | bilevel optimization with median score outer objective function | Section 2.1.2. |
| "Bilevel-meanscore" | bilevel optimization with mean score outer objective function | Section 2.1.2. |
| "Robust optimization" | single level robust optimization approach | Section 2.2. |
| "Expert" | weight adjustment done by the expert (only for the A14 dataset, see Section 4.3) | [3] |
| "All-weights-equal" | no optimization is used and all observable weights are set to 1 | |

$\chi^2_{c,\mathcal{B}}$ for all subsets $\mathcal{B}$ of observable $\mathcal{O}$. This can become tedious especially for observables with a large number of bins. To avoid this, we also propose a polynomial-time algorithm based on the maximum sub-array problem [31]. This algorithm is described in Section A.2 in the online supplement. In some cases, the bins to keep may not be unique, i.e., there may be multiple ranges of $\{s, \dots, e\}$ that are of the same maximum length and satisfy the null hypothesis (or satisfy the constraint in Eq. (15)). In practice, this is not a problem, since selecting any one of these bin subsets does not change the outcome of the filtering or the optimization in Eq. (2).

## 4 Numerical Experiments and Comparison of Different Tunes

In this section, we describe the setup of our numerical experiments, the datasets we use in our study, and the results. A comparison of the computation times required by the different optimizers is provided in Section 4.9. More details can be found in the online supplement.

### 4.1 Setup of the numerical experiments

We compare the results of using the methods shown in Table 2 for adjusting the weights of the observables in our datasets. The performance of each method is evaluated with and without data pre-processing (observable-filtering and bin-filtering approaches, see Sections 3.1 and 3.2), and when using a cubic polynomial (results presented in the online supplement) versus a rational approximation for $f_b(\mathbf{p})$ in APPRENTICE. We found relatively good performance using the degrees 3 and 1 for the numerator and denominator polynomial, respectively, for the rational approximation.

For the bilevel optimization formulation (see Eq. (3)), we made the following choices: The initial experimental design for the outer optimization has $|\mathcal{S}_\mathcal{O}| + 1$ points, where $|\mathcal{S}_\mathcal{O}|$ is the number of observables (number of weights to be adjusted). The total number of allowed outer objective function evaluations (number of weight vectors tried) is 1000. Because the inner optimization function is multimodal, we use 100 multi-starts with APPRENTICE to solve it. The bilevel optimization with each method (portfolio, meanscore, medianscore) is repeated three times with different random seeds and we report the results of the best run.

For the robust optimization formulation (Eq. (11)), a total of 100 random values of $\mu \in (0, 100]$ are used when evaluating Eq. (11c) and, for each $\mu$, the algorithm is run once. The best run amongst these is returned as the best $\mu$ for the robust optimization. The procedure to select the best $\mu$ is described in Section 2.2.

### 4.2 Comparison metrics and optimal tuning parameters

There are many ways to assess the quality of a tune. In many cases, the domain experts visually inspect a potentially large number of histograms to make a judgment. As an objective measure, we propose three metrics, each represented as a single number that can be used to compare the quality of the model fits obtained by the different methods:

1. *Weighted $\chi^2$*: the sum over all $\chi^2$ at the best $\widehat{\mathbf{p}}_{\mathbf{w}^*}$,

$$\sum_{\mathcal{O} \in \mathcal{S}_{\mathcal{O}}} w_{\mathcal{O}}^* \sum_{b \in \mathcal{O}} \frac{(f_b(\widehat{\mathbf{p}}_{\mathbf{w}^*}) - \mathcal{R}_b)^2}{\Delta f_b(\widehat{\mathbf{p}}_{\mathbf{w}^*})^2 + \Delta \mathcal{R}_b^2},$$

2. *A-optimality*:

$$\mathrm{Tr}\left(\mathbf{\Gamma}_{\mathrm{post}}(\widehat{\mathbf{p}}_{\mathbf{w}^*}, \mathbf{w}^*)\right) = \sum_{i=1}^{d} \lambda_i,$$

3. log *D-optimality*:

$$\log\det\left(\mathbf{\Gamma}_{\mathrm{post}}(\widehat{\mathbf{p}}_{\mathbf{w}^*}, \mathbf{w}^*)\right) = \sum_{i=1}^{d} \log \lambda_i,$$

where $\lambda_i$ are the eigenvalues of $\mathbf{\Gamma}_{\mathrm{post}}$, $\mathbf{\Gamma}_{\mathrm{post}}$ is the weighted posterior covariance matrix in the Bayesian formulation of the inverse problem, $d$ is the dimension of $\widehat{\mathbf{p}}_{\mathbf{w}^*}$. To find $\mathbf{\Gamma}_{\mathrm{post}}$, we compute the optimal parameter point $\widehat{\mathbf{p}}_{\mathbf{w}^*}$, which is also referred to as the maximum a posteriori probability estimate in the context of Bayesian inverse problems [32]. Given the optimal parameters, we can find a linearization of the model as

$$\mathbf{F}_{\mathcal{O}}(\widehat{\mathbf{p}}_{\mathbf{w}^*}) = \left[ \frac{\partial f_1(\widehat{\mathbf{p}}_{\mathbf{w}^*})}{\partial \mathbf{p}}, \frac{\partial f_2(\widehat{\mathbf{p}}_{\mathbf{w}^*})}{\partial \mathbf{p}}, \ldots, \frac{\partial f_{|\mathcal{O}|}(\widehat{\mathbf{p}}_{\mathbf{w}^*})}{\partial \mathbf{p}} \right]^\top,$$

for each observable $\mathcal{O}$. Then the weighted posterior can be approximated by a Gaussian $\mathcal{N}\left(\widehat{\mathbf{p}}_{\mathbf{w}^*}, \mathbf{\Gamma}_{\mathrm{post}}\right)$. Here

$$\mathbf{\Gamma}_{\mathrm{post}}(\widehat{\mathbf{p}}_{\mathbf{w}^*}, \mathbf{w}^*) = \left( \sum_{\mathcal{O} \in \mathcal{S}_{\mathcal{O}}} w_{\mathcal{O}}^* \mathbf{F}_{\mathcal{O}}^\top(\widehat{\mathbf{p}}_{\mathbf{w}^*}) \mathbf{\Gamma}_{\mathrm{noise}}^{-1} \mathbf{F}_{\mathcal{O}}(\widehat{\mathbf{p}}_{\mathbf{w}^*}) \right)^{-1}, \tag{16}$$

where $\mathbf{\Gamma}_{\mathrm{noise}}[\mathcal{O}] = \mathrm{diag}\left(\Delta f_1(\widehat{\mathbf{p}}_{\mathbf{w}^*})^2 + \Delta \mathcal{R}_1^2, \Delta f_2(\widehat{\mathbf{p}}_{\mathbf{w}^*})^2 + \Delta \mathcal{R}_2^2, \ldots, \Delta f_{|\mathcal{O}|}(\widehat{\mathbf{p}}_{\mathbf{w}^*})^2 + \Delta \mathcal{R}_{|\mathcal{O}|}^2\right)$. In the computation of all three metrics, $w_{\mathcal{O}}^*$ is the weight of observable $\mathcal{O}$ obtained from the methods and is scaled such that $w_{\mathcal{O}}^* \in [0, 1]$ and $\sum_{\mathcal{O} \in \mathcal{S}_{\mathcal{O}}} w_{\mathcal{O}}^* = 1$.

The $\mathbf{\Gamma}_{\mathrm{post}}(\widehat{\mathbf{p}}_{\mathbf{w}^*}, \mathbf{w}^*)$ calculated at the optimal parameters and the optimal weights in (16) are used here to describe the confidence region around the tuned parameters $\widehat{\mathbf{p}}_{\mathbf{w}^*}$. In order to summarize the multidimensional nature of $\mathbf{\Gamma}_{\mathrm{post}}$ into a scalar quantity, we use the A- and log D-optimality criteria. A graphical representation of the optimality criteria is shown in Figure 5.1 of [33]. The A-optimality criterion computes the trace of $\mathbf{\Gamma}_{\mathrm{post}}$, which is equivalent to the sum of its eigenvalues. This metric is proportional to the sum of the semiaxis lengths of the confidence ellipsoid of the parameters (lower is better), which corresponds to the average sum of the variances of the estimated parameters for the model [34]. The log D-optimality criterion computes the log of the determinant of $\mathbf{\Gamma}_{\mathrm{post}}$, which is equivalent to the sum of the log of the eigenvalues of $\mathbf{\Gamma}_{\mathrm{post}}$. This metric is proportional to the log volume of the confidence ellipsoid of the parameters (lower is better) [35]. It can be interpreted in terms of Shannon information. Note that since the weighted posterior is approximated as a Gaussian, a Gaussianity test should reveal that the posterior is normally distributed.

## 4.3 The A14 dataset

We chose the A14 tune [3] of the PYTHIA[3] event generator [36] as one benchmark for developing and testing the methods proposed in this work. This tune has been widely used for Large Hadron Collider (LHC) simulations, and is thus relevant to the particle physics community.

---

[3]To match the original study, we used version v8.186.

Table 3: Parameter values for A14 published tune (left), and A14 corrected expert tune and corresponding eigentune 68% confidence intervals (right).

| | A14 published expert tune | A14 corrected expert tune | | |
|---|---|---|---|---|
| Parameter name | NNPDF | Expert | min | max |
| SigmaProcess:alphaSvalue | 0.140 | 0.143 | 0.075 | 0.193 |
| BeamRemnants:primordialKThard | 1.88 | 1.904 | 1.903 | 1.906 |
| SpaceShower:pT0Ref | 1.56 | 1.643 | 1.636 | 1.653 |
| SpaceShower:pTmaxFudge | 0.91 | 0.908 | 0.905 | 0.912 |
| SpaceShower:pTdampFudge | 1.05 | 1.046 | 1.044 | 1.048 |
| SpaceShower:alphaSvalue | 0.127 | 0.123 | 0.121 | 0.124 |
| TimeShower:alphaSvalue | 0.127 | 0.128 | 0.043 | 0.197 |
| MultipartonInteractions:pT0Ref | 2.09 | 2.149 | 1.665 | 2.543 |
| MultipartonInteractions:alphaSvalue | 0.126 | 0.128 | 0.068 | 0.177 |
| BeamRemnants:reconnectRange | 1.71 | 1.792 | 1.788 | 1.795 |

The A14 dataset contains 406 observables (thus, 406 weights to optimize) and there are 10 tunable physics parameters $\mathbf{p}$. The parameters are primarily related to the production of additional jets in the collisions, the distribution of energy within those jets, and the kinematics (angles and momenta) of the jets. They also relate to the sharing and spread of energy in the soft portion of the event, the portion that is less dependent on the hard process (e.g., top-quark production or $Z$-boson production). Further explanation of the generator parameters and settings are available in Sections A.3 and A.16, respectively. The bounds over which we optimize the parameters were carefully chosen such that the polynomial parameterizations are valid within the bounds and to give a physically meaningful coverage such that the experimentally observed data was "covered" by the range of predictions. We use the RIVET [37] package to compare our predictions to data. The motivation for and selection of observables and parameters is explained in the A14 tune paper.

Because the coefficients of the cubic interpolation used in [3] were not available to us, we start by reproducing the hand-tuned parameter values published in [3, Table 3], which we refer to as *NNPDF*. In particular, we use the weights given in [3, Table 2], use their optimal parameter values as a starting point for the $\chi^2$ minimization, and apply our optimizer to Eq. (2). The resulting parameter values are reassuringly close to the values reported in [3] as shown in Table 3 where we label the original parameters as NNPDF, and the re-optimized parameter values as *Expert*. We observe that most of the NNPDF parameter values lie within the confidence interval derived from eigentunes (see Section 5) for the re-optimized Expert values. Additionally, to check whether the parameters $\mathbf{p}$ reported in [3] are within the confidence ellipsoid centered on the parameters $\widehat{\mathbf{p}}_{\mathbf{w}}$ obtained from the $\chi^2$ minimization (i.e., Expert parameter values), we calculate $s \equiv \left\| \mathbf{L}^T(\mathbf{p} - \widehat{\mathbf{p}}_{\mathbf{w}}) \right\|_2$, where $\mathbf{L}$ is the Cholesky factor of $\mathbf{\Gamma}_{\text{post}}(\widehat{\mathbf{p}}_{\mathbf{w}}, \mathbf{w})$ from Eq. (16) with weights $\mathbf{w}$ given in [3]. Since $s = 2.73 \times 10^{-3}$ is less than one, we say that the parameter $\mathbf{p}$ is covered within the confidence ellipsoid centered on $\widehat{\mathbf{p}}_{\mathbf{w}}$ [38].

In the remainder of this paper, we use the Expert parameter values for comparison. This provides a fairer comparison because we found that the original NNPDF parameter values did not correspond to a minimizer of the $\chi^2$ optimization, Eq. (2). The main reason for this discrepancy is the fact that we use an improved optimization routine (APPRENTICE) that takes advantage of exact gradient and Hessian information and that requires significantly less time than the previous optimizer, and thus allows for an efficient multistart local optimization that increases the possibility to find better optima.

The A14 observables are measurements of properties of proton-proton collisions at $\sqrt{s} = 7$ TeV performed by the ATLAS collaboration. These include event properties (e.g., the $Z$-boson transverse momentum, or the opening angles between the highest transverse mo-

mentum jets in the event) and properties of jets (e.g., the spread of energy within a jet, or the momentum of particles within a jet). In [3], the 406 observables are categorized into 10 groups (see Table 7), namely *Track jet* properties (200 observables) [39], *Jet shapes* (59 observables) [40], *Dijet decorr* (9 observables) [41], *Multijets* (8 observables) [42], $p_T^Z$ (fit range < 50GeV, 20 observables) [43, 44], *Substructure* (36 observables) [45], $t\bar{t}$ *gap* (4 observables) [46], $t\bar{t}$ *jet shapes* (20 observables) [47], *Track-jet UE* (8 observables) [48, 49], and *Jet UE* (42 observables) [50, 51]. The highest weights in [3] are assigned to observables that relate to the production of additional high-momentum partons (the ratios of 3-jet to 2-jet events, and the fraction of top-quark production events that do not have an additional central jet). On the other hand, low weights are assigned to observables that measure the same physical phenomenon in several kinematic regimes. The weighting of these observables ensures that the additional radiation and soft part of the events are consistent and well-modeled for all hard processes. In addition, these parameters are difficult or impossible to constrain using data from $e^+e^-$ collision events, and they must be tuned using data from the LHC.

### 4.4 The SHERPA dataset

As a second benchmark, we tune a set of parameters for the SHERPA event generator [52]. To our knowledge, the default parameters were not optimized by weighting data, and thus serve as an unbiased cross-check of our results. The data are confined to observables at $e^+e^-$ colliders and they include event shapes and charged particle inclusive spectra from $Z$-boson decays, differential and integrated jet rates, measurements of $B$-hadron fragmentation, and the multiplicity of various hadrons [53–56]. Accordingly, the parameters are limited to those of the SHERPA hadronization model.

The SHERPA dataset contains 88 observables, hence 88 weights to optimize. This is significantly less than the set of observables available in the RIVET analyses (126) for the following reasons. First, we reduce the number of observables to 114 by removing those that measure more than 3 jets, since this is beyond the scope of the physics simulation. Then, we apply a pre-filter step that removes distributions where *none* of the data bins fall within the envelope of predictions from our surrogate model. These all correspond to single-bin particle counts (such as the number of $f_0$ mesons) that the SHERPA hadronization model either grossly under- or over-estimates. There are 13 tunable physics parameters whose definition and ranges are shown in Table 16 in Section A.3 of the online supplement. These parameters are all part of the cluster model that produces physical particles from quarks and gluons. The reason for including this dataset in our study is to show the general applicability of our optimizers and to try them out on a dataset for which an expert tune is not provided.

### 4.5 Data pre-processing: filtering out observables and bins

In this subsection, we present the results of applying the filtering methods. First, we consider the outlier detection method described in Section 3.1. We find that the filtering results differ based on the choice of surrogate function (cubic polynomial versus a rational approximation). Based on the comparison of surrogate function predictions to the full MC simulations, we believe that the rational approximation yields a more faithful representation. Therefore, we present our main results using only the rational approximation. The names of the outlier observables in the A14 and the SHERPA dataset using a cubic polynomial and a rational approximation, respectively, are shown in the online supplement in Sections A.5 and A.6. Table 4 shows a distribution of the $\chi^2_{\mathcal{O}}$ values obtained for each observable $\mathcal{O}$ from Eq. (12) for A14 (left) and SHERPA (right) when using the rational approximation. We find that the per-observable ideal parameters yield mostly small $\chi^2_{\mathcal{O}}$ values (in $[0,1)$), but outliers are present in both datasets. Using the rational approximation, 9 and 3 outlier observables are filtered

Table 4: Distribution of the $\chi^2_{\mathcal{O}}$ values for A14 (left) and SHERPA (right). 2.0438 and 2.0177 correspond to the values where the Z-score equals 3 (see Section 3.1). The observables with $\chi^2_{\mathcal{O}} \geq 2.0438$ for A14 and $\chi^2_{\mathcal{O}} \geq 2.0177$ for SHERPA are the outliers. There are 9 outliers (6+2+1) in A14 and 3 outliers (1+2+0) in SHERPA.

| A14 | | SHERPA | |
|---|---|---|---|
| $\chi^2_{\mathcal{O}}$ range | Number of observables | $\chi^2_{\mathcal{O}}$ range | Number of observables |
| $[0, 1)$ | 367 | $[0, 1)$ | 82 |
| $[1, 2.0438)$ | 30 | $[1, 2.0177)$ | 3 |
| $[2.0438, 3)$ | **6** | $[2.0177, 3)$ | **1** |
| $[3, 4)$ | **2** | $[3, 4)$ | **2** |
| $[4, 5)$ | **1** | $[4, 5)$ | **0** |

from the A14 and SHERPA datasets, respectively.

Figure 1 shows the outcomes of the bin-filtering approach described in Section 3.2 for each observable $\mathcal{O}$ in A14 (top) and SHERPA (bottom) when using the rational approximation. In both datasets, multiple bins are removed. More specifically, most bins are removed in the *Track jet properties* and $p_T^Z$ groups of the A14 dataset. The patterns in the A14 plot result from the partitioning of the data. For *Tracked jet properties* (labeled A), the observables are replicated for two values of jet cone size ($R = 0.4, 0.6$), explaining the similarities between bins (1, 100) and (101, 200). Furthermore, 4 types of observables are considered, and each is sliced into different ranges of transverse momentum and rapidity.

In the SHERPA dataset, all bins are removed from some observables whereas from two observables, we remove only two and five bins (see observables in bold font in Table 20). Additionally, since the number of degrees of freedom of the $\chi^2$ distribution is reduced by the number of parameters that the bins share in each observable (see Eq. (14)), the bin filter is not applied to any observable with fewer than 10 and 13 bins in the A14 and the SHERPA datasets, respectively. The names of the observables from which the bins have been filtered and their $\chi^2$ test statistic and critical $\chi^2$ values are given in Sections A.7 and A.8 of the online supplement. The single bin observables correspond to counts of a particular type of particle.

## 4.6 Results for the A14 dataset

In this section, we present a detailed analysis of our results with the A14 dataset.

### 4.6.1 Comparison metric outcomes for the A14 dataset

In this section, we consider the three metrics introduced in Section 4.2 to compare various tunes. For the A14 dataset, Table 5 shows the results when using the rational approximation for the full data, the observable-filtered data, and the bin-filtered data, respectively. The results when using the cubic polynomial approximation are shown in the online supplement in Section A.12.1. Note that smaller numbers indicate better performance.

Based on these results we can see that no method performs the best for all metrics in all cases. In fact, for the full dataset, the *Expert* tune has the best score for two of our three metrics. The robust optimization consistently achieves the best performance under the Weighted $\chi^2$ criterion.

The Bilevel-portfolio method performs the best under the A-optimality criteria, and the *Expert* tune performs the best under the D-optimality criteria for the observable-filtered datasets. The Bilevel-portfolio method performs the best under the A- and D-optimality criteria for the bin-filtered datasets. In comparison to the results obtained with the cubic polynomial approxi-

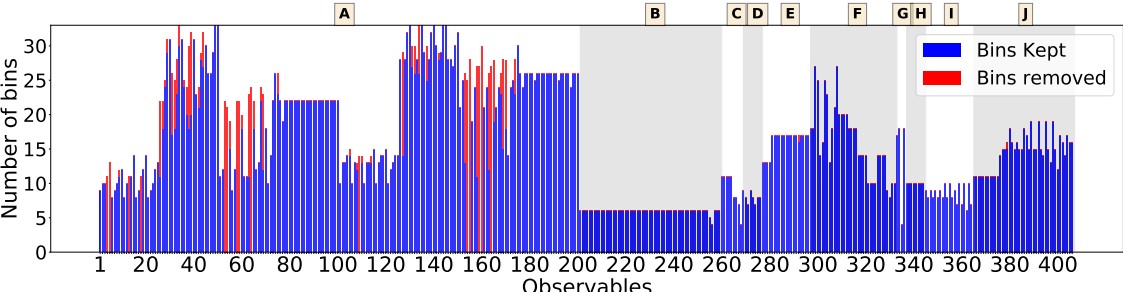

(a) Bins kept and removed by the bin filter in all A14 observables organized by the observable group. Group A is *Track jet properties*, group B is *Jet shapes*, group C is *Dijet decorr*, group D is *Multijets*, group E is $p_T^Z$, group F is *Substructure*, group G is $t\bar{t}$ *gap*, group H is *Track-jet UE*, group I is $t\bar{t}$ *jet shapes*, and group J is *Jet UE*.

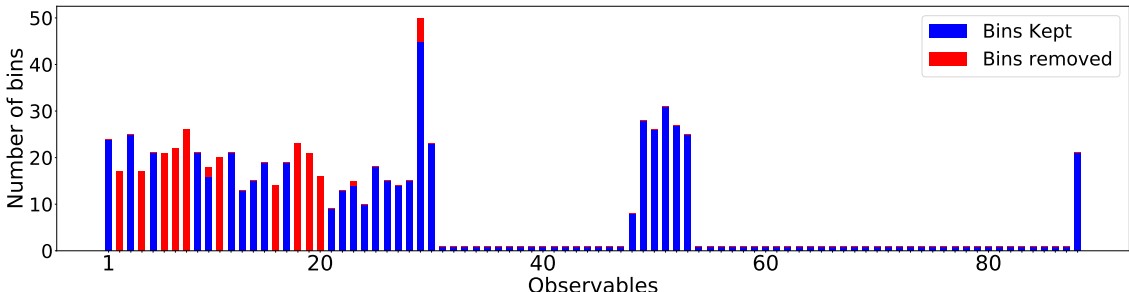

(b) Bins kept and removed by the bin filter in all SHERPA observables.

Figure 1: Illustration of the bin filtering results.

mation (see Section A.12.1 of the online supplement), the rational approximation yields better results for all methods under the Weighted $\chi^2$ criterion.

When comparing across Table 5, we see that in most cases, results with the observable-filtered data and bin-filtered data provide smaller values compared with those using the full dataset. We observe that by filtering out the observables and bins that cannot be well explained by the model, there is an improvement in the best values (in bold) of the metrics. This is an expected result because the excluded bins and observables no longer "distract" the optimizer by yielding large errors and thereby dominating the optimization. The selection of the best optimization method depends on the goals and preferences of the user as there is no one method that performs best for all metrics (no free lunch). However, we note here that lower A- log D-optimality values in the observable- and bin-filtered case indicate more confidence in the parameter predictions. We show in Section A.9 that by excluding the filtered bins and observables from the fitting process, the quality of the model does not deteriorate.

Table 5: A14 results with the *full dataset*, *observable-filtered dataset* and *bin-filtered dataset* when using the *rational* approximation. Lower numbers are better. The best results are in bold. In each dataset, W-$\chi^2$ refers to the Weighted $\chi^2$ metric, A-o refers to the A-opt metric, and l-D-o refers to the log D-opt metric.

| Data | Full dataset | | | Observable-filtered dataset | | | Bin-filtered dataset | | |
|---|---|---|---|---|---|---|---|---|---|
| Method | W-$\chi^2$ | A-o | l-D-o | W-$\chi^2$ | A-o | l-D-o | W-$\chi^2$ | A-o | l-D-o |
| Meanscore | 0.1119 | 0.8513 | -63.6805 | 0.0671 | 0.6793 | -65.1939 | 0.0923 | 0.7738 | -64.8949 |
| Medscore | 0.1320 | 0.7673 | -63.3846 | 0.0823 | 0.7008 | -64.3410 | 0.1175 | 0.7734 | -64.0170 |
| Portfolio | 0.1224 | 0.9425 | -61.1694 | 0.1372 | **0.5130** | -68.0382 | 0.1652 | **0.4788** | **-68.8998** |
| Expert | 0.0965 | **0.5705** | **-68.4091** | 0.0781 | 0.5765 | **-68.4674** | 0.0947 | 0.5868 | -68.3093 |
| Equal-weights | 0.0815 | 0.7673 | -64.0008 | 0.0563 | 0.7179 | -64.5198 | 0.0640 | 0.7384 | -65.2606 |
| Robust opt | **0.0402** | 1.0526 | -65.7547 | **0.0388** | 1.1086 | -65.7182 | **0.0485** | 0.8445 | -67.3645 |

### 4.6.2 Comparison of the cumulative distribution of bins at different variance levels

In this section, we introduce a new summarized graphical comparison of the results. We study the distribution of the $\chi^2$ values per bin obtained using different tuning approaches. For each parameter set, we compute the ratio $r_b(\mathbf{p}) = \dfrac{(f_b(\mathbf{p}) - \mathcal{R}_b)^2}{\Delta f_b(\mathbf{p})^2 + \Delta \mathcal{R}_b^2}$ of the residual between the data and the prediction divided by the variance per bin. The $r_b$ values are sorted from the smallest to largest, and the cumulative distribution is formed.

The cumulative distribution plot for all bins in the A14 dataset is shown in Figure 2 and for the bins in each category in Figure 3. The more bins that reside on the bands of variance levels less than 1 the better, as this indicates smaller deviations of the model from the experimental data. Note that even though all the category plots have a scale between 0 and 1 on the y-axis, the number of bins in the individual categories of A14 are very different, e.g., more than 50% of all bins in the A14 dataset belong to *Track Jet Properties*. Note, however, that we can see from the optimal weights assigned to each observable category (see Table 7, Section 4.6.4), the robust optimization approach is able to recognize the redundancy in the data and gives the observables in three of four subcategories little to no weight. On the other hand, the goal of the bilevel approaches is to fit each observable approximately equally well and the optimal weights mimic the expert's hand-tuning.

It can be seen from Figure 2 that there is only a small difference among the approaches when all A14 bins are considered. Near the variance boundary, the difference between the approaches is even smaller and all approaches perform better than the *Expert* tune. For sample data $x$ distributed normally as $\mathcal{N}(\mu, \sigma)$, the $\chi^2$ distribution with one degree of freedom is a distribution of the squared standard normal deviate given by $((x - \mu)/\sigma)^2$ [57]. Hence, for a normally distributed sample, the CDF of the bins with variance values $r_b(\mathbf{p})$ should theoretically follow a $\chi^2$ distribution with one degree of freedom. We compare the CDF of the bins against the CDF of this theoretical distribution in Figure 2. We observe that the CDF of the bins obtained from the different methods does not match the CDF of the theoretical distribution. In particular, we observe that bins whose residuals are $10^{-1} < r_b(\mathbf{p}) \leq 10^{1.5}$ arise from samples that are not normally distributed.

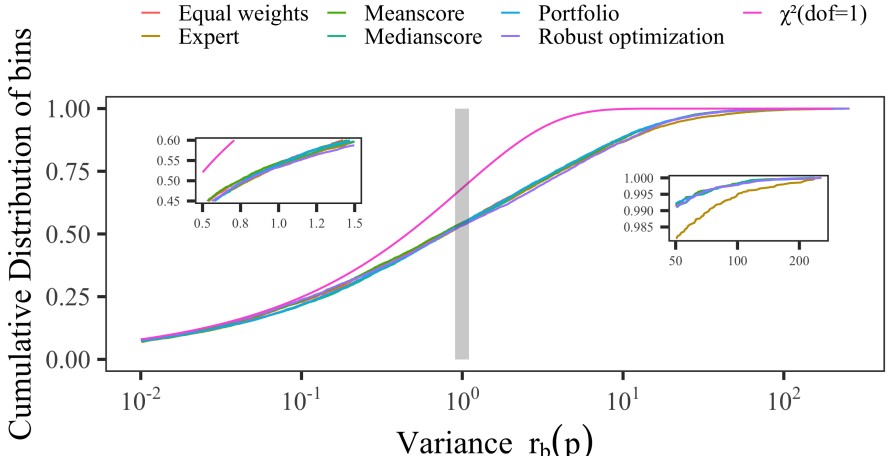

Figure 2: Cumulative distribution function (CDF) of all bins (y-axis) in the A14 dataset at different bands of variance levels (x-axis) given by $r_b(\mathbf{p}) = \dfrac{(f_b(\mathbf{p}) - \mathcal{R}_b)^2}{\Delta f_b(\mathbf{p})^2 + \Delta \mathcal{R}_b^2}$ and the theoretical $\chi^2$ distribution with one degree of freedom.

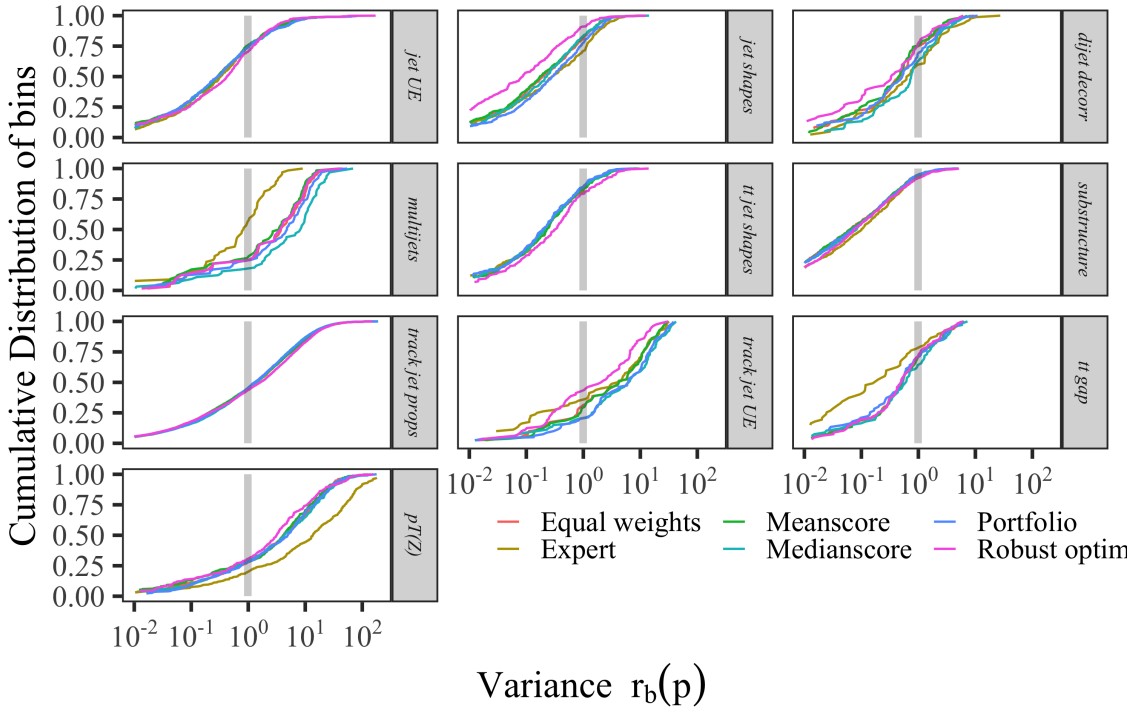

Figure 3: Cumulative distribution of bins (y-axis) in each category of the A14 dataset at different bands of variance levels (x-axis) given by $r_b(\mathbf{p}) = \frac{(f_b(\mathbf{p}) - \mathcal{R}_b)^2}{\Delta f_b(\mathbf{p})^2 + \Delta \mathcal{R}_b^2}$.

Figure 3 shows that these differences become more prominent for individual categories of the A14 data. For instance, robust optimization performs well for *Jet shapes* and *Track-jet UE*. Near the variance boundary, the parameters obtained from the *Expert* tune perform better for *Multijets* and $t\bar{t}$ *gap* whereas the other approaches perform better for *Substructure*. These plots also show that there is a trade-off in fitting among the different approaches, which enables the physicist to use these results as guidance for selecting the most appropriate tuning method depending on the categories that are of greater significance.

### 4.6.3 Optimal parameter values for the A14 dataset with rational approximation

The optimal parameter values for the A14 dataset when using the full dataset, the outlier-filtered dataset, and the bin-filtered dataset are shown in Table 6. For a better visual comparison of the different solutions obtained with our methods, we illustrate the [0,1]-scaled optimal values in the online supplement Section A.11. We have also computed the Euclidean distance between the *Expert* tune and our tunes after normalizing the parameter values to [0,1].

In Table 6, we can see that there are differences between the optimal parameters obtained with different methods and when using different filtering approaches. In particular, the results of the Bilevel-meanscore method tend to be approximately equally far from the expert solution no matter the filtering approach. This indicates that the bilevel-meanscore method is less sensitive to the data used in the optimization. The other methods show a larger variability of the optimal parameter values depending on the filtering approach. The eigentune results corresponding to the solutions in Table 6 are discussed in Section 5.

Table 6: Optimal parameter values for the A14 dataset when using the rational approximation in the optimization. Euclidean distances are calculated based on the normalized parameter values.

| | ID | Parameter name | Expert | Meanscore | Medianscore | Portfolio | Robust opt | Equal-weights |
|---|---|---|---|---|---|---|---|---|
| All observables | 1 | SigmaProcess:alphaSvalue | 0.143 | 0.138 | 0.133 | 0.136 | 0.139 | 0.137 |
| | 2 | BeamRemnants:primordialKThard | 1.904 | 1.855 | 1.723 | 1.796 | 1.883 | 1.851 |
| | 3 | SpaceShower:pT0Ref | 1.643 | 1.532 | 1.184 | 1.322 | 1.588 | 1.493 |
| | 4 | SpaceShower:pTmaxFudge | 0.908 | 1.014 | 1.083 | 1.041 | 1.025 | 1.026 |
| | 5 | SpaceShower:pTdampFudge | 1.046 | 1.071 | 1.084 | 1.061 | 1.084 | 1.067 |
| | 6 | SpaceShower:alphaSvalue | 0.123 | 0.128 | 0.129 | 0.128 | 0.127 | 0.128 |
| | 7 | TimeShower:alphaSvalue | 0.128 | 0.130 | 0.129 | 0.128 | 0.132 | 0.129 |
| | 8 | MultipartonInteractions:pT0Ref | 2.149 | 2.033 | 1.883 | 1.937 | 2.052 | 2.076 |
| | 9 | MultipartonInteractions:alphaSvalue | 0.128 | 0.124 | 0.118 | 0.120 | 0.126 | 0.125 |
| | 10 | BeamRemnants:reconnectRange | 1.792 | 2.082 | 1.914 | 1.987 | 2.602 | 1.980 |
| | | Euclidean distance from the expert solution | | 0.290 | 0.664 | 0.475 | 0.268 | 0.301 |
| Observable-filtered | 1 | SigmaProcess:alphaSvalue | 0.143 | 0.140 | 0.138 | 0.141 | 0.138 | 0.139 |
| | 2 | BeamRemnants:primordialKThard | 1.904 | 1.865 | 1.839 | 1.861 | 1.879 | 1.843 |
| | 3 | SpaceShower:pT0Ref | 1.643 | 1.574 | 1.603 | 1.593 | 1.614 | 1.550 |
| | 4 | SpaceShower:pTmaxFudge | 0.908 | 0.953 | 0.906 | 0.984 | 1.006 | 0.950 |
| | 5 | SpaceShower:pTdampFudge | 1.046 | 1.076 | 1.081 | 1.060 | 1.075 | 1.062 |
| | 6 | SpaceShower:alphaSvalue | 0.123 | 0.128 | 0.128 | 0.129 | 0.128 | 0.127 |
| | 7 | TimeShower:alphaSvalue | 0.128 | 0.123 | 0.123 | 0.118 | 0.132 | 0.124 |
| | 8 | MultipartonInteractions:pT0Ref | 2.149 | 2.064 | 2.017 | 2.095 | 2.022 | 2.039 |
| | 9 | MultipartonInteractions:alphaSvalue | 0.128 | 0.126 | 0.125 | 0.129 | 0.125 | 0.126 |
| | 10 | BeamRemnants:reconnectRange | 1.792 | 1.852 | 1.903 | 1.801 | 2.719 | 1.937 |
| | | Euclidean distance from the expert solution | | 0.227 | 0.293 | 0.273 | 0.291 | 0.254 |
| Bin-filtered | 1 | SigmaProcess:alphaSvalue | 0.143 | 0.139 | 0.140 | 0.131 | 0.137 | 0.140 |
| | 2 | BeamRemnants:primordialKThard | 1.904 | 1.877 | 1.885 | 1.811 | 1.822 | 1.876 |
| | 3 | SpaceShower:pT0Ref | 1.643 | 1.572 | 1.561 | 2.227 | 1.426 | 1.627 |
| | 4 | SpaceShower:pTmaxFudge | 0.908 | 0.964 | 0.968 | 0.869 | 0.948 | 0.943 |
| | 5 | SpaceShower:pTdampFudge | 1.046 | 1.056 | 1.053 | 1.481 | 1.053 | 1.068 |
| | 6 | SpaceShower:alphaSvalue | 0.123 | 0.128 | 0.128 | 0.136 | 0.128 | 0.128 |
| | 7 | TimeShower:alphaSvalue | 0.128 | 0.128 | 0.129 | 0.126 | 0.136 | 0.130 |
| | 8 | MultipartonInteractions:pT0Ref | 2.149 | 2.028 | 2.175 | 2.338 | 1.931 | 2.080 |
| | 9 | MultipartonInteractions:alphaSvalue | 0.128 | 0.124 | 0.128 | 0.135 | 0.120 | 0.126 |
| | 10 | BeamRemnants:reconnectRange | 1.792 | 2.047 | 1.854 | 1.820 | 2.404 | 2.001 |
| | | Euclidean distance from the expert solution | | 0.232 | 0.179 | 1.076 | 0.426 | 0.194 |

### 4.6.4 Comparison of optimal weights for the A14 dataset with rational approximation

We compare the optimal weights obtained by the different tuning methods in Table 7. We normalize the weights obtained to match the scale of weights assigned by *Expert* published in [3]. In each group, we report the average weight of observables in that group. The *Expert* tune assigned the highest weights to the categories *Multijets* and $t\bar{t}$ *gap*, which result in better fits as illustrated in the corresponding plots in Figure 5. The weights for the robust optimization approach are almost all either 0 or 17.85, which corresponds to unscaled $0-1$ weights that we would expect from this approach. We note that the weights for the four *Track-jet properties* classes are similar for the expert and the bilevel approaches (approx. 10), while the robust approach returns weights of $(17.85, 0, 1.62, 0)$. We believe that these weights indicate that the corresponding observables are nearly dependent resulting in redundant components of least-square residuals. We observe in Figure 3 that setting these weights to zero does not degrade the residuals of these observables, confirming that redundant information is present. This observation indicates that even though *Track jet properties* dominates the tune in terms of the number of observables, the inherent redundancy in the data does not dominate the final fit, and can be detected by the robust optimization approach.

### 4.6.5 Impact of data pre-processing by filtering on optimal results

In Table 8, we show the number of filtered and unfiltered bins in the A14 and SHERPA datasets that lie within a one $\sigma$ variance level. A large number of bins within a one $\sigma$ level indicates smaller deviations of the model from the experimental data. The cumulative distribution plot

Table 7: Comparison of the optimal weights obtained by each method using the rational approximation. The observable grouping corresponds to the same grouping as in [3].

| | Expert | Bilevel-meanscore | Bilevel-medianscore | Bilevel-portfolio | Robust opt |
|---|---|---|---|---|---|
| **Track jet properties** | | | | | |
| Charged jet multiplicity (50 distributions) | 10 | 11.41 | 11.92 | 11.43 | 17.85 |
| Charged jet $z$ (50 distributions) | 10 | 11.01 | 10.00 | 10.28 | 0.00 |
| Charged jet $p_T^{rel}$ (50 distributions) | 10 | 9.47 | 10.20 | 13.11 | 1.62 |
| Charged jet $\rho_{ch}(r)$ (50 distributions) | 10 | 10.63 | 12.72 | 12.19 | 0.00 |
| **Jet shapes** | | | | | |
| Jet shape $\rho$ (59 distributions) | 10 | 12.46 | 8.49 | 9.69 | 17.85 |
| **Dijet decorr** | | | | | |
| Decorrelation $\Delta\phi$ (Fit range: $\Delta\phi > 0.75$) (9 distributions) | 20 | 18.82 | 10.32 | 18.50 | 15.87 |
| **Multijets** | | | | | |
| 3-to-2 jet ratios (8 distributions) | 100 | 15.06 | 11.18 | 11.06 | 17.85 |
| $p_T^Z$ (Fit range: $p_T^Z < 50 \text{GeV}$) | | | | | |
| Z-boson $p_T$ (20 distributions) | 10 | 12.16 | 11.85 | 9.25 | 17.85 |
| **Substructure** | | | | | |
| Jet mass, $\sqrt{d_{12}}, \sqrt{d_{23}}, \tau_{21}, \tau_{23}$ (36 distributions) | 5 | 10.71 | 12.75 | 14.23 | 17.85 |
| $t\bar{t}$ **gap** | | | | | |
| Gap fraction vs $Q_0$, $Q_{\text{sum}}$ for $|y| < 0.8$ | 100 | 24.56 | 5.05 | 1.97 | 17.85 |
| Gap fraction vs $Q_0$, $Q_{\text{sum}}$ for $0.8 < |y| < 1.5$ | 80 | 23.73 | 47.01 | 4.01 | 17.85 |
| Gap fraction vs $Q_0$, $Q_{\text{sum}}$ for $1.5 < |y| < 2.1$ | 40 | 2.39 | 14.20 | 7.35 | 17.85 |
| Gap fraction vs $Q_0$, $Q_{\text{sum}}$ for $|y| < 2.1$ | 10 | 5.47 | 19.00 | 12.82 | 17.85 |
| **Track-jet UE** | | | | | |
| Transverse region $N_{ch}$ profiles (5 distributions) | 10 | 13.01 | 24.18 | 7.46 | 17.85 |
| Transverse region mean $p_T$ profiles for $R = 0.2, 0.4, 0.6$ (3 distributions) | 10 | 7.91 | 16.89 | 9.68 | 17.85 |
| $t\bar{t}$ **jet shapes** | | | | | |
| Jet shapes $\rho(r), \psi(r)$ (20 distributions) | 5 | 10.44 | 11.47 | 10.29 | 15.17 |
| **Jet UE** | | | | | |
| Transverse, trans-max, trans-min sum $p_T$ incl. profiles (3 distributions) | 20 | 12.11 | 5.32 | 10.51 | 17.85 |
| Transverse, trans-max, trans-min $N_{ch}$ incl. profiles (3 distributions) | 20 | 6.16 | 14.42 | 6.56 | 17.85 |
| Transverse sum $E_T$ incl. profiles (2 distributions) | 20 | 5.11 | 2.71 | 7.72 | 17.85 |
| Transverse sum $ET/\text{sum } p_T$ ratio incl., excl. profiles (2 distributions) | 5 | 11.94 | 10.81 | 11.65 | 17.85 |
| Transverse mean $p_T$ incl. profiles (2 distributions) | 10 | 12.47 | 7.28 | 10.45 | 17.85 |
| Transverse, trans-max, trans-min sum $p_T$ incl. distributions (15 distributions) | 1 | 10.54 | 14.44 | 8.27 | 17.85 |
| Transverse, trans-max, trans-min sum $N_{ch}$ incl. distributions (15 distributions) | 1 | 11.62 | 10.33 | 11.48 | 17.85 |

with the parameters obtained from the robust optimization approach for filtered and unfiltered data for the different categories is shown in Figure 4 (the plots for the other methods are shown in Section A.9 of the online supplement).

From these results, we observe that there is no significant difference in the number of bins within the one $\sigma$ variance level between the optimal parameters $\mathbf{p}_a^*$ obtained when all bins were used for tuning and the optimal parameters $\mathbf{p}_b^*$ and $\mathbf{p}_o^*$ obtained when only the bin filtered and observable filtered bins are used for tuning, respectively. Additionally, when comparing across Table 5, we see that in most cases, the results with the observable-filtered data and bin-filtered data provide smaller values in the proposed criteria compared with those using the full dataset. These observations indicate that the MC generator cannot explain the removed bins and that the information contained in these bins does not add significant information to the tune.

Table 8: Number of bins in the A14 and SHERPA datasets within the one $\sigma$ variance level. Larger numbers are better. The variance level for each bin is calculated as $r_b(\mathbf{p}) = \frac{(f_b(\mathbf{p}) - \mathcal{R}_b)^2}{\Delta f_b(\mathbf{p})^2 + \Delta \mathcal{R}_b^2}$. *Test data type* specifies the data over which $r_b(\mathbf{p})$ is calculated, where *All* means that all bins are used, *Not Filtered* refers to only the bins that remain after filtering, and *Filtered* refers to the bins that were filtered out by the respective filter specified in the *Filtering Method* as well as the envelope filter. For each data type, the number of bins in the corresponding dataset is also specified. *Parameters* specify the type of optimal parameters used in $r_b(\mathbf{p})$ where $\mathbf{p}_a^*$ are the parameters obtained when all bins were used during tuning whereas $\mathbf{p}_b^*$ and $\mathbf{p}_o^*$ the parameters obtained when only the bin filtered and observable filtered date are used, respectively.

| Dataset | Filtering method | Test data type | Parameters | Robust optimization | Bilevel-meanscore | Bilevel-medianscore | Bilevel-portfolio |
|---|---|---|---|---|---|---|---|
| A14 | Bin Filtered | All (# 7010) | $\mathbf{p}_a^*$ | 3730 | 3724 | 3687 | 3693 |
| | | | $\mathbf{p}_b^*$ | 3625 | 3775 | 3765 | 3573 |
| | | Not filtered (# 5199) | $\mathbf{p}_a^*$ | 3350 | 3317 | 3265 | 3273 |
| | | | $\mathbf{p}_b^*$ | 3248 | 3365 | 3342 | 3185 |
| | | Filtered (# 1811) | $\mathbf{p}_a^*$ | 380 | 407 | 422 | 420 |
| | | | $\mathbf{p}_b^*$ | 377 | 410 | 423 | 388 |
| | Observable Filtered | All (# 7010) | $\mathbf{p}_a^*$ | 3730 | 3724 | 3687 | 3693 |
| | | | $\mathbf{p}_o^*$ | 3732 | 3734 | 3695 | 3509 |
| | | Not filtered (# 6707) | $\mathbf{p}_a^*$ | 3675 | 3660 | 3624 | 3630 |
| | | | $\mathbf{p}_o^*$ | 3679 | 3672 | 3629 | 3444 |
| | | Filtered (# 303) | $\mathbf{p}_a^*$ | 55 | 64 | 63 | 63 |
| | | | $\mathbf{p}_o^*$ | 53 | 62 | 66 | 65 |
| SHERPA | Bin Filtered | All (# 792) | $\mathbf{p}_a^*$ | 320 | 337 | 371 | 256 |
| | | | $\mathbf{p}_b^*$ | 343 | 328 | 345 | 243 |
| | | Not filtered (# 588) | $\mathbf{p}_a^*$ | 272 | 283 | 317 | 214 |
| | | | $\mathbf{p}_b^*$ | 282 | 270 | 292 | 200 |
| | | Filtered (# 204) | $\mathbf{p}_a^*$ | 48 | 54 | 54 | 42 |
| | | | $\mathbf{p}_b^*$ | 61 | 58 | 53 | 43 |
| | Observable Filtered | All (# 792) | $\mathbf{p}_a^*$ | 320 | 337 | 371 | 256 |
| | | | $\mathbf{p}_o^*$ | 286 | 348 | 386 | 252 |
| | | Not filtered (# 727) | $\mathbf{p}_a^*$ | 304 | 319 | 355 | 237 |
| | | | $\mathbf{p}_o^*$ | 271 | 331 | 370 | 235 |
| | | Filtered (# 65) | $\mathbf{p}_a^*$ | 16 | 18 | 16 | 19 |
| | | | $\mathbf{p}_o^*$ | 15 | 17 | 16 | 17 |

### 4.6.6 Comparison of rational approximation and the MC simulator

Similar to the analysis conducted in Section 4.6.2, we compare the cumulative distribution of bins at different bands of variance levels computed using the approximation model as $r_b(\mathbf{p}) = \frac{(f_b(\mathbf{p}) - \mathcal{R}_b)^2}{\Delta f_b(\mathbf{p})^2 + \Delta \mathcal{R}_b^2}$ and the MC generator model as $\widetilde{r_b(\mathbf{p})} = \frac{(\mathrm{MC}_b(\mathbf{p}) - \mathcal{R}_b)^2}{\Delta \mathrm{MC}_b(\mathbf{p})^2 + \Delta \mathcal{R}_b^2}$, where $\mathbf{p}$ are the parameters obtained from the tuning approaches. The more bins that are on the bands of variance levels less than one, the better. Figure 5 shows the plot of this comparison for bins in each category of the A14 dataset.[4] To avoid making the plot too busy, we show the results using the parameters from three approaches. A similar plot showing the results with parameters from the remaining approaches is given in Section A.10 in the online supplement.

We observe in Figure 5 that the *Dijet decorr, Jet shapes, $p_T^Z$, Track-jet UE,* and *$t\bar{t}$ gap* categories show differences in the performance between $r_b(\mathbf{p})$ and $\widetilde{r_b(\mathbf{p})}$ for each approach. Additionally, for the *robust optimization* and *Bilevel-meanscore* approaches, this difference in the performance is not as wide as that of the *Expert* (for e.g., see $p_T^Z$, *Track-jet UE* categories). This suggests that (a) there are categories where the approximations are not able to capture the MC generator perfectly, and (b) in general, the rational approximation is a better surrogate for the MC generator than the polynomial approximation.

### 4.7 Results for the SHERPA dataset

In this section, we present the detailed results for the SHERPA dataset.

---

[4]The *Jet UE* comparison is missing from this figure because the internal ATLAS analysis is not available to us.

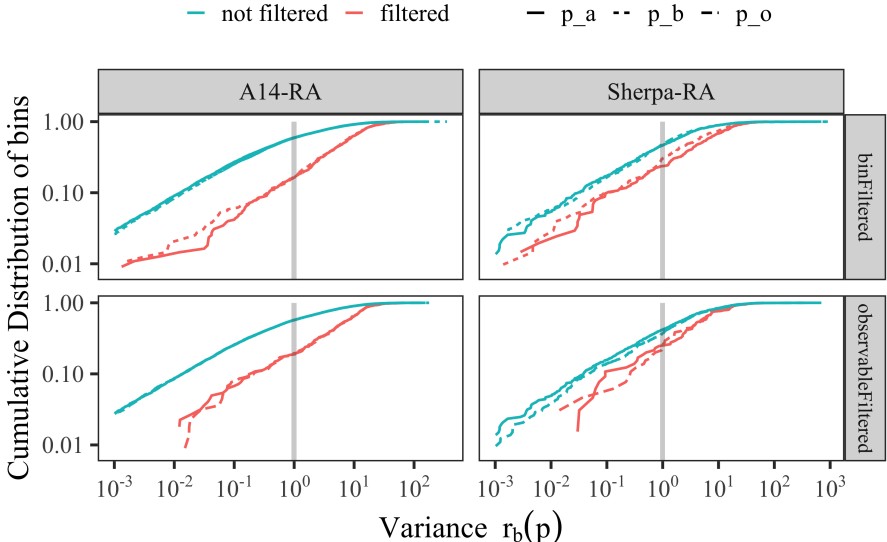

Figure 4: Cumulative distribution of bins remaining after filtering (*not filtered*) and of those filtered out (*filtered*) on the y-axis at different bands of variance levels on the x-axis. The variance level for each bin is calculated as $r_b(\mathbf{p}) = \frac{(f_b(\mathbf{p}) - \mathcal{R}_b)^2}{\Delta f_b(\mathbf{p})^2 + \Delta \mathcal{R}_b^2}$ with parameters $\mathbf{p}_a^*$, which is obtained when all bins were used, and parameters $\mathbf{p}_b^*$ and $\mathbf{p}_o^*$, which are obtained when only the bin filtered and observable filtered data are used, respectively.

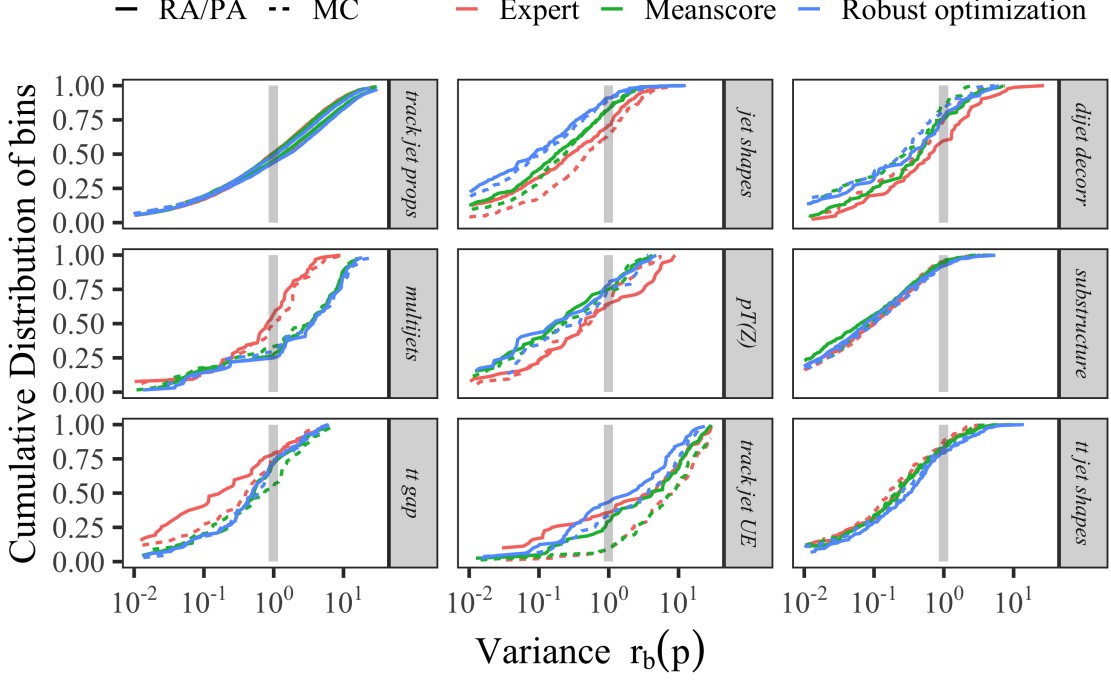

Figure 5: Cumulative distribution of bins (y-axis) in each category of the A14 dataset at different bands of variance levels (x-axis) computed with cubic polynomial approximation (PA) or rational approximation (RA) and the MC simulation.

Table 9: Results for the comparison metrics for the full, observable-filtered, and bin-filtered SHERPA dataset using the rational approximation. The best results are in bold. In each dataset, W-$\chi^2$ refers to the Weighted $\chi^2$ metric, A-o refers to the A-opt metric, and l-D-o refers to the log D-opt metric.

| Data | Full dataset | | | Observable-filtered dataset | | | Bin-filtered dataset | | |
|---|---|---|---|---|---|---|---|---|---|
| Method | W-$\chi^2$ | A-o | l-D-o | W-$\chi^2$ | A-o | l-D-o | W-$\chi^2$ | A-o | l-D-o |
| Meanscore | 0.2201 | 9.0147 | -39.3957 | 0.3621 | 11.1570 | -36.5249 | 0.1490 | 17.9602 | -33.5825 |
| Medscore | 0.2249 | 43.2031 | -25.7164 | 0.2315 | 13.0679 | -35.3498 | 0.2136 | 21.9361 | -31.4329 |
| Portfolio | 0.1510 | 11.9869 | -35.7488 | 0.4728 | **8.5578** | **-38.6042** | 0.1239 | 16.8518 | -35.2237 |
| Equal-weights | 0.2794 | **6.8428** | **-42.0325** | 0.3930 | 59.9885 | -18.8193 | 0.1753 | **11.5372** | **-36.0252** |
| Robust opt | **0.0603** | 55.8079 | -22.0884 | **0.0509** | 32.9470 | -30.5536 | **0.0919** | 17.9858 | -33.6522 |

### 4.7.1 Comparison metric outcomes for the SHERPA dataset

Table 9 shows the results when using the rational approximation (results for the cubic polynomial approximation are in the online supplement Section A.12.6). Smaller numbers indicate better performance. The smallest number of each metric is bold for better visualization. Similar to A14, we find that the robust optimization approach achieves the best performance in terms of the Weighted $\chi^2$ criterion. Assigning equal weights to all observables yields the best results in terms of A- and D-optimality for the full and the bin-filtered dataset. The portfolio approach yields the best A- and D-optimality values when using the observable-filtered dataset.

Compared with the results of A14, we see that the magnitudes of all metrics are much larger. The large A- and D-optimality values reflect that we have larger regions of uncertainty associated with the optimal parameters, and thus we have less confidence in the validity of the results obtained for the SHERPA dataset than for the A14 dataset.

### 4.7.2 Comparison of the cumulative distribution of bins at different variance levels

Similar to the analysis conducted in Section 4.6.2, we compare the cumulative distribution of bins at different bands of variance level computed using the optimal parameters **p** obtained from the tuning approaches (see Figure 6). The results show that fewer bins lie within the variance boundary of one when using the parameters of the bilevel-portfolio approach. On the other hand, the bilevel-medianscore approach finds parameters that yield the most bins at lower bands of variance levels.

### 4.7.3 Optimal parameter values for the SHERPA dataset with rational approximation

The optimal parameter values for the SHERPA dataset when no filtering, observable-filtering, and bin-filtering were applied, respectively, are shown in Table 10. For a visualization of the different solutions obtained with our methods, we illustrate the [0,1]-scaled optimal parameters in the online supplement Section A.12.4. We see that many of the parameters lie on the boundary of the parameter space (shown in the table in bold), indicating that we might need to change the size of the parameter domain to avoid model extrapolation.

Note that for the SHERPA dataset, we do not have an "expert" solution for benchmark comparison. Instead, we compare the solutions to the chosen reasonable default setting. The parameter range is constructed by multiplying the default value by 0.5 and 1.5 to obtain the lower and the upper bound, respectively, i.e., the default values lie in the middle of the parameter range. We see that there are differences between the optimal parameters obtained with the different methods, in particular, bilevel-medianscore gives a very similar solution to the default setting when no filtering is applied.

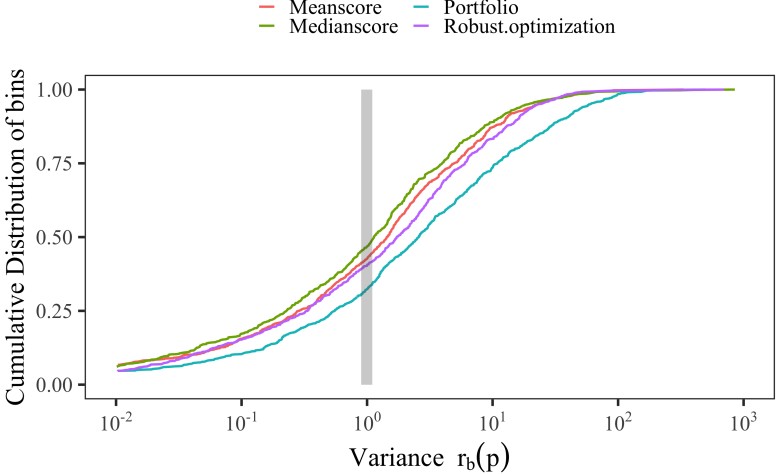

Figure 6: Cumulative distribution function (CDF) of all bins (y-axis) in the SHERPA dataset at different bands of variance levels (x-axis) given by $r_b(\mathbf{p}) = \frac{(f_b(\mathbf{p}) - \mathcal{R}_b)^2}{\Delta f_b(\mathbf{p})^2 + \Delta \mathcal{R}_b^2}$. This function is a normal CDF with mean 1 and a different standard deviation for each method.

Table 10: Optimal parameter values obtained with all methods using rational approximation when no filtering (88 observables used), observable-filtering (3 observables were filtered out), and bin-filtering (7 bins were filtered out) was applied. The parameter values on the boundaries of the parameter space are indicated in bold.

| | ID | Parameter name | Default | Mean-score | Median-score | Portfolio | Robust-opt | Equal-weights |
|---|---|---|---|---|---|---|---|---|
| **All observables** | 1 | KT_0 | 1.00 | 0.888 | 0.789 | 0.919 | 0.909 | 0.872 |
| | 2 | ALPHA_G | 1.25 | **0.626** | 1.500 | **0.626** | 1.874 | **0.626** |
| | 3 | ALPHA_L | 2.50 | **3.749** | 1.890 | **3.749** | 1.252 | **3.749** |
| | 4 | BETA_L | 0.10 | **0.150** | **0.050** | 0.087 | 0.150 | **0.150** |
| | 5 | GAMMA_L | 0.50 | 0.274 | 0.339 | **0.750** | 0.683 | 0.293 |
| | 6 | ALPHA_H | 2.50 | 3.400 | 2.897 | **1.251** | 2.841 | 3.440 |
| | 7 | BETA_H | 0.75 | 0.827 | 0.536 | 0.783 | 0.540 | 0.795 |
| | 8 | GAMMA_H | 0.10 | 0.148 | **0.050** | 0.082 | 0.150 | **0.150** |
| | 9 | STRANGE_FRACTION | 0.50 | 0.517 | 0.498 | 0.583 | 0.508 | 0.546 |
| | 10 | BARYON_FRACTION | 0.18 | 0.100 | 0.175 | 0.106 | 0.136 | **0.090** |
| | 11 | P_QS_by_P_QQ_norm | 0.48 | **0.720** | 0.419 | 0.572 | 0.613 | **0.720** |
| | 12 | P_SS_by_P_QQ_norm | 0.02 | **0.010** | 0.015 | **0.030** | 0.030 | **0.010** |
| | 13 | P_QQ1_by_P_QQ0 | 1.00 | **1.499** | 1.206 | 0.948 | 1.190 | **1.499** |
| | | Euclidean distance from the default solution | | 1.513 | 0.984 | 1.244 | 1.289 | 1.531 |
| **Observable-filtered** | 1 | KT_0 | 1.00 | 0.867 | 0.744 | 0.952 | 0.876 | 0.886 |
| | 2 | ALPHA_G | 1.25 | 0.775 | **0.626** | **0.626** | 0.626 | 0.957 |
| | 3 | ALPHA_L | 2.50 | **3.749** | 1.252 | **3.749** | 1.252 | 2.424 |
| | 4 | BETA_L | 0.10 | 0.109 | **0.050** | **0.050** | 0.150 | 0.113 |
| | 5 | GAMMA_L | 0.50 | **0.250** | 0.437 | 0.413 | 0.750 | 0.460 |
| | 6 | ALPHA_H | 2.50 | 3.053 | 2.318 | **1.251** | 2.826 | 3.132 |
| | 7 | BETA_H | 0.75 | 0.827 | 0.625 | 0.750 | 0.375 | 0.969 |

| | | | | | | | | |
|---|---|---|---|---|---|---|---|---|
| | 8 | GAMMA_H | 0.10 | **0.050** | 0.134 | 0.094 | 0.050 | 0.131 |
| | 9 | STRANGE_FRACTION | 0.50 | 0.479 | 0.580 | 0.651 | 0.506 | 0.511 |
| | 10 | BARYON_FRACTION | 0.18 | **0.270** | 0.137 | **0.090** | 0.137 | 0.180 |
| | 11 | P_QS_by_P_QQ_norm | 0.48 | **0.720** | 0.469 | 0.495 | 0.470 | 0.601 |
| | 12 | P_SS_by_P_QQ_norm | 0.02 | **0.010** | **0.030** | **0.030** | **0.030** | 0.019 |
| | 13 | P_QQ1_by_P_QQ0 | 1.00 | **0.500** | **1.499** | **1.499** | 1.499 | 0.958 |
| | | Euclidean distance from the default solution | | 1.408 | 1.249 | 1.372 | 1.446 | 0.637 |
| Bin-filtered | 1 | KT_0 | 1.00 | 0.895 | 0.821 | 0.948 | 0.820 | 0.899 |
| | 2 | ALPHA_G | 1.25 | 0.893 | 1.483 | **0.626** | 1.874 | **0.626** |
| | 3 | ALPHA_L | 2.50 | **3.749** | 2.334 | 2.567 | 3.749 | **3.749** |
| | 4 | BETA_L | 0.10 | 0.050 | **0.150** | 0.074 | 0.050 | 0.067 |
| | 5 | GAMMA_L | 0.50 | 0.390 | **0.250** | **0.750** | 0.250 | 0.454 |
| | 6 | ALPHA_H | 2.50 | **1.251** | 3.670 | **1.251** | 1.969 | **1.251** |
| | 7 | BETA_H | 0.75 | 0.715 | 0.534 | 0.739 | 1.125 | 0.715 |
| | 8 | GAMMA_H | 0.10 | 0.119 | 0.142 | 0.105 | 0.050 | 0.089 |
| | 9 | STRANGE_FRACTION | 0.50 | 0.556 | 0.542 | 0.570 | 0.531 | 0.559 |
| | 10 | BARYON_FRACTION | 0.18 | 0.122 | 0.120 | 0.124 | 0.138 | 0.124 |
| | 11 | P_QS_by_P_QQ_norm | 0.48 | 0.595 | **0.720** | 0.492 | 0.497 | 0.577 |
| | 12 | P_SS_by_P_QQ_norm | 0.02 | **0.030** | 0.030 | **0.030** | 0.030 | **0.030** |
| | 13 | P_QQ1_by_P_QQ0 | 1.00 | **1.499** | **1.499** | **1.499** | 1.499 | **1.499** |
| | | Euclidean distance from the default solution | | 1.266 | 1.377 | 1.201 | 1.462 | 1.242 |

The distribution of weights from the different methods has a similar pattern as for the tunes based on the A14 dataset. These patterns are displayed in Fig. 19 in the online supplement. Robust optimization selects only one of the event shape observables as relevant, while applying the same equal weight to most of the particle multiplicity (one bin) distributions. The other methods have weights that are more widely distributed among the observables with a small number of weights far from the average.

## 4.8 Closure test

In order to show that our proposed optimization methods are able to find the "correct" solutions, we construct a simple toy model with linear approximations that has two parameters and four observables. Each observable has five bins. The approximation $f_b(\mathbf{p})$ for each bin $b$ is a linear function of the form $\mathbf{a}^T \mathbf{p} + c$. The coefficients of the linear function are given in Section A.14. The deviation $\Delta f_b(\mathbf{p})$ is 0 for all bins. The experimental data is made up of standard deviation $\Delta \mathcal{R}_b$ and mean values $\mathcal{R}_b$ for each bin $b$. The standard deviation is a constant of 0.005 for all bins. The mean values of the bins are obtained by evaluating the linear function at known parameter values. For the bins in the first three observables, the parameter value of $\overline{\mathbf{p}} = (-0.7778, 0.2729)$ is used whereas for the bins in the fourth observable, the parameter value of $\widehat{\mathbf{p}} = (-0.0448, -0.3878)$ is used. We expect that the combined weight of the first three observables is larger than the weight of the fourth observable (with optimal tune $\overline{\mathbf{p}}$) since the number of bins that fit well to the experimental data is greater from the first three observables than from the fourth observable alone, thus resulting in lower objective value in the optimization algorithms.

For the bilevel optimization methods, we perform the outlier detection technique to see if the fourth observable will be removed. For the robust optimization method, we expect that the optimal weights should be [1,1,1,0], or equivalently, [0.3333, 0.3333, 0.3333, 0] after

normalization.

Table 11 shows that all proposed methods can recognize that the fourth observable should not be involved in the optimization, and all methods can find the optimal parameter tune $\bar{\mathbf{p}}$. The table also summarizes the comparison metric results obtained with all proposed methods, and the results show that the meanscore method performs the best under the Weighted $\chi^2$ metric, and the medianscore method performs the best under the A- and D-optimality criteria.

Table 11: Results for the closure test. Shown are the optimal weights obtained with each method, the optimal parameters, and the outcomes for our performance metrics.

|  | Bilevel-meanscore | Bilevel-medscore | Bilevel-portfolio | Robust optimization |
|---|---|---|---|---|
| Weights |  |  |  |  |
| Observable 1 | 0.8060 | 0.5485 | 0.2550 | 0.3333 |
| Observable 2 | 0.0070 | 0.3100 | 0.3663 | 0.3333 |
| Observable 3 | 0.1870 | 0.1415 | 0.3787 | 0.3333 |
| Observable 4 | 0 | 0 | 0 | 0 |
| Parameters |  |  |  |  |
| $p_0$ | -0.7778 | -0.7780 | -0.7775 | -0.7781 |
| $p_1$ | 0.2726 | 0.2729 | 0.2728 | 0.2731 |
| Performance metrics (lower numbers are better, best results are in bold) |  |  |  |  |
| Weighted $\chi^2$ | **0.5866** | 0.7631 | 0.9867 | 1.0023 |
| A-optimality | 3.21E-06 | **2.25E-06** | 2.74E-06 | 2.58E-06 |
| log-D-optimality | -29.6887 | **-30.0576** | -29.8999 | -29.9521 |

## 4.9 A note on computation times

The bilevel optimization approaches are run on a 4-core, 32 GB RAM machine running at 1.1 GHz. For the results of robust optimization, 100 values for $\mu$ are used that are run on 100 threads in parallel on a server with 64 Intel Xeon Gold CPU cores running at 2.30 GHz. There are two threads per core, but each run of robust optimization is done on a single thread. Additionally, this server is equipped with 1.5TB DDR4 2666 MHz of memory. A simple comparison to find the best $\mu$ takes one minute. The all-weights-equal approach is run on a 4-core, 32 GB RAM machine running at 1.1 GHz. Note that in our numerical experiments we were not primarily concerned with architecture-dependent run times, but rather to ensure that our codes for automated optimization can be executed on different architectures.

The time taken by all the tuning approaches for unfiltered (*All data*) as well as for bin-filtered and observable-filtered A14 data is given in Table 12. In the unfiltered data case, the bilevel optimization approaches take approximately 14.5 hours and each run (i.e., one $\mu$) of robust optimization takes an average of about 0.8 hours. Since all 100 values of $\mu$ were run in parallel, the total time to complete all 100 runs of robust optimization is approximately two hours. In comparison, campaigns to tune weights by hand takes many weeks or months. Given our results, we can see that the automated weight adjustment by optimization is significantly faster than hand-tuning. The all-weights-equal approach took less than 10 minutes, but it leads to inferior results.

The observable filtering method requires a single-tune to obtain the $\chi^2$ values per observable which takes 1647 seconds (0.45 hours) for all observables in the A14 dataset, which is followed by applying the Z-score method to filter out outliers (see Section 3.1) and this takes about 10 seconds. Once the single-tune to obtain the $\chi^2$ values per observable is performed, the bin filtering takes an additional 300 seconds for the A14 dataset. Thus, the total pre-processing time required for observable filtering is 1657 seconds (0.46 hours) and for bin-filtering is 1947 seconds (0.54 hours).

Table 12: CPU time (in seconds) and time per iteration (in seconds) taken by all approaches when using all, the observable-filtered, and the bin-filtered A14 data. The robust optimization approach converges after 69, 105, and 83 iterations, respectively. The bilevel-medianscore, -meanscore, and -portfolio approaches are all run for 1000 iterations.

| Method | All data | | Bin filtered | | Observable filtered | |
|---|---|---|---|---|---|---|
| | CPU time | Time per iteration | CPU time | Time per iteration | CPU time | Time per iteration |
| Robust optimization | 3035 | 44 | 2989 | 28 | 3327 | 40 |
| Bilevel-medianscore | 52326 | 52 | 23600 | 24 | 49057 | 49 |
| Bilevel-meanscore | 52169 | 52 | 23600 | 24 | 49018 | 49 |
| Bilevel-portfolio | 52366 | 52 | 23609 | 24 | 49084 | 49 |

From Table 12, we observe that the time taken to tune parameters in the observable-filtered and bin-filtered data case is significantly smaller than for the unfiltered data case. For the bilevel optimization approaches, the time required per iteration for the observable- and bin-filtered cases is 6% and 55% less, respectively, and for each run of robust optimization, it is 9% and 36% less, respectively. Also, the overhead of performing observable and bin filtering is small compared to the time it takes to tune parameters. Since the results from Section 4.6.5 show that the bins filtered by bin and observable filtering do not add significant information to the tune, we can claim that using filtered data provides a significant improvement in compute-time performance for tuning parameters.

# 5 Eigentunes

We use the eigentune approach to calculate confidence intervals for the optimal parameters. We note that the A- and D-optimality criteria provide the size of confidence ellipsoid around the optimal parameters. Here, we expand this information by scanning generator parameters along the principal axes of this ellipsoid. Details of this method are described in [5] and a similar approach is used in estimating the uncertainties of predictions from the parton distribution functions [58]. The interval defines a boundary beyond which the value of the objective function is larger than the objective function value at the minimum by a criterion. The criterion is normally chosen to be the number of degrees of freedom $n$, which is defined as the total number of bins of all observables minus the number of generator parameters, $d$, i.e., $n = \sum_{\mathcal{O} \in \mathcal{S}_{\mathcal{O}}} |\mathcal{O}| - d$. However, to properly take into account the weights assigned to observables, we use the scaled effective sample size as the criteria, which is calculated as follows:

$$n = \gamma \times \left( \frac{(\sum_i w_i)^2}{\sum_i w_i^2} - d \right).$$

The weights are normalized so that the sum of weights associated with all observables equals one. $\gamma$ is iteratively tuned and chosen to be 0.01. The interval would represent the uncertainties of the parameters should the objective function follow a $\chi^2$ distribution. Smaller intervals associated with the tuned parameters indicate that the parameters are better constrained by the experimental data.

Given the non-linearity of the objective function and parameter correlations, a reliable approach to find the 68% confidence interval is to evaluate the objective function for all possible parameter values. However, this poses a computational challenge. Instead, we project the multidimensional parameter space into two directions defined by the eigenvectors $u_{1,2}$ associated with the largest and smallest eigenvalues of the covariance matrix of the parameters,

which are calculated using the inverse of Eq. (16). Then we find an offset $\alpha$ such that the sum of all $\chi^2$ satisfies

$$\chi^2(\mathbf{p}'_{1,2}) = \chi^2(\mathbf{p}^*) + n\,, \tag{17}$$

where $\mathbf{p}'_{1,2} = \mathbf{p}^* \pm u_{1,2} \times \alpha$. For each eigenvector, we obtain two vectors $\mathbf{p}'$ from Eq. (17). Finally, the procedure results in a matrix of sizes of 4 times $d$. Each column represents a generator parameter; the minimum and maximum in each column are used to define the eigentune as shown in Tables 13 and 14 for the A14 and the SHERPA dataset, respectively, using the rational approximation. The same surrogate model is used for all methods. It is possible that the determined intervals go beyond the predefined parameter range. In this case, the MC predictions are extrapolated by the surrogate model. When the lower part of the interval goes negative, we force the value to be zero.

For the A14 data, different optimization methods result in similar intervals for all parameters[5]. The beam remnants (e.g. `BeamRemnants:reconnectRange`) and space-like showering parameters (e.g. `SpaceShower:pT0Ref`) are better constrained; their intervals are within 1% of their optimized parameters. However, the strong coupling parameter in hard scattering processes (`SigmaProcess:alphaSvalue`) and time-like showering (`TimeShower:alphaSvalue`) are less constrained.

For the SHERPA data, different optimization methods produce quite different intervals. Overall, the bilevel-meanscore method results in relatively small intervals for all parameters. The heavy quark fragmentation parameters (e.g. `ALPHA_H`) are well-constrained thanks to the $B$-hadron fragmentation measurements, but the light quark fragmentation parameters are not.

The eigentune results serve as a good platform for comparing our automated optimization algorithms, but it requires manual adjustment of the criteria $n$ and the exploitation of all eigenvectors to produce a realistic uncertainty band. We tried to generate new events with the eigentunes based on the robust optimization outcomes as shown in Table 13 using the PYTHIA8 generator configured closely to the one used in the A14 tune. The uncertainty band was too large to be practically used. To find a reasonable uncertainty band, we performed the eigentune for all ten eigenvectors separately and concluded the strong coupling constant affects most observables. Therefore, we manually adjust the strong coupling values and with an uncertainty of 5% on the strong coupling constant we produced a reasonable uncertainty band. Figure 7 shows two exemplary distributions with the uncertainty band (blue and red lines) included.

Table 13: Eigentune results for the A14 data using the rational approximation for different optimization methods.

| Parameters | Expert | | Bilevel-meanscore | | Bilevel-mediansocre | | Bilevel-portfolio | | Robust optimization | |
|---|---|---|---|---|---|---|---|---|---|---|
| | min | max | min | max | min | max | min | max | min | max |
| `SigmaProcess:alphaSvalue` | 0.075 | 0.193 | 0.079 | 0.192 | 0.079 | 0.190 | 0.074 | 0.195 | 0.085 | 0.183 |
| `BeamRemnants:primordialKThard` | 1.903 | 1.906 | 1.805 | 1.910 | 1.674 | 1.769 | 1.744 | 1.850 | 1.876 | 1.892 |
| `SpaceShower:pT0Ref` | 1.636 | 1.653 | 1.516 | 1.547 | 1.142 | 1.228 | 1.298 | 1.344 | 1.586 | 1.591 |
| `SpaceShower:pTmaxFudge` | 0.905 | 0.912 | 1.012 | 1.016 | 1.069 | 1.096 | 1.037 | 1.046 | 1.025 | 1.026 |
| `SpaceShower:pTdampFudge` | 1.044 | 1.048 | 1.064 | 1.076 | 1.082 | 1.086 | 1.058 | 1.064 | 1.078 | 1.091 |
| `SpaceShower:alphaSvalue` | 0.121 | 0.124 | 0.125 | 0.131 | 0.127 | 0.130 | 0.124 | 0.133 | 0.123 | 0.129 |
| `TimeShower:alphaSvalue` | 0.043 | 0.197 | 0.044 | 0.192 | 0.039 | 0.213 | 0.030 | 0.213 | 0.051 | 0.198 |
| `MultipartonInteractions:pT0Ref` | 1.665 | 2.543 | 1.649 | 2.562 | 1.780 | 1.979 | 1.160 | 2.829 | 1.461 | 2.528 |
| `MultipartonInteractions:alphaSvalue` | 0.068 | 0.177 | 0.072 | 0.161 | 0.115 | 0.121 | 0.062 | 0.186 | 0.094 | 0.151 |
| `BeamRemnants:reconnectRange` | 1.788 | 1.795 | 2.065 | 2.105 | 1.912 | 1.915 | 1.972 | 2.000 | 2.589 | 2.618 |

---

[5]See Table 15 for a description of the physics parameters.

Table 14: Eigentune results for the SHERPA data using the rational approximation for different optimization methods. Parameters with negative values are set to zero.

| Parameters | Bilevel-meanscore | | Bilevel-mediansocre | | Bilevel-portfolio | | Robust optimization | |
|---|---|---|---|---|---|---|---|---|
| | min | max | min | max | min | max | min | max |
| KT_0 | 0.815 | 0.970 | 0.688 | 0.957 | 0.524 | 1.254 | 0.491 | 1.273 |
| ALPHA_G | 0.438 | 0.792 | 1.325 | 1.604 | 0.571 | 0.691 | 1.597 | 2.115 |
| ALPHA_L | 3.683 | 3.824 | 1.309 | 2.863 | 3.525 | 3.939 | 0.291 | 2.088 |
| BETA_L | 0 | 0.460 | 0.043 | 0.062 | 0 | 0.440 | 0 | 0.387 |
| GAMMA_L | 0.175 | 0.362 | 0.330 | 0.352 | 0.688 | 0.823 | 0.220 | 1.087 |
| ALPHA_H | 3.245 | 3.537 | 2.843 | 2.988 | 1.200 | 1.311 | 2.289 | 3.475 |
| BETA_H | 0.747 | 0.898 | 0.484 | 0.585 | 0.623 | 0.972 | 0.350 | 0.759 |
| GAMMA_H | 0.059 | 0.249 | 0 | 0.080 | 0.013 | 0.133 | 0 | 0.469 |
| STRANGE_FRACTION | 0.496 | 0.556 | 0.395 | 0.595 | 0.415 | 0.706 | 0.440 | 0.567 |
| BARYON_FRACTION | 0 | 0.459 | 0.129 | 0.218 | 0.018 | 0.170 | 0 | 0.342 |
| P_QS_by_P_QQ_norm | 0.552 | 0.809 | 0.319 | 0.524 | 0.552 | 0.588 | 0.594 | 0.629 |
| P_SS_by_P_QQ_norm | 0. | 0.031 | 0. | 0.103 | 0 | 0.081 | 0 | 0.068 |
| P_QQ1_by_P_QQ0 | 1.492 | 1.512 | 1.202 | 1.210 | 0.945 | 0.952 | 1.167 | 1.210 |

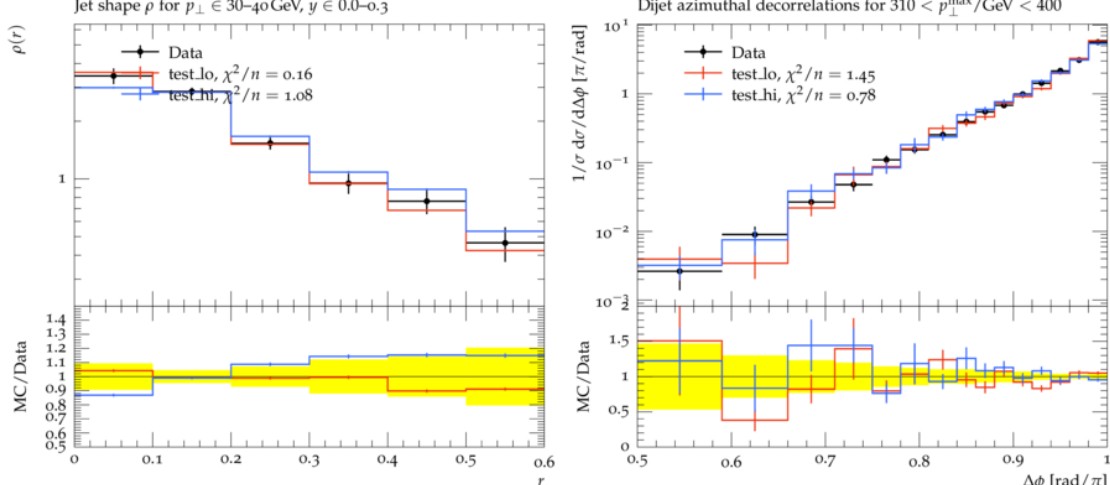

Figure 7: Two exemplary distributions with uncertainty band included. The upper band is in blue and lower band in red. The bottom panel shows the ratio of MC predictions over the data where the yellow band shows the uncertainties associated with the data. Left: jet shape $\rho$ as a function of the distance to the jet axis $r$; Right: the differential cross section of dijet events as a function of the azimuth angle differences between the two jets $\Delta\phi$.

## 6 Discussion

The results presented in the previous sections demonstrate that automated tuning methods can produce better fits of the generator predictions to data. Several figures of merit for comparing different tunes were considered. The automation of the process means that tuning can be performed in less time and with less subjective bias. In this section, we discuss the physics impact of various tuning results.

## 6.1 Implications of our results on physics

Physics event generators are imperfect tools. They contain a mixture of solid physics predictions, approximations, and *ad hoc* models. The approximations and models are expected to be incomplete, and thus are unlikely to describe the full range of observables accessible by the experiment. Despite this fact, for a certain choice of parameters, a model may be able to describe parts of the data. This agreement would be accidental and would likely compromise predictions of this model for different parts of the data. The weighting of data by an expert is a primitive attempt to force the model to agree with data in a region of interest to the physicist – which, most of the time, corresponds to a region where a model should be applied. It is equivalent to adding a large systematic uncertainty to the data that is de-emphasized by the weighting.

Here, we address whether the automated methods accomplish this weighting of data without explicit input from the physicist. First, we should state our expectations for a tune to the A14 dataset. The features of the expert tune were previously discussed in [3, Section. 2.2.1]. The A14 data is all of interest to the physicist, but some of those observables are expected *a priori* to be described better by the event generator than others. The parton shower and hadronization model are expected to describe well *Tracked jet properties* and *Jet shapes*. The description of jets is essential for all hadron collider analyses and is the *raison d'être* for event generators. $t\bar{t}$ *jet shapes* emphasize the final state parton shower, and is critical to be described well when making precision predictions that are sensitive to the top quark mass. *Dijet decorr* and $p_T^Z$ observables provide constraints on initial state parton shower and intrinsic transverse momentum parameters free from most other parameters, and are generically important to be described well. Additional properties, such as the number of jets produced in di-jet or *Z* events or the production of jets at extreme angles, are beyond the scope of the PYTHIA predictions. *Track-jet UE* and *Jet UE* observables are sensitive to PYTHIA's multi-parton-interaction model, which describes most of the particles produced in a high-energy collision. The addition of *Multijets* observables is biasing the parton shower to describe a next-to-leading order observable, while the leading-logarithm parton shower includes only an approximation to the full result. Experience shows that this biasing provides a globally better description of many observables of interest to the physicist with little effort and without significantly impacting other predictions. This feature was built into the *Expert* tune by applying a large weight to this dataset. Finally, adding the $t\bar{t}$ *gap* category is asking for the description of an exclusive observable, which has very strong requirements in its construction, whereas the PYTHIA prediction here is valid for more inclusive observables. Including this data in the tune is a very specific physics requirement that may be beyond the scope of the PYTHIA approximations.

## 6.2 Observables with improved descriptions

Examples of observable predictions with a lower $\chi^2_{\mathcal{O}}$ value than the *expert* tune are displayed in Figures 8a-8c. These reflect an improvement in a class of observables and are indicative of all the comparisons between predictions and data.

All of our methods produce a better description of the data than the expert tune for the category *Jet shapes*, though the expert prediction is mainly differing in only the first bin. This observable is expected to be described well, in general, since it lies in a physics regime compatible with the PYTHIA approximations.

The predictions for the $p_T^Z$ and *Dijet decorr* categories are also improved. We note that the weights found for these analyses are not substantially different than for the expert tune, but that other categories have their weights reduced (see Table 7 for reference). This implies some tension between these observables and the *Multijets* category (to be discussed below).

The comparisons between predictions and data shown in our figures are based on runs

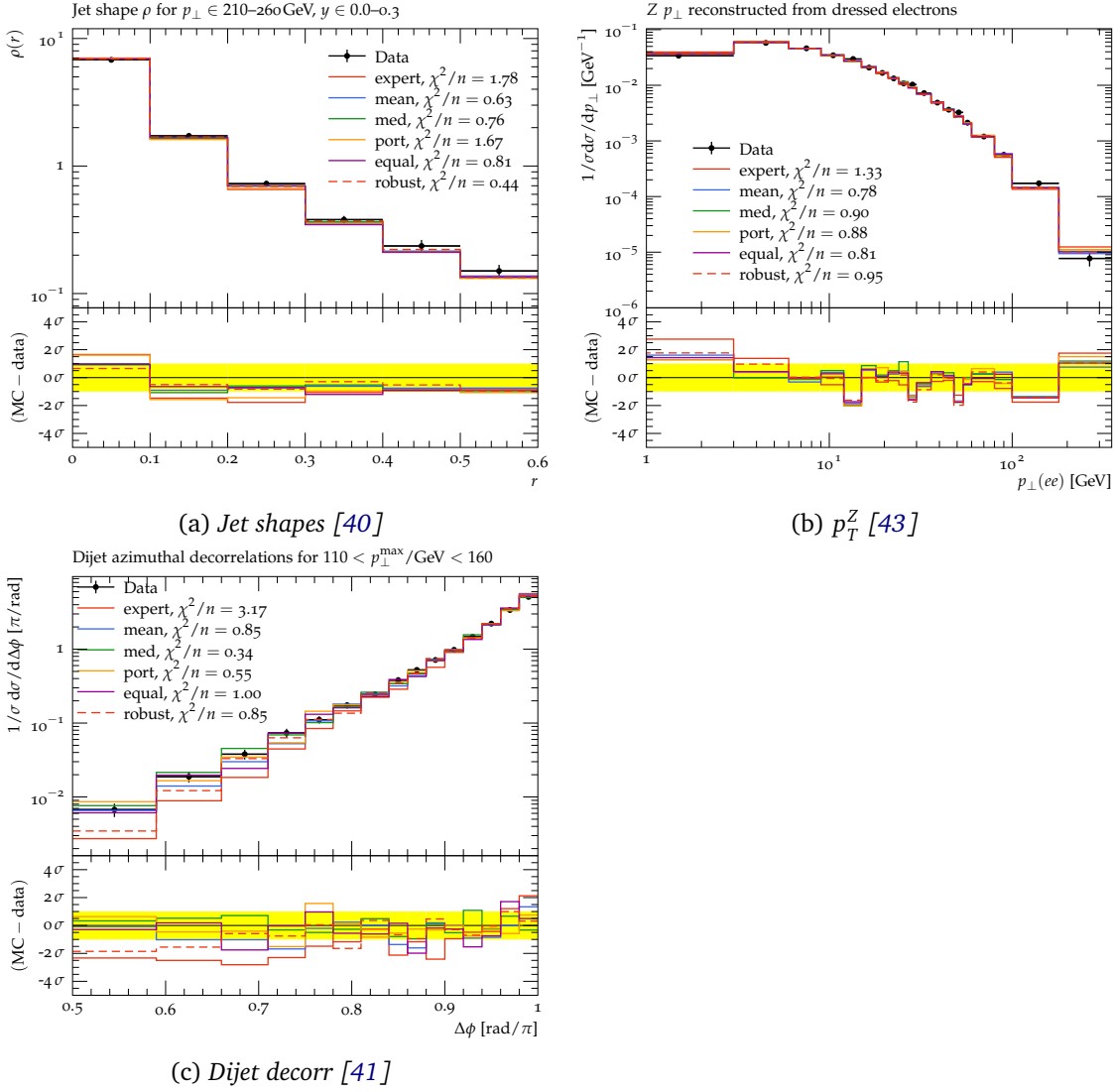

(a) *Jet shapes [40]*

(b) $p_T^Z$ *[43]*

(c) *Dijet decorr [41]*

Figure 8: Examples of A14 observables and their $\chi^2_{\mathcal{O}}$ values for which the automated tuning leads to better fits than the expert's hand tuning.

of the MC event generator for the parameter values derived using the surrogate model. Before continuing, we should comment on the differences in Figure 5 (and in Figure 16 in Section A.9 of the online supplement) between the surrogate model (RA) and explicit runs of the event generator (MC) at the output tuned parameters. The surrogate model would be unreliable if the output tune parameters were outside or near the boundary of the parameter range used to derive the inputs for the surrogate. A comparison of the parameter values relative to the expert tune and Figure 17 shows the distribution of parameter values normalized to the sampling range: $r_{param} = \frac{p - p_{min}}{p_{max} - p_{min}}$. All of the central values for the parameters are well within the sampling range. Only the parameters SpaceShower:pTdampFudge and BeamRemnants:reconnectRange come near the boundaries. For the former, the minimum sampling value was 1.0, and the tuning results only indicate that this parameter should be near 1.0. For the latter, the maximum sampling value was chosen quite large so that all results appear to be close to the minimum value.

Furthermore, the most noticeable differences between the RA surrogate predictions and MC occur for rather small values of the variance between the data and predictions. These values have a negligible impact on the full $\chi^2$, and are within the expected range of validity

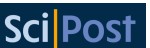

(a) *Multijets [42]*

(b) *$t\bar{t}$ gap [46]*

(c) *Track jet UE [39]*

Figure 9: Examples of A14 observables and their $\chi^2_\mathcal{O}$ values for which the automated tuning approach performs worse than the expert's hand tuning.

of the surrogate model.

## 6.3 Observables with worse descriptions

The predictions for *Track jet properties* and *Substructure* are not significantly improved, but also not degraded. Most of the observables in these categories were designed to tune and test the multi-parton interaction model, and thus it is no surprise that they are described well.

Two categories stand out as being better described by the expert tune. These are the *Multijets* and *$t\bar{t}$ gap* categories that were given a particularly large weight in the expert tune. Some examples can be seen in Figure 9a-9c. It is no surprise that these categories are not described as well as the expert tune. It is surprising that the parameters sensitive to this observable, namely TimeShower:alphaSvalue and SpaceShower:alphaSvalue are actually somewhat larger than the expert tune values, see Table 6. Larger values for these parameters should mean forcing the prediction to look *more* like a higher-order calculation. Clearly, other data, such as *Dijet decorr* and $p_T^Z$ prefer larger values for these parameters than the *Multijets* category alone.



(a) *Jet shapes [54]*

(b) *Jet rates [53]*

(c) *Particle count [56]*

(d) *Several particle counts side-by-side [56]*

Figure 10: Examples of histogram plots of the $\chi^2_{\mathcal{O}}$ values for the SHERPA tune.

Without the expert input, our automated methods do not emphasize these observables. The PYTHIA predictions for *Multijets* and $t\bar{t}$ processes are based on calculations that could be made more accurate (by performing matched or merged calculations based on external input – see [59]), but only at the expense of breaking the universality of the tune. The expert weighting used the flexibility of the PYTHIA model to imitate these more accurate calculations and force agreement with the data. The A14 tune was meant to be applied to physics predictions from the internal PYTHIA model for which the corrections were not readily available or easily applicable. However, if the goal is to provide a tune that can be used even in association with process-dependent corrections, then those provided in this study are more appropriate.

## 6.4 Results for SHERPA tuning

Some of the results of the SHERPA tuning are shown in Figure 10. In general, all of the parameter selection methods applied here yield an improved global $\chi^2_{\mathcal{O}}$ over the default values. The parameters varied in this tuning exercise are all related to the formation of physical particles. This is a phenomenon that occurs at a low-energy scale and cannot be described realistically

(currently, at least) from theory. The model employed in SHERPA is a cluster model that fissions colorless blobs of energy into particles using a parameterized probability distribution. Despite the fact that hadronization occurs at a low-energy scale, it has an impact on observables that are used to test perturbative predictions at relatively high-energy scales. For these observables, it is impossible to entirely disentangle the perturbative prediction from the non-perturbative hadronization model prediction. Figures 10a-10b show comparisons of our tunes to the default, demonstrating a significant improvement in most cases. Figure 10c shows mixed results for the production of one particular species of particle. Figure 10d is an example of an inclusive observable that counts the number of particles produced without any direct reference to their energy or position in the detector.

All of these results are for a certain precision of perturbation theory. There are both technical and mathematical reasons to truncate perturbation theory in a certain order. These calculations were based on the lowest order perturbation theory with an improved parton shower approximation to simulate additional perturbative effects. The lowest order prediction produces 2 jets using exact perturbation theory and any additional jets using the parton shower approximation. Figure 10b is an observable that counts the number of 3-jet events as a function of the jet definition. While our results are improved over the default, this indicates higher-order perturbative calculations might improve the description even more (e.g., 3 jets calculated in exact perturbation theory and 4 or more jets from the parton shower approximation).

Table 10 shows the parameters values for the various tunes. The simplest comparison is between the default values and "All-weights-equal." The all-weights-equal method yields the tune that would result if only the data considered in this study were used. One result is that several of the parameters take on the extremum of the values considered here. Without any additional direction to choose the range for our parameter scan, we chose 1/2 of the default value to define our sampling window.

One surprising result is that the parameter `P_QQ1_by_P_QQ0`, which represents the ratio of spin-1 to spin-0 diquarks, is driven to a value $> 1$. While there is no obvious reason that the cluster model breaks down, spin-1 diquark production is usually expected to be suppressed. The fact that the parameter `BARYON_FRACTION` is driven to its minimal value compensates for this large value.

While the type of large scale parameter tuning we have in mind here can only be performed practically using surrogate models, the fact that some tuned parameters are pushed to the boundaries suggests another direction of algorithmic development. In particular, we would like our algorithm to have the capability to recognize a trust region and update the surrogate model with dedicated simulations when necessary.

## 7  Conclusions

In this paper, we propose several algorithms for automating the weighting the importance of data used in the tuning process for Monte Carlo event generators. We performed two studies. The first used particle collider data and predictions are from the Large Hadron Collider (LHC) and had an *expert* selection of analysis weights as a benchmark. The second used data and predictions are from the Large Electron-Positron (LEP) Collider and had only the default parameter choices as a reference. The algorithms considered included a bilevel optimization based on several scoring procedures and a single-level robust optimization. We find that our automatic methods produce parameter tunes that are comparable to labor-intensive, by-hand tunes. For the LHC tuning, filtering of hard-to-describe observables can lead to tunes of superior quality by identifying observables or subsets of observables that cannot be described by the event generator. For the LEP tuning, many of the tuned parameters were driven to

the extremum of our sampling range, suggesting that the current models are missing some important physics. We note here that filtering approaches only eliminate parts of the model that are highly unlikely to be explained by data. Hence, it is a conservative approach since the range of the function within the domain is usually much larger than the range of the values that could be used to fit the data. The filtering is based on the intuition that the models that are highly unlikely to be explained by data could be removed to (a) get a better estimate of the tune, and (b) prevent the algorithms from going into regions of extrapolation.

First, the results show that the parameter values we found agree with and have the potential to improve the physicists' hand-tuned results. Second, since we automate the weight adjustment for the tune-relevant observables, physicists do not need to hand-tune the weights for observables anymore; we propose several methods for adjusting the weights, so physicists are not involved in the subjective re-weighting anymore. Third, by filtering out and excluding observables and bins, we can save computational time during optimization and improve the parameter values. Fourth, we derived new metrics to easily compare different tunes, and it shows that our methods can perform better than the physicists' hand-tuning approach.

To get the baseline recommendation among the proposed methods, we suggest that the physicist first select a metric to be minimized. Then, from Tables 5 and 9, we see that if the goal is to minimize the weighted $\chi^2$ metric, the robust optimization approach should be chosen. On the other hand, if the goal is to minimize the uncertainty of the estimate, we recommend performing the observable- or bin-filtering first and then using the bilevel-portfolio method.

For the SHERPA data, most of the optimal parameters are on the boundaries of the parameter space, indicating that we might need to change the size of the parameter domain to avoid model extrapolation. One possible solution to this problem is to build an outer loop for moving the center of the parameter search space and apply the trust region method. We leave this to future research.

In this work, we assumed that each bin is completely independent of all the other bins. To consider correlations, we need to solve $\widehat{\mathbf{p}}_{\mathbf{w}} \in \arg\min_{\mathbf{p} \in \Omega} ||\mathcal{F}(\mathbf{p}) - \mathcal{D}||^2_{\Gamma^{-1/2}(\mathbf{p})\mathbf{W}\Gamma^{-1/2}(\mathbf{p})}$, where $\mathcal{F}(\mathbf{p})$ is an aggregate vector of central values of the model prediction obtained using a polynomial or rational approximation, $\mathcal{D}$ is the aggregated vector of data, $\mathbf{W}$ is the weight vector, and $\Gamma(\mathbf{p})$ is the covariance matrix. As we see, the inclusion of the covariance matrix only affects the inner optimization and the methods proposed here for automatic weight adjustment would be unchanged. However, including the covariance matrix has its challenges. Specifically, (a) the information of the correlations among the bins is currently unavailable, (b) since the covariance matrix depends on the parameter values, we would need to approximate it using a kernel function, and (c) solving this optimization problem is non-trivial since it would require the inversion and taking the square root of the covariance kernel for each objective function evaluation. Tackling these issues is outside the scope of this paper and hence, taking into account bin correlations is left as future work.

In this work, we do not address the issue of the gap that may exist between the model and the MC event generator. However, this gap only affects the inner optimization. As a result, the parameter tune obtained from minimizing the weighted $\chi^2$ objective in the inner optimization problem may not yield the same $\chi^2$ value when used in the MC event generator. Another issue is how to select the bounds of the parameter domain $\Omega$. To overcome these issues, we need an approach that queries the MC event generator directly in the inner optimization. This can be achieved by using a derivative-free optimization approach. However, this task is non-trivial since doing this efficiently would require using the correct fidelity of the MC, the number of parameters at which to run the MC, and also deal with other issues that would affect the convergence of such an algorithm. Hence, we leave this work as future research topic.

# Acknowledgements

SM thanks Stefan Hoeche for discussions about our SHERPA results.

**Funding information**  This work was supported by the U.S. Department of Energy, Office of Science, Advanced Scientific Computing Research, under Contract DE-AC02-06CH11357. Support for this work was provided through the Scientific Discovery through Advanced Computing (SciDAC) program funded by U.S. Department of Energy, Office of Science, Advanced Scientific Computing Research. This work was also supported by the U.S. Department of Energy through grant DE-FG02-05ER25694, and by Fermi Research Alliance, LLC under Contract No. DE-AC02-07CH11359 with the U.S. Department of Energy, Office of Science, Office of High Energy Physics. This work was supported in part by the U.S. Department of Energy, Office of Science, Office of Advanced Scientific Computing Research and Office of Nuclear Physics, SciDAC program through the FASTMath Institute under Contract No. DE-AC02-05CH11231 at Lawrence Berkeley National Laboratory.

# A  Online Supplement

Online supplement for "BROOD: Bilevel and Robust Optimization and Outlier Detection for Efficient Tuning of High-Energy Physics Event Generators".

## A.1  Solving the outer problem with derivative-free surrogate optimization

Solving the inner optimization problem can become computationally demanding as it depends on the number of observables involved, the number of bins per observable (and therefore the number of parameters), and the starting guess (and therefore the number of iterations needed). Thus, the goal is to determine the optimal weights $\mathbf{w}^*$ within as few iterations of the outer loop as possible since this number determines how often we have to solve the inner optimization problem. We do not have a full analytic expression of $g(\mathbf{w}, \widehat{\mathbf{p}}_{\mathbf{w}})$ (black box) since computing this value involves solving the inner optimization problem. Thus, also derivatives of $g(\mathbf{w}, \widehat{\mathbf{p}}_{\mathbf{w}})$ are not available. A widely used approach for optimizing computationally expensive black-box functions is to use computationally cheap approximations (surrogates, metamodels) of the expensive function and to use the approximation throughout the optimization to make iterative sampling decisions [60]. Here, we approximate $g(\mathbf{w}, \widehat{\mathbf{p}}_{\mathbf{w}})$ with a radial basis function (RBF) [61], although in general any approximation model could be used. An RBF interpolant is defined as follows:

$$s(\mathbf{w}) = \sum_{i=1}^{n} \gamma_i \phi(\|\mathbf{w} - \mathbf{w}_i\|_2) + q(\mathbf{w}), \tag{18}$$

where $s : \mathbb{R}^{|\mathcal{S}_{\mathcal{O}}|} \mapsto \mathbb{R}$, $\mathbf{w}_i$, $i = 1, \ldots, n$, are the weight vectors for which we have already evaluated the objective function of the outer optimization problem, $\gamma_i$ are parameters that must be determined, $\phi(\cdot)$ is the radial basis function (here, we choose the cubic, $\phi(r) = r^3$, but other options are possible), $\|\cdot\|_2$ denotes the Euclidean norm, and $q(\cdot)$ is a polynomial tail whose order depends on the choice of $\phi$. When using the cubic RBF, the polynomial tail must be at least linear ($q(\mathbf{w}) = \beta_0 + \boldsymbol{\beta}^\top \mathbf{w}$) in order to uniquely determine the RBF parameters ($\gamma_i, i = 1, \ldots, n, \beta_0, \boldsymbol{\beta} = [\beta_1, \ldots, \beta_{|\mathcal{S}_{\mathcal{O}}|}]^\top$). The RBF interpolant $s(\mathbf{w})$ then predicts the value of the objective function at the point $\mathbf{w}$. It is interpolating, and thus the prediction at an already evaluated point $\mathbf{w}_i$ will agree with the observed function value. Using the RBF, we thus have $g(\mathbf{w}, \widehat{\mathbf{p}}_{\mathbf{w}}) = s(\mathbf{w}) + e(\mathbf{w})$, where $e(\mathbf{w})$ denotes the difference between the RBF and the true

function value and it is 0 at already evaluated vectors $\mathbf{w}_i$. The values of the RBF parameters are determined by solving a linear system of equations:

$$\begin{bmatrix} \boldsymbol{\Phi} & \mathbf{W} \\ \mathbf{W}^\top & \mathbf{0} \end{bmatrix} \begin{bmatrix} \boldsymbol{\gamma} \\ \boldsymbol{\beta}' \end{bmatrix} = \begin{bmatrix} \mathbf{G} \\ \mathbf{0} \end{bmatrix}, \tag{19}$$

where the elements of $\boldsymbol{\Phi}$ are $\Phi_{\iota\nu} = \phi(\|\mathbf{w}_\iota - \mathbf{w}_\nu\|_2)$, $\iota, \nu = 1\dots n$, $\mathbf{0}$ is a matrix with all entries 0 of appropriate dimension, and

$$\mathbf{W} = \begin{bmatrix} \mathbf{w}_1^\top & 1 \\ \vdots & \vdots \\ \mathbf{w}_n^\top & 1 \end{bmatrix} \quad \boldsymbol{\gamma} = \begin{bmatrix} \gamma_1 \\ \gamma_2 \\ \vdots \\ \gamma_n \end{bmatrix} \quad \boldsymbol{\beta}' = \begin{bmatrix} \beta_1 \\ \beta_2 \\ \vdots \\ \beta_{|\mathcal{S}_\mathcal{O}|} \\ \beta_0 \end{bmatrix} \quad \mathbf{G} = \begin{bmatrix} g(\mathbf{w}_1, \widehat{\mathbf{p}}_{\mathbf{w}_1}) \\ g(\mathbf{w}_2, \widehat{\mathbf{p}}_{\mathbf{w}_2}) \\ \vdots \\ g(\mathbf{w}_n, \widehat{\mathbf{p}}_{\mathbf{w}_n}) \end{bmatrix}. \tag{20}$$

The linear system in Eq. (19) has a solution if and only if $\mathrm{rank}(\mathbf{W}) = |\mathcal{S}_\mathcal{O}| + 1$ [15]. During the optimization, we use the RBF prediction at unsampled points to determine a new vector $\mathbf{w}$ for which we solve the inner optimization problem. It is important that at this step only weights that sum up to 1 are chosen. The steps of the iterative sampling algorithm are summarized in Algorithm A.1.

---

**Algorithm A.1:** Derivative-free optimization of the outer equality-constrained optimization problem

---

**Input:** Number of initial experimental design points $n_0$; the maximum number of evaluations $n_{\max}$

**Output:** The best weight vector $\mathbf{w}^*$ and corresponding $\widehat{\mathbf{p}}^*_{\mathbf{w}^*}$

1: Create an initial experimental design with $n_0$ points; ensure that Eq. (3b) is satisfied for all points;

2: Compute the value of the outer optimization objective function at all points in the initial design;

3: Fit an RBF model to the sample data pairs $\{(\mathbf{w}_i, g(\mathbf{w}_i, \widehat{\mathbf{p}}_{\mathbf{w}_i}))\}_{i=1}^{n_0}$

4: Set $n = n_0$

5: **while** $n < n_{\max}$ **do**

6:     Use the RBF to determine a new point $\mathbf{w}_{\text{new}}$ and ensure that Eq. (3b) is satisfied;

7:     Solve the inner optimization problem for $\mathbf{w}_{\text{new}}$ and obtain $\widehat{\mathbf{p}}_{\mathbf{w}_{\text{new}}}$;

8:     Compute the value of the outer optimization objective function for $(\mathbf{w}_{\text{new}}, \widehat{\mathbf{p}}_{\mathbf{w}_{\text{new}}})$;

9:     Update the RBF model with the new data;

10:     $n \leftarrow n + 1$;

11: **end while**

12: **return** the best parameter values $(\mathbf{w}^*, \widehat{\mathbf{p}}^*_{\mathbf{w}^*})$;

---

The inputs that must be supplied to the algorithm are the number of points $n_0$ to be used in the initial experimental design and the maximum number $n_{\max}$ of outer objective function evaluations (i.e., the number of inner optimizations) one is willing to allow. The number $n_0$ should in our case be at least $|\mathcal{S}_\mathcal{O}| + 1$, since this is the minimum number of points we need to fit the RBF model. $n_{\max}$ should depend on how long the inner optimization takes and the time budget of the user.

When creating the initial experimental design in Step 1, we have to ensure that the constraint (3b) is satisfied. Also, we have the condition that the weights lie in $[0, 1]$ and are uniform in their support. This means that the weights follow the Dirichlet distribution, i.e., the set of points are uniformly distributed over the open standard $(|\mathcal{S}_\mathcal{O}| - 1)$-simplex. To achieve

this, we generate an initial design where all weights are drawn from the symmetric Dirichlet distribution, $Dir(\alpha_1 = \alpha_2 = \ldots = \alpha_{|\mathcal{S}_{\mathcal{O}}|} = 1)$ [62–64].

We evaluate the outer objective function at these points, i.e., we solve the inner optimization problem at each point and we obtain $\mathbf{G}$ in Eq. (20). With the sum-one-scaled initial experimental design, however, the rank of the matrix $W$ is now only $|\mathcal{S}_{\mathcal{O}}|$ (and not the required $|\mathcal{S}_{\mathcal{O}}| + 1$). Thus, we solve the problem as one of dimension $|\mathcal{S}_{\mathcal{O}}| - 1$, i.e., for fitting the RBF model, we only use the first $|\mathcal{S}_{\mathcal{O}}| - 1$ values of each sample point (the "reduced" sample points). Thus, we use

$$
W = \begin{bmatrix}
w_{1,1} & w_{1,2} & \ldots & w_{1,|\mathcal{S}_{\mathcal{O}}|-1} & 1 \\
w_{2,1} & w_{2,2} & \ldots & w_{2,|\mathcal{S}_{\mathcal{O}}|-1} & 1 \\
\vdots & \vdots & \vdots & \vdots & \vdots \\
w_{n,1} & w_{n,2} & \ldots & w_{n,|\mathcal{S}_{\mathcal{O}}|-1} & 1
\end{bmatrix}
\tag{21}
$$

and the coefficient vector for the polynomial tail thus becomes $[\beta_1, \ldots, \beta_{|\mathcal{S}_{\mathcal{O}}|-1}, \beta_0]^\top$. The vector $\gamma$ and the matrix $G$ do not change. The elements of $\Phi$ are computed from the $(|\mathcal{S}_{\mathcal{O}}|-1)$-dimensional sample vectors. Note, however, that when we evaluate the objective function in Eq. (3a), we always evaluate it for the full-dimensional vectors, as we can simply compute $w_{j,|\mathcal{S}_{\mathcal{O}}|} = 1 - \sum_{i=1}^{|\mathcal{S}_{\mathcal{O}}|-1} w_i$ for each $j = 1, \ldots, n$.

In the iterative sampling procedure (Steps 5-11), we use the RBF model to determine a new vector $\mathbf{w}_{\text{new}}$ at which we will do the next evaluation of Eq. (3a). Since we do not know whether the objective function is multimodal, we have to balance local and global search steps, i.e., we have to balance our sample point selection such that we select points with low predicted function values but also points that are far away from already evaluated points. Moreover, the new sample point must satisfy Eq. (3b). In order to do so, we generate a large set of candidate points from the Dirichlet distribution. We use the RBF to predict the function values at the candidate points. Since the RBF is defined over the $(|\mathcal{S}_{\mathcal{O}}|-1)$-dimensional space, we use only the first $|\mathcal{S}_{\mathcal{O}}|-1$ parameter values of each candidate point. We denote the $(|\mathcal{S}_{\mathcal{O}}|-1)$-dimensional candidate points by $\mathbf{x}_1, \ldots, \mathbf{x}_{N_{\text{cand}}}$, where we choose $N_{\text{cand}}$ large (for example, $500|\mathcal{S}_{\mathcal{O}}|$). For each candidate point, we use the RBF to predict its function value using (18) and we obtain $s(\mathbf{x}_k), k = 1, \ldots, N_{\text{cand}}$. We scale these values to $[0,1]$ according to

$$
V_s(\mathbf{x}_k) = \frac{s(\mathbf{x}_k) - s_{\min}}{s_{\max} - s_{\min}}, k = 1, \ldots, N_{\text{cand}},
\tag{22}
$$

where

$$
s_{\min} = \min\{s(\mathbf{x}_k), k = 1, \ldots, N_{\text{cand}}\} \text{ and } s_{\max} = \max\{s(\mathbf{x}_k), k = 1, \ldots, N_{\text{cand}}\}.
\tag{23}
$$

We also compute the distances $d(\mathbf{x}_k, S)$ of each candidate point to the set of already evaluated points $S$ (in the $(|\mathcal{S}_{\mathcal{O}}|-1)$-dimensional Euclidean space), and we scale these distances to $[0,1]$ according to

$$
V_d(\mathbf{x}_k) = \frac{d_{\max} - d(\mathbf{x}_k)}{d_{\max} - d_{\min}}, k = 1, \ldots, N_{\text{cand}},
\tag{24}
$$

where

$$
d_{\min} = \min\{d(\mathbf{x}_k, S), k = 1, \ldots, N_{\text{cand}}\} \text{ and } d_{\max} = \max\{d(\mathbf{x}_k, S), k = 1, \ldots, N_{\text{cand}}\}.
\tag{25}
$$

The ideal new sample point $\mathbf{w}_{\text{new}}$ will have a large distance to the set of already evaluated points $S$ and a low predicted objective function value. Using the two criteria defined above, we compute a weighted sum of both (following [65])

$$
V(\mathbf{x}_k) = \nu V_s(\mathbf{x}_k) + (1 - \nu)V_d(\mathbf{x}_k), k = 1, \ldots, N_{\text{cand}},
\tag{26}
$$

where $\nu \in [0, 1]$ is a parameter that determines how much emphasis we put on either criterion. If $\nu$ is large, it means we put most emphasis on $V_s$, and we favor candidate points that have low predicted objective function values. This also means that the search is more local as low function values are usually predicted around the best point found so far. If $\nu$ is small, we put more emphasis on $V_d$ and we favor points that are far away from the set of already evaluated points, and thus the search is more global. By varying the weights $\nu$ between different values in $[0,1]$, we can achieve a repeated transition between local and global search, and therefore we can avoid becoming stuck in a local optimum. The candidate point with the lowest $V$ value will become the new sample point $\mathbf{w}_{\text{new}}$. We evaluate the objective function (inner optimization) at the new point (augmented with the missing parameter value), and given the new data, we update the RBF model. The algorithm iterates until the maximum number of function evaluations $n_{\text{max}}$ has been reached.

## A.2 Polynomial-time algorithm for filtering bins by hypothesis testing

In this section, we describe the polynomial-time algorithm to solve the problem of finding the largest contiguous subset of bins $\mathcal{B} \subset \mathcal{O}$ to be kept for tuning, i.e., finding the largest contiguous subset of bins $\mathcal{B} \subset \mathcal{O}$ such that $\chi_{\mathcal{B}}^2 \leq \chi_{c,\mathcal{B}}^2$, where $\chi_{c,\mathcal{B}}^2$ is the critical value for bins in $\mathcal{B}$. This algorithm is described in Algorithm A.2 and it is based on the maximum subarray problem [31].

In this algorithm, we first find the critical value for each bin in line 1 as described in Section 3.2. The degree of the freedom is given by $\rho_{\mathcal{B}} = |\mathcal{B}| - d$ and since $\rho_{\mathcal{B}}$ cannot be negative, the critical values for only the bin index $b > d$ is calculated in line 1. Then the $\chi^2$ test statistic is computed for each bin in $\mathcal{O}$ in lines 2-3. Then, while iterating through the bins in $\mathcal{O}$, in lines 6, we check whether the current bin $b$ can be added to $\mathcal{B}$ and if so, we update the counters and add the current bin $b$ to the end of $\mathcal{B}$ (via $e$) in lines 7-10. If the current bin $b$ cannot be added to $\mathcal{B}$, then in lines 12-13 we shift the start $s$ of $\mathcal{B}$ (through $\tau$) such that the start is now at the bin index where the condition in line 6 could be satisfied in future iterations. Finally, in lines 14-19, we perform a sanity check to make sure that $\mathcal{B}$ contains the set of bins that yield the lowest $\chi_{\mathcal{B}}^2$ test statistic.

## A.3 A14 and SHERPA physics parameters

The A14 tunable physics parameters, their definitions and tuning ranges are shown in Table 15. The SHERPA parameters, their definitions and tuning ranges are shown in Table 16.

## A.4 Selection of the best hyperparameter in robust optimization

In order to find the best value for $\mu$ in the robust optimization, we first build for each run (each $\mu$) a cumulative density curve of the number of observables for which $\frac{\chi_{\mathcal{O}}^2(\mathbf{p}^*, \mathbf{w})}{|\mathcal{O}|} \leq \tau$, where $\mathbf{p}^*$ is the optimal parameter obtained from the robust optimization run, $\mathbf{w} = \mathbf{1}$, $\tau \in \mathbb{R}^+$ and $\mathcal{O} \in \mathcal{S}_{\mathcal{O}}$. Then, we construct the "ideal" cumulative density curve, for which $\mathbf{p}^*$ in $\frac{\chi_{\mathcal{O}}^2(\mathbf{p}^*, \mathbf{w})}{|\mathcal{O}|} \leq \tau$ is obtained by optimizing for each observable $\mathcal{O}$ separately. An example plot showing the cumulative density curve from the ideal case to some of the robust optimization runs is shown in Figure 11.

Then, the area between the cumulative density curve for each robust optimization run and the ideal cumulative density curve is computed. For the A14 dataset and all runs completed for robust optimization, the area between the curve is given in Table 17 (smaller values are better). Finally, for completeness, the best values of $\mu$ found for both the A14 and SHERPA datasets are given in Table 18.

---

**Algorithm A.2:** Algorithm to find bins $\mathcal{B}$ in observable $\mathcal{O}$ to keep for tuning

    **Input**   : $f_b, \mathcal{R}_b, \Delta f_b, \Delta \mathcal{R}_b, \forall b \in \mathcal{O}$; significance level $\alpha$

    **Output:** start index $s$ and end index $e$ of bins, i.e., $\mathcal{B} = \{s, \ldots, e\}$ to keep in $\mathcal{O}$

**1** Calculate the critical values for each bin:

$$k_b = \begin{cases} \chi^2_{c,b} = f(\rho_b, \alpha), & \text{if } b > d \\ \infty, & \text{otherwise} \end{cases}, \quad \forall b \in \{1, 2, \ldots, |\mathcal{O}|\}, \mathbf{p} \in \Omega \subset \mathbb{R}^d$$

**2** Find $\mathbf{p}^*$ by minimizing $\chi^2_{\mathcal{O}}$ in Eq. (12)

**3** Calculate the test statistic values for each bin:

    $\chi^2_b(\mathbf{p}^*) = \frac{(f_b(\mathbf{p}^*) - \mathcal{R}_b)^2}{\Delta f_b(\mathbf{p}^*)^2 + \Delta \mathcal{R}_b^2}, \quad \forall b \in \{1, 2, \ldots, |\mathcal{O}|\}$

**4** Initialize $\Sigma \leftarrow 0, \widehat{b} \leftarrow 0, s \leftarrow 0, e \leftarrow 0, \tau \leftarrow 0$

**5** **for** $b \in \{1, 2, \ldots, |\mathcal{O}|\}$ **do**

**6**     **if** $\Sigma + \chi^2_b \leq k_{\widehat{b}+1}$ **then**

**7**         $\Sigma \leftarrow \Sigma + \chi^2_b$

**8**         $\widehat{b} \leftarrow \widehat{b} + 1$

**9**         $s \leftarrow \tau$

**10**        $e \leftarrow b$

**11**     **else if** $\Sigma \neq 0$ **then**

**12**         $\Sigma \leftarrow \Sigma - \chi^2_{b-\widehat{b}} + \chi^2_b$

**13**         $\tau \leftarrow b - \widehat{b} + 1$

**14** **if** $s > 0$ *and* $\chi^2_{s-1} < \chi^2_e$ **then**

**15**     $e \leftarrow e - 1$

**16**     $s \leftarrow s - 1$

**17** **else if** $e < |\mathcal{O}|$ *and* $\chi^2_s > \chi^2_{e+1}$ **then**

**18**     $e \leftarrow e + 1$

**19**     $s \leftarrow s + 1$

**20** **return** $\mathcal{B} = \{s, \ldots, e\}$.

---

Table 15: PYTHIA physics parameters used in the A14 tune, their definitions, and tuning ranges (min, max). More details of the parameters can be found in the online PYTHIA manual: pythia.org/latest-manual/Welcome.html

| Parameter | Description | min | max |
|---|---|---|---|
| SigmaProcess:alphaSvalue | Strong coupling parameter $\alpha_S$, at the scale $Q^2 = M_Z^2$, used to calculate QCD cross sections | 0.12 | 0.15 |
| BeamRemnants:primordialKThard | Hard process scale dependence of the primordial $k_\perp$ added to hard scattering subsystems. | 1.5 | 2.0 |
| SpaceShower:pT0Ref | Regulator of the $p_T \to 0$ divergence of the initial state (ISR) parton shower kernels | 0.75 | 2.5 |
| SpaceShower:pTmaxFudge | Factor to modify the starting ISR evolution scale | 0.5 | 1.5 |
| SpaceShower:pTdampFudge | Factor to dampen the ISR evolution scale | 1.0 | 1.5 |
| SpaceShower:alphaSvalue | Similar to SigmaProcess:alphaSvalue, but for ISR | 0.10 | 0.15 |
| TimeShower:alphaSvalue | Similar to SigmaProcess:alphaSvalue, but for final state (FSR) parton showers | 0.10 | 0.15 |
| MultipartonInteractions:pT0Ref | Similar to SpaceShower:pT0Ref, but used in the multiparton interaction (MPI) model | 1.5 | 3.0 |
| MultipartonInteractions:alphaSvalue | Similar to SigmaProcess:alphaSvalue, but for MPI | 0.10 | 0.15 |
| BeamRemnants:reconnectRange | Sets probability for color reconnections between lower and higher $p_T$ systems | 1.0 | 10.0 |

Table 16: SHERPA physics parameters, their definitions and tuning ranges (min, max).

| Parameters | Definition | min | max |
|---|---|---|---|
| KT_0 | generic parameter for non-perturbative transverse momentum | 0.5 | 1.5 |
| ALPHA_G | gluon fragmentation | 0.62 | 1.88 |
| ALPHA_L | light quark fragmentation $z$ power | 1.25 | 3.75 |
| BETA_L | light quark fragmentation $1-z$ power | 0.05 | 0.15 |
| GAMMA_L | light quark fragmentation exp power | 0.25 | 0.75 |
| ALPHA_H | heavy quark fragmentation $z$ power | 1.25 | 3.75 |
| BETA_H | heavy quark fragmentation $1-z$ power | 0.375 | 1.125 |
| GAMMA_H | heavy quark fragmentation exp power | 0.05 | 0.15 |
| STRANGE_FRACTION | suppression of $s$ quarks | 0.25 | 0.75 |
| BARYON_FRACTION | suppression of baryons | 0.09 | 0.27 |
| P_QS_by_P_QQ_norm | fraction of di-quarks with one strange quark | 0.24 | 0.72 |
| P_SS_by_P_QQ_norm | fraction of di-quarks with two strange quarks | 0.01 | 0.03 |
| P_QQ1_by_P_QQ0 | fraction of di-quarks with spin-1 to spin-0 | 0.5 | 1.5 |

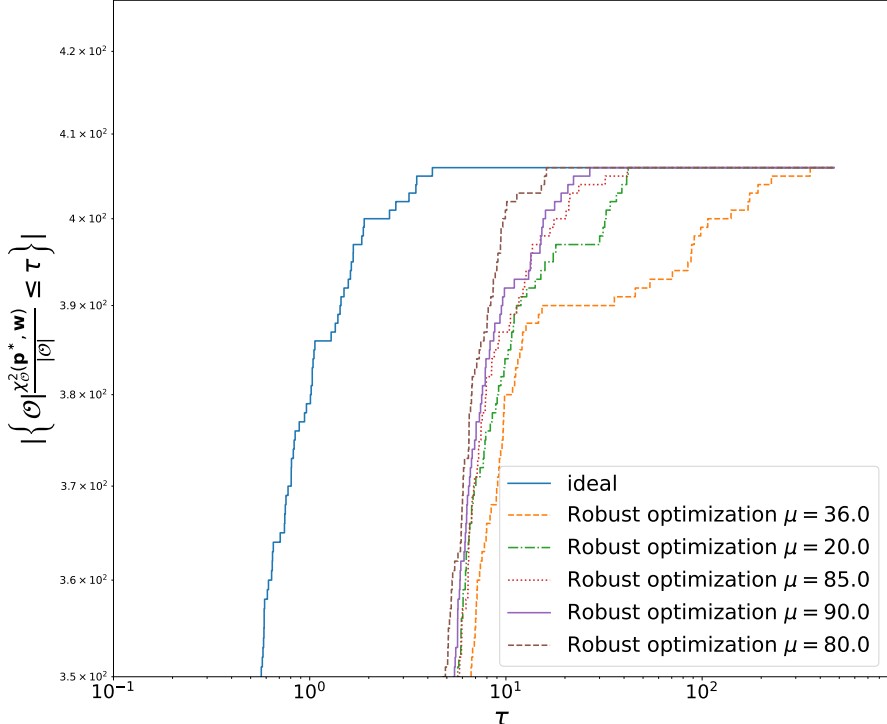

Figure 11: Ideal cumulative density curve and the cumulative density curves of robust optimization runs with selected hyperparameter values $\mu$ for the A14 dataset.

## A.5 Outlier observables in the A14 dataset

There are 12 outlier observables using the cubic polynomial approximation and 9 outlier observables using the rational approximation in the A14 dataset.

| Cubic Polynomial Model | Rational Approximation Model |
|---|---|
| /ATLAS_2011_I919017/d01-x02-y02 | /ATLAS_2011_I919017/d01-x02-y02 |
| /ATLAS_2011_I919017/d01-x02-y03 | /ATLAS_2011_I919017/d01-x04-y04 |
| /ATLAS_2011_I919017/d01-x03-y02 | /ATLAS_2011_I919017/d02-x04-y03 |
| /ATLAS_2011_I919017/d01-x03-y07 | /ATLAS_2011_I919017/d02-x04-y04 |
| /ATLAS_2011_I919017/d01-x04-y07 | /ATLAS_2011_I919017/d02-x04-y05 |
| /ATLAS_2011_I919017/d01-x04-y08 | /ATLAS_2011_I919017/d02-x04-y09 |

```
/ATLAS_2011_I919017/d01-x04-y09   /ATLAS_2011_I919017/d02-x04-y10
/ATLAS_2011_I919017/d02-x04-y04   /ATLAS_2011_I919017/d02-x04-y14
/ATLAS_2011_I919017/d02-x04-y10   /ATLAS_2011_I919017/d02-x04-y15
/ATLAS_2011_I919017/d02-x04-y13
/ATLAS_2011_I919017/d02-x04-y14
/ATLAS_2011_I919017/d02-x04-y15
```

### A.6   Outlier observables in the SHERPA dataset

There are 2 outlier observables using the cubic polynomial approximation and 3 outlier observables using the rational approximation in the SHERPA dataset.

| Cubic Polynomial Model | Rational Approximation Model |
|---|---|
| /DELPHI_1996_S3430090/d07-x01-y01 | /DELPHI_1996_S3430090/d02-x01-y01 |
| /DELPHI_1996_S3430090/d08-x01-y01 | /DELPHI_1996_S3430090/d07-x01-y01 |
|  | /DELPHI_1996_S3430090/d08-x01-y01 |

### A.7   Bin filtered data for A14 dataset

In Table 19, we give the names of the A14 observables from which bins have been filtered, the number of bins filtered out, critical $\chi^2$ value, and $\chi^2$ test statistic before and after filtering the bins.

### A.8   Bin filtered data for SHERPA dataset

In Table 20, we give the names of the SHERPA observables from which bins have been filtered, the number of bins filtered out, critical $\chi^2$ value, and $\chi^2$ test statistic before and after filtering the bins.

### A.9   Complete results from filtering out observables and bins

In Figures 12 and 13, the cumulative distribution plots for parameters obtained after bin filtering and observable filtering for the A14 data are presented. In Figures 14 and 15, the cumulative distribution plots for parameters obtained after bin filtering and observable filtering for the SHERPA data are presented. From these figures, we observe that there is no significant difference in the number of bins within the 1 $\sigma$ variance level between the optimal parameters $\mathbf{p}_a^*$ obtained when all bins were used for tuning and the optimal parameters $\mathbf{p}_b^*$ and $\mathbf{p}_o^*$ obtained when only the bin filtered and observable filtered bins are used for tuning, respectively. There is some disagreement in the cumulative distribution of bins when the variance level is less than $10^{-1}$. But this is not significant since the number of these bins is small and all of them have small levels of variance. Additionally, to get $\mathbf{p}_a^*$, the filtered bins were used. So we see that for variance levels less than $10^{-1}$, $\mathbf{p}_a^*$ performs better on the filtered data (solid blue line) than $\mathbf{p}_b^*$ or $\mathbf{p}_o^*$ (dashed blue line). However for variance levels beyond $10^{-1}$, this difference is negligible. This means that filtering the bins or observables does not deteriorate the variance of the bins for levels greater than $10^{-1}$. To summarize, we conclude that the MC generator cannot explain very well the data of the bins that were removed by filtering. Hence, removing these bins from the tuning process does not reduce the information required to achieve a good tune as the performance for bins with moderate and high variance in the filtered case is very similar to the case when all the bins are included.

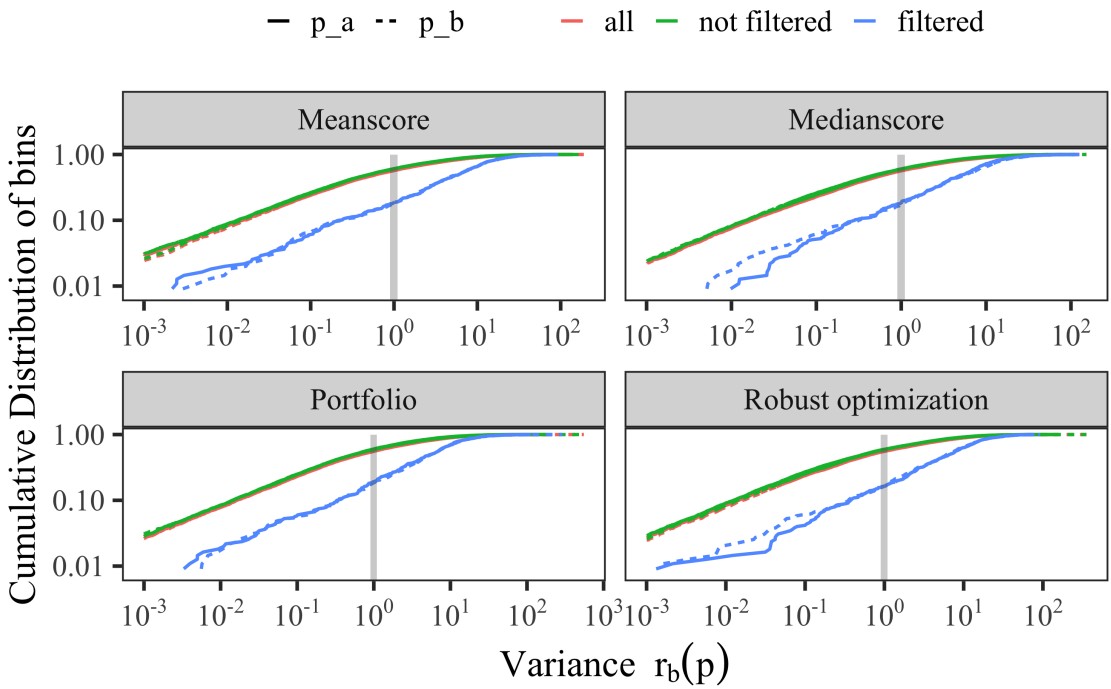

Figure 12: Cumulative distribution of bins for the A14 dataset at different bands of variance levels using different approaches. Results are shown using the parameters $\mathbf{p}_a^*$ obtained using all bins during optimization, and the parameters $\mathbf{p}_b^*$ obtained when only the bin filtered bins are used during optimization.

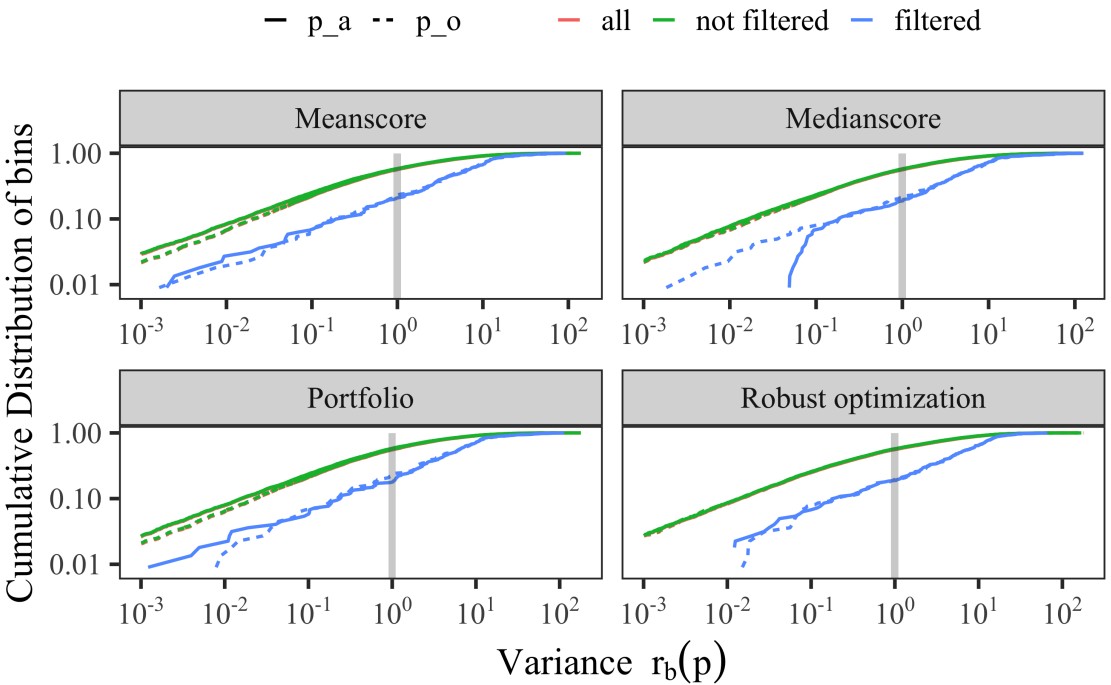

Figure 13: Same as Figure 12, but using observable filtering.

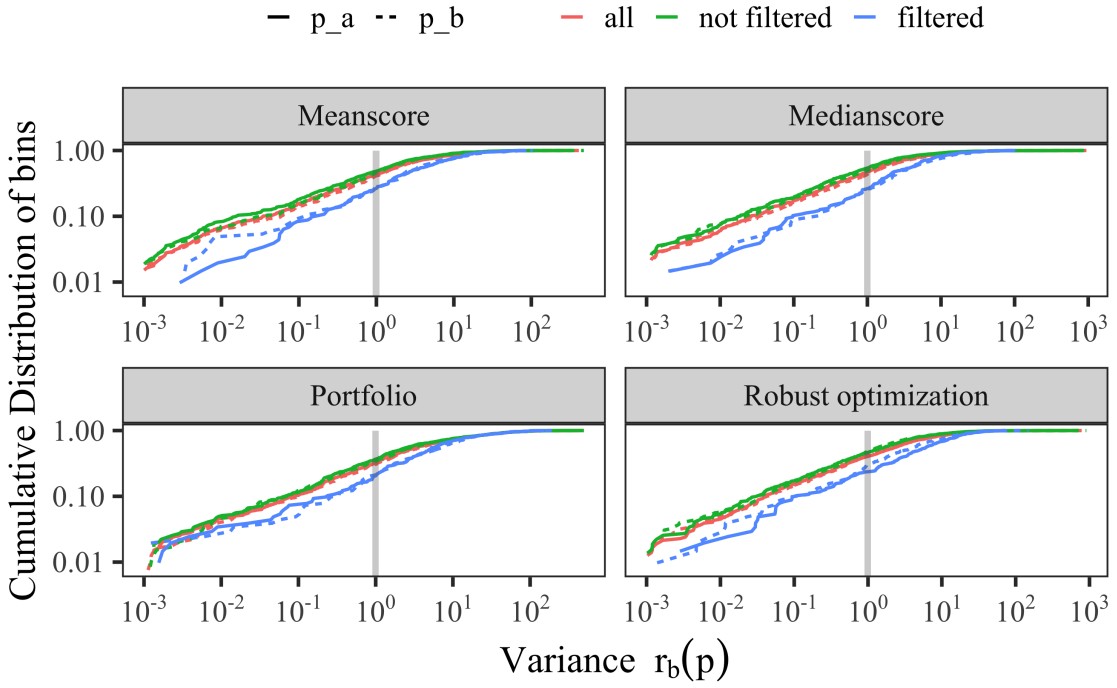

Figure 14: Same as Figure 12 , but for the SHERPA dataset.

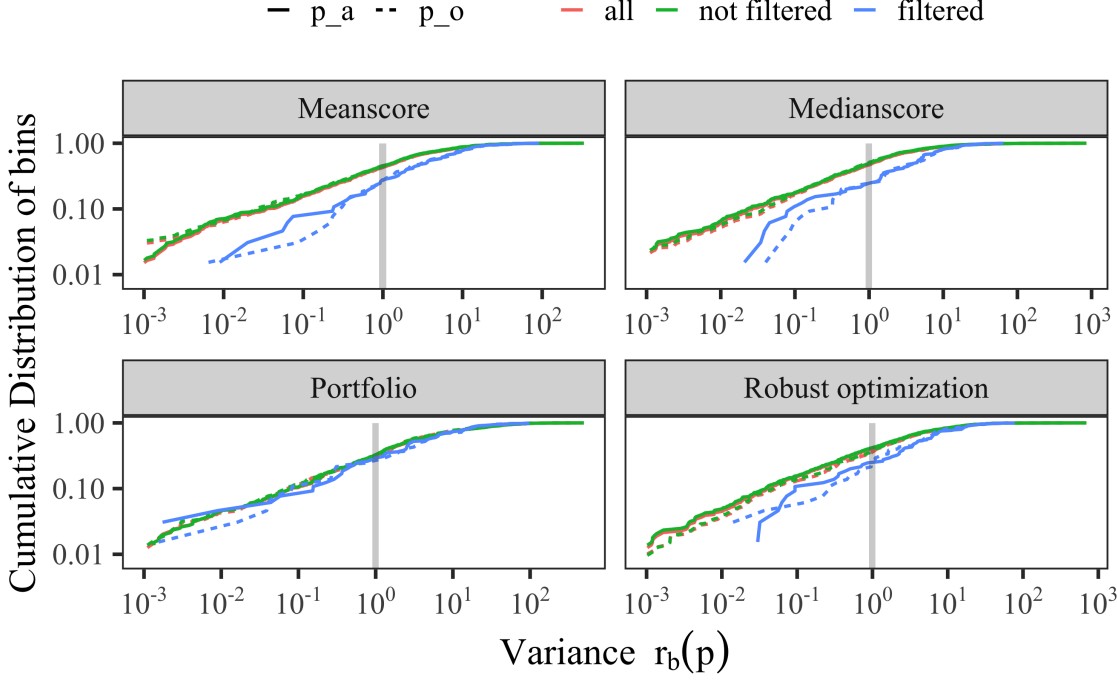

Figure 15: Same as Figure 14, but using observable filtering.

Table 17: Area between ideal cumulative density curve and the cumulative density curves of the robust optimization runs for various hyperparameters $\mu$ for the A14 full dataset (smaller values are better). The data are organized in ascending order of the area between the curves.

| rank | $\mu$ | Area | rank | $\mu$ | Area | rank | $\mu$ | Area | rank | $\mu$ | Area |
|---|---|---|---|---|---|---|---|---|---|---|---|
| 1 | 80 | 7.51e+02 | 26 | 79 | 1.05e+03 | 51 | 24 | 1.32e+03 | 76 | 38 | 1.66e+03 |
| 2 | 78 | 7.93e+02 | 27 | 81 | 1.12e+03 | 52 | 35 | 1.32e+03 | 77 | 29 | 1.72e+03 |
| 3 | 76 | 7.95e+02 | 28 | 71 | 1.13e+03 | 53 | 93 | 1.35e+03 | 78 | 51 | 1.72e+03 |
| 4 | 77 | 8.15e+02 | 29 | 95 | 1.14e+03 | 54 | 89 | 1.39e+03 | 79 | 49 | 1.73e+03 |
| 5 | 73 | 8.53e+02 | 30 | 10 | 1.15e+03 | 55 | 45 | 1.40e+03 | 80 | 57 | 1.73e+03 |
| 6 | 70 | 8.91e+02 | 31 | 11 | 1.17e+03 | 56 | 42 | 1.41e+03 | 81 | 50 | 1.74e+03 |
| 7 | 90 | 8.96e+02 | 32 | 12 | 1.17e+03 | 57 | 41 | 1.43e+03 | 82 | 43 | 1.74e+03 |
| 8 | 26 | 9.09e+02 | 33 | 18 | 1.18e+03 | 58 | 68 | 1.43e+03 | 83 | 44 | 1.76e+03 |
| 9 | 88 | 9.11e+02 | 34 | 3 | 1.19e+03 | 59 | 67 | 1.44e+03 | 84 | 55 | 1.79e+03 |
| 10 | 74 | 9.14e+02 | 35 | 21 | 1.19e+03 | 60 | 46 | 1.44e+03 | 85 | 47 | 1.87e+03 |
| 11 | 86 | 9.42e+02 | 36 | 20 | 1.19e+03 | 61 | 39 | 1.46e+03 | 86 | 60 | 1.93e+03 |
| 12 | 72 | 9.43e+02 | 37 | 16 | 1.19e+03 | 62 | 30 | 1.48e+03 | 87 | 37 | 1.94e+03 |
| 13 | 27 | 9.44e+02 | 38 | 69 | 1.20e+03 | 63 | 63 | 1.48e+03 | 88 | 59 | 1.95e+03 |
| 14 | 83 | 9.47e+02 | 39 | 22 | 1.20e+03 | 64 | 40 | 1.51e+03 | 89 | 33 | 1.96e+03 |
| 15 | 75 | 9.53e+02 | 40 | 23 | 1.21e+03 | 65 | 64 | 1.52e+03 | 90 | 54 | 1.97e+03 |
| 16 | 87 | 9.59e+02 | 41 | 19 | 1.21e+03 | 66 | 28 | 1.55e+03 | 91 | 58 | 1.99e+03 |
| 17 | 8 | 9.61e+02 | 42 | 13 | 1.22e+03 | 67 | 61 | 1.56e+03 | 92 | 53 | 2.08e+03 |
| 18 | 82 | 9.72e+02 | 43 | 25 | 1.23e+03 | 68 | 98 | 1.56e+03 | 93 | 94 | 2.13e+03 |
| 19 | 2 | 9.77e+02 | 44 | 97 | 1.24e+03 | 69 | 62 | 1.58e+03 | 94 | 56 | 2.14e+03 |
| 20 | 84 | 9.80e+02 | 45 | 7 | 1.25e+03 | 70 | 66 | 1.58e+03 | 95 | 52 | 2.23e+03 |
| 21 | 5 | 9.90e+02 | 46 | 15 | 1.27e+03 | 71 | 32 | 1.58e+03 | 96 | 99 | 2.74e+03 |
| 22 | 85 | 9.99e+02 | 47 | 17 | 1.28e+03 | 72 | 48 | 1.59e+03 | 97 | 36 | 3.00e+03 |
| 23 | 1 | 1.01e+03 | 48 | 14 | 1.29e+03 | 73 | 31 | 1.60e+03 | 98 | 34 | 3.02e+03 |
| 24 | 9 | 1.03e+03 | 49 | 92 | 1.30e+03 | 74 | 65 | 1.61e+03 | 99 | 91 | 3.05e+03 |
| 25 | 6 | 1.04e+03 | 50 | 4 | 1.31e+03 | 75 | 96 | 1.64e+03 | 100 | 100 | 3.89e+03 |

Table 18: Best $\mu$ obtained for A14 and SHERPA datasets when unfiltered (*All data*), bin filtered and observable filtered data are used for parameter tuning.

| Dataset | All data | Bin filtered | Observable filtered |
|---|---|---|---|
| A14 | 80 | 76 | 80 |
| SHERPA | 82 | 71 | 73 |

## A.10 Comparison of the rational approximation with the MC generator

We compare the cumulative distribution of bins at different bands of variance levels computed using the rational approximation (RA) model as $r_b(\mathbf{p}) = \frac{(f_b(\mathbf{p}) - \mathcal{R}_b)^2}{\Delta f_b(\mathbf{p})^2 + \Delta \mathcal{R}_b^2}$ and the MC generator model as $\widetilde{r_b(\mathbf{p})} = \frac{(\mathrm{MC}_b(\mathbf{p}) - \mathcal{R}_b)^2}{\Delta \mathrm{MC}_b(\mathbf{p})^2 + \Delta \mathcal{R}_b^2}$, where $\mathbf{p}$ are the parameters obtained from the different tuning approaches. In Figure 5, we showed the plot of this comparison for bins in each category of the A14 dataset using the parameters from three approaches. For completeness, in Figure 16, we show the plot of this comparison for the remaining three approaches.

We observe in Figure 16 that around the variance boundary, except for in the *Track-jet UE* and *Multijets* categories, there is no significant difference in performance between $r_b(\mathbf{p})$ and $\widetilde{r_b(\mathbf{p})}$ for each approach. In the case of *Track-jet UE* and *Multijets* categories, the number of bins that lie within the variance boundary is quite low compared to other categories. This suggests that many bins in these categories cannot be explained well by either the MC generator or the approximation for the optimal tuning parameters reported by the approaches. Additionally, we observe in these categories that the approximations are not able to capture the MC generator perfectly.

Table 19: Bin filtering of A14 data: Shown are the observables from which bins were removed and the number of bins removed. We also show the critical $\chi^2$ values and the $\chi^2$ test statistic before and after bin filtering. If all the bins were removed from the observable then the number of bins removed is shown in bold font and the $\chi^2$ test statistic before and after bin filtering is the same.

| Observable Name | No. of filtered bins | $\chi^2_{c,\mathcal{B}}$ | $\chi^2_{\mathcal{B}}$ before filtering bins | $\chi^2_{\mathcal{B}}$ after filtering bins |
|---|---|---|---|---|
| /ATLAS_2011_I919017/d01-x02-y04 | **11** | 3.84 | 9.77 | 9.77 |
| /ATLAS_2011_I919017/d01-x02-y05 | **13** | 7.81 | 13.51 | 13.51 |
| /ATLAS_2011_I919017/d01-x02-y13 | **11** | 3.84 | 9.43 | 9.43 |
| /ATLAS_2011_I919017/d01-x02-y18 | **11** | 3.84 | 6.20 | 6.20 |
| /ATLAS_2011_I919017/d01-x03-y01 | 11 | 21.03 | 24.00 | 3.57 |
| /ATLAS_2011_I919017/d01-x03-y02 | 4 | 21.03 | 48.57 | 19.72 |
| /ATLAS_2011_I919017/d01-x03-y03 | 2 | 25.00 | 28.99 | 24.72 |
| /ATLAS_2011_I919017/d01-x03-y04 | 2 | 32.67 | 35.36 | 32.19 |
| /ATLAS_2011_I919017/d01-x03-y06 | 10 | 26.30 | 59.81 | 26.13 |
| /ATLAS_2011_I919017/d01-x03-y07 | 7 | 25.00 | 58.78 | 23.91 |
| /ATLAS_2011_I919017/d01-x03-y08 | 5 | 28.87 | 36.98 | 28.27 |
| /ATLAS_2011_I919017/d01-x03-y09 | 6 | 35.17 | 41.10 | 35.10 |
| /ATLAS_2011_I919017/d01-x03-y12 | 3 | 23.68 | 31.51 | 21.33 |
| /ATLAS_2011_I919017/d01-x03-y13 | 15 | 31.41 | 58.77 | 26.18 |
| /ATLAS_2011_I919017/d01-x03-y14 | 12 | 33.92 | 69.51 | 32.38 |
| /ATLAS_2011_I919017/d01-x03-y17 | 3 | 23.68 | 30.48 | 22.60 |
| /ATLAS_2011_I919017/d01-x03-y18 | 1 | 30.14 | 30.65 | 26.75 |
| /ATLAS_2011_I919017/d01-x03-y19 | 12 | 33.92 | 43.45 | 6.13 |
| /ATLAS_2011_I919017/d01-x04-y03 | **22** | 21.03 | 45.11 | 45.11 |
| /ATLAS_2011_I919017/d01-x04-y04 | **21** | 19.68 | 93.99 | 93.99 |
| /ATLAS_2011_I919017/d01-x04-y05 | 4 | 16.92 | 22.81 | 16.74 |
| /ATLAS_2011_I919017/d01-x04-y08 | **22** | 21.03 | 65.21 | 65.21 |
| /ATLAS_2011_I919017/d01-x04-y09 | **22** | 21.03 | 71.99 | 71.99 |
| /ATLAS_2011_I919017/d01-x04-y10 | 2 | 18.31 | 25.18 | 18.27 |
| /ATLAS_2011_I919017/d01-x04-y13 | 12 | 22.36 | 49.09 | 2.36 |
| /ATLAS_2011_I919017/d01-x04-y14 | **24** | 23.68 | 71.30 | 71.30 |
| /ATLAS_2011_I919017/d01-x04-y15 | 4 | 21.03 | 27.53 | 20.80 |
| /ATLAS_2011_I919017/d01-x04-y18 | 2 | 23.68 | 23.77 | 22.57 |
| /ATLAS_2011_I919017/d01-x04-y19 | 8 | 22.36 | 36.75 | 16.78 |
| /ATLAS_2011_I919017/d01-x04-y25 | 3 | 26.30 | 29.14 | 24.98 |
| /ATLAS_2011_I919017/d02-x02-y05 | 1 | 11.07 | 13.84 | 9.87 |
| /ATLAS_2011_I919017/d02-x02-y09 | 1 | 9.49 | 12.32 | 8.52 |
| /ATLAS_2011_I919017/d02-x02-y14 | 1 | 9.49 | 12.19 | 9.18 |
| /ATLAS_2011_I919017/d02-x03-y02 | 15 | 30.14 | 40.31 | 7.63 |
| /ATLAS_2011_I919017/d02-x03-y06 | 3 | 31.41 | 36.64 | 28.59 |
| /ATLAS_2011_I919017/d02-x03-y07 | 4 | 31.41 | 55.51 | 29.12 |
| /ATLAS_2011_I919017/d02-x03-y12 | 7 | 31.41 | 45.41 | 30.04 |
| /ATLAS_2011_I919017/d02-x03-y17 | 1 | 30.14 | 30.64 | 28.11 |
| /ATLAS_2011_I919017/d02-x04-y03 | 10 | 26.30 | 46.87 | 19.20 |
| /ATLAS_2011_I919017/d02-x04-y04 | **25** | 25.00 | 136.83 | 136.83 |
| /ATLAS_2011_I919017/d02-x04-y05 | **28** | 28.87 | 74.75 | 74.75 |
| /ATLAS_2011_I919017/d02-x04-y08 | 16 | 27.59 | 82.23 | 25.29 |
| /ATLAS_2011_I919017/d02-x04-y09 | **27** | 27.59 | 156.13 | 156.13 |
| /ATLAS_2011_I919017/d02-x04-y10 | **30** | 31.41 | 126.00 | 126.00 |
| /ATLAS_2011_I919017/d02-x04-y13 | 14 | 26.30 | 71.23 | 23.47 |
| /ATLAS_2011_I919017/d02-x04-y14 | **27** | 27.59 | 103.20 | 103.20 |
| /ATLAS_2011_I919017/d02-x04-y15 | 9 | 28.87 | 70.47 | 26.49 |
| /ATLAS_2011_I919017/d02-x04-y18 | 3 | 26.30 | 32.01 | 25.80 |
| /ATLAS_2011_I919017/d02-x04-y19 | 13 | 28.87 | 67.53 | 23.91 |

| | | | | |
|---|---|---|---|---|
| /ATLAS_2011_I919017/d02-x04-y20 | 10 | 28.87 | 57.69 | 28.31 |
| /ATLAS_2011_I919017/d02-x04-y24 | 10 | 28.87 | 43.46 | 28.24 |
| /ATLAS_2011_I919017/d02-x04-y25 | 3 | 31.41 | 39.98 | 28.20 |
| /ATLAS_2011_ZPT/d02-x01-y01 | 1 | 14.07 | 15.77 | 14.06 |
| /ATLAS_2011_ZPT/d02-x02-y02 | 2 | 14.07 | 16.94 | 13.93 |
| /ATLAS_2011_ZPT/d03-x01-y01 | 1 | 14.07 | 15.32 | 13.85 |
| /ATLAS_2013_JETUE/d08-x01-y03 | 1 | 12.59 | 19.97 | 11.51 |

Table 20: Bin filtering of SHERPA data: Shown are the observables from which bins were removed and the number of bins removed. We also show the critical $\chi^2$ values and the $\chi^2$ test statistic before and after bin filtering. If all the bins were removed from the observable then the number of bins removed is shown in bold font and the $\chi^2$ test statistic before and after bin filtering is the same.

| Observable Name | No. of filtered bins | $\chi^2_{c,\mathcal{B}}$ | $\chi^2_{\mathcal{B}}$ before filtering bins | $\chi^2_{\mathcal{B}}$ after filtering bins |
|---|---|---|---|---|
| /DELPHI_1996_S3430090/d02-x01-y01 | **17** | 9.49 | 35.19 | 35.19 |
| /DELPHI_1996_S3430090/d04-x01-y01 | **17** | 9.49 | 24.89 | 24.89 |
| /DELPHI_1996_S3430090/d06-x01-y01 | **21** | 15.51 | 41.59 | 41.59 |
| /DELPHI_1996_S3430090/d07-x01-y01 | **22** | 16.92 | 83.91 | 83.91 |
| /DELPHI_1996_S3430090/d08-x01-y01 | **26** | 22.36 | 80.11 | 80.11 |
| /DELPHI_1996_S3430090/d10-x01-y01 | 2 | 11.07 | 15.90 | 10.59 |
| /DELPHI_1996_S3430090/d11-x01-y01 | **20** | 14.07 | 94.30 | 94.30 |
| /DELPHI_1996_S3430090/d16-x01-y01 | **14** | 3.84 | 17.63 | 17.63 |
| /DELPHI_1996_S3430090/d18-x01-y01 | **23** | 18.31 | 101.31 | 101.31 |
| /DELPHI_1996_S3430090/d19-x01-y01 | **21** | 15.51 | 59.12 | 59.12 |
| /DELPHI_1996_S3430090/d20-x01-y01 | **16** | 7.81 | 20.48 | 20.48 |
| /DELPHI_1996_S3430090/d33-x01-y01 | 5 | 52.19 | 75.18 | 50.31 |

## A.11 Optimal parameter values for the A14 dataset with the rational approximation

To better visually compare the different solutions obtained with our optimization methods, we show the [0,1]-scaled optimal parameter values in Figure 17.

## A.12 Results for using the cubic polynomial to approximate the MC simulation

In the main paper, we showed the numerical results when using a rational approximation of the MC simulation during tuning. In the A14 publication [3], a cubic polynomial was used. Thus, in this section, we present the results obtained with our optimization methods when using a cubic polynomial instead of a rational approximation.

### A.12.1 Comparison metric outcomes for the A14 dataset using the cubic polynomial approximation

Tables 22 shows the comparison metrics we introduced in the main paper in Section 4.2 when using the cubic polynomial approximation for the full data, the observable-filtered data, and the bin-filtered data, respectively. We see that for all three cases and most criteria (except for the D-optimality in the observable-filtered case), our automated methods for adjusting the observable weights perform better than the expert solution (i.e., using the parameters published in [3]).

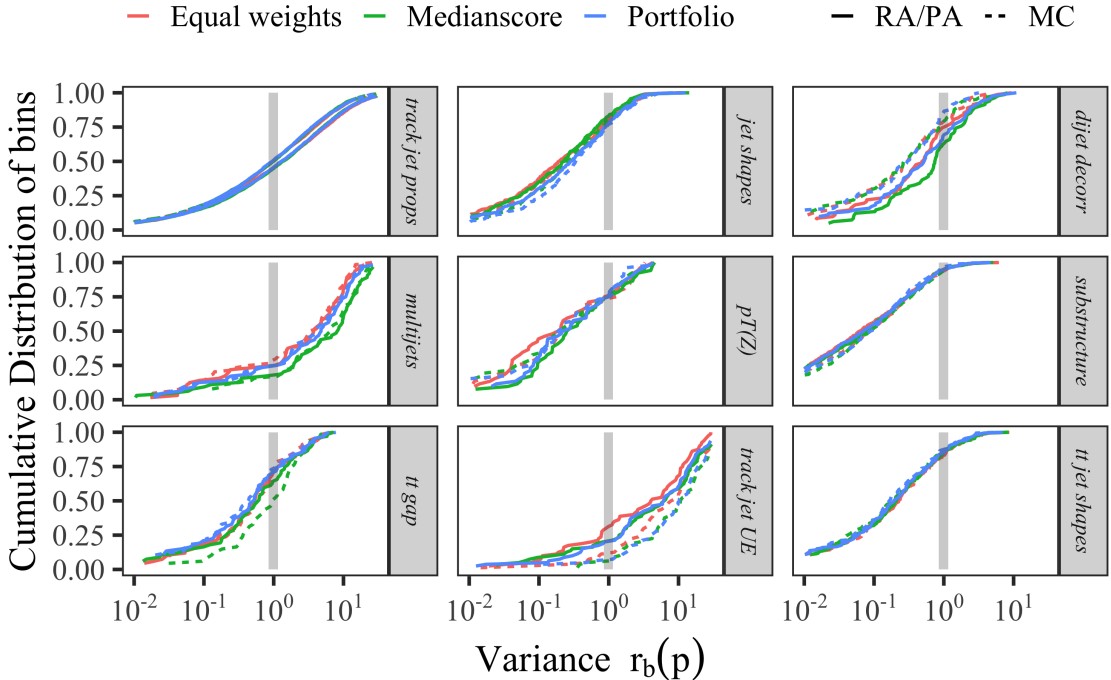

Figure 16: Cumulative distribution of bins in each category of the A14 dataset at different bands of variance levels computed with the rational approximation (RA) given by, $r_b(\mathbf{p}) = \frac{(f_b(\mathbf{p}) - \mathcal{R}_b)^2}{\Delta f_b(\mathbf{p})^2 + \Delta \mathcal{R}_b^2}$ and the MC simulation given by $\widetilde{r_b(\mathbf{p})} = \frac{(MC_b(\mathbf{p}) - \mathcal{R}_b)^2}{\Delta MC_b(\mathbf{p})^2 + \Delta \mathcal{R}_b^2}$

Table 21: A14 results with the *full dataset*, *observable-filtered dataset* and *bin-filtered dataset* when using the *cubic polynomial* approximation, calculated on the full dataset. Lower numbers are better. The best results are in bold. In each dataset, W-$\chi^2$ refers to the Weighted $\chi^2$ metric, A-o refers to the A-opt metric, and l-D-o refers to the log D-opt metric.

| Data | full dataset | | | observable-filtered dataset | | | bin-filtered dataset | | |
|---|---|---|---|---|---|---|---|---|---|
| Method | W-$\chi^2$ | A-o | l-D-o | W-$\chi^2$ | A-o | l-D-o | W-$\chi^2$ | A-o | l-D-o |
| Bilevel-meanscore | 0.1290 | 0.5358 | -66.0364 | 0.1079 | 0.8082 | -61.9210 | 0.1244 | 0.7147 | -64.7848 |
| Bilevel-medscore | 0.1645 | **0.4114** | **-70.0545** | 0.1702 | **0.4955** | -66.6920 | 0.2171 | 0.5433 | -69.7202 |
| Bilevel-portfolio | 0.1900 | 0.6590 | -63.0378 | 0.1764 | 0.7408 | -61.3839 | **0.1159** | 0.5205 | **-70.1573** |
| Expert tune | 0.1306 | 0.5466 | -68.6511 | 0.1306 | 0.5466 | **-68.6511** | 0.1306 | 0.5466 | -68.6511 |
| All-weights-equal | 0.1034 | 0.5553 | -65.6099 | 0.1049 | 0.6689 | -63.6502 | 0.1406 | **0.4122** | -69.2732 |
| Robust optimization | **0.0697** | 0.9749 | -66.7931 | **0.0829** | 1.0574 | -66.3665 | 0.1234 | 0.8075 | -67.1015 |

### A.12.2 Optimal parameter values for the A14 dataset using the cubic polynomial approximation

Table 23 shows the optimal values for the tuned parameters obtained by all methods for the A14 dataset when using all observables in the tune. For Bilevel-meanscore, -medianscore and -portfolio, we repeated the experiments three times using different random number seeds and we report the best results among the three trials based on their respective objective functions.

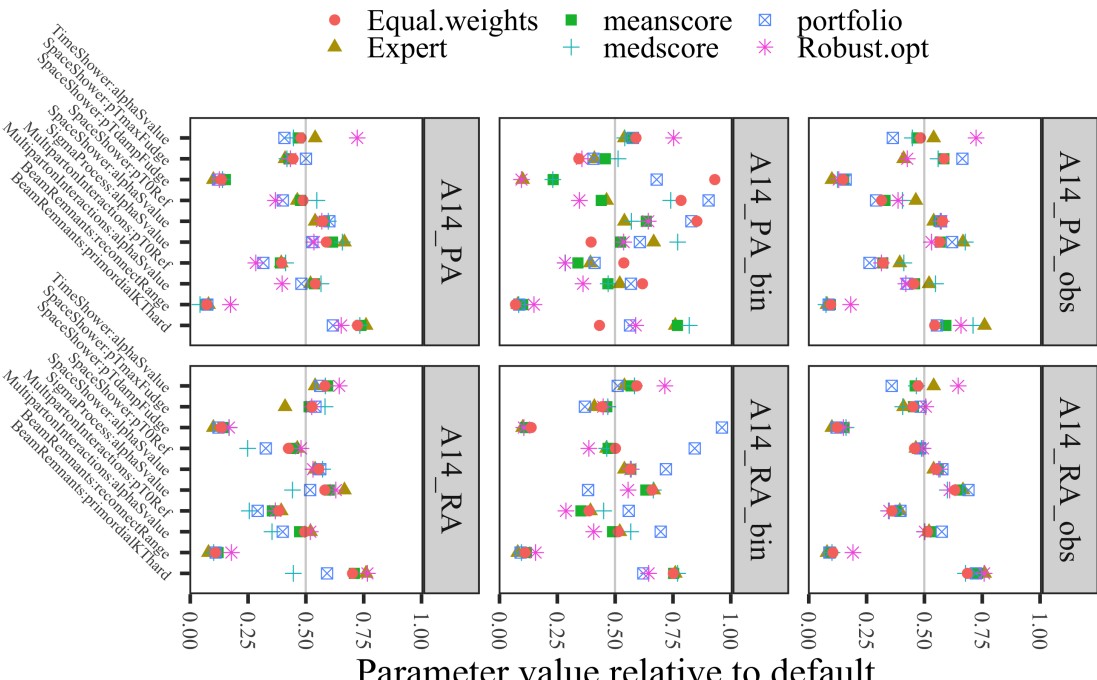

Figure 17: Optimal parameter values for the A14 dataset obtained when using all, bin-filtered (_bin) and observable-filtered (_obs) data in the optimization and the polynomial approximation (PA) and rational approximation (RA). Values are normalized to [0,1].

Table 22: A14 results with the *full dataset*, *observable-filtered dataset* and *bin-filtered dataset* when using the *cubic polynomial* approximation, calculated on the reduced dataset. Lower numbers are better. The best results are in bold. In each dataset, W-$\chi^2$ refers to the Weighted $\chi^2$ metric, A-o refers to the A-opt metric, and l-D-o refers to the log D-opt metric.

| Data | full dataset | | | observable-filtered dataset | | | bin-filtered dataset | | |
|---|---|---|---|---|---|---|---|---|---|
| Method | W-$\chi^2$ | A-o | l-D-o | W-$\chi^2$ | A-o | l-D-o | W-$\chi^2$ | A-o | l-D-o |
| Bilevel-meanscore | 0.1290 | 0.5358 | -66.0364 | 0.1079 | 0.8082 | -61.9210 | 0.0778 | 1.0199 | -60.6441 |
| Bilevel-medscore | 0.1645 | **0.4114** | **-70.0545** | 0.1702 | **0.4955** | -66.6920 | 0.1085 | 0.7208 | -67.4322 |
| Bilevel-portfolio | 0.1900 | 0.6590 | -63.0378 | 0.1764 | 0.7408 | -61.3839 | 0.0738 | **0.4231** | **-69.4117** |
| Expert tune | 0.1306 | 0.5466 | -68.6511 | 0.0799 | 0.5542 | **-68.6748** | 0.0456 | 0.8985 | -63.5606 |
| All-weights-equal | 0.1034 | 0.5553 | -65.6099 | 0.0857 | 0.6769 | -63.6881 | **0.0379** | 0.7390 | -63.8424 |
| Robust optimization | **0.0697** | 0.9749 | -66.7931 | **0.0829** | 1.0574 | -66.3665 | 0.0642 | 0.9559 | -64.8659 |

From these results, we can see that the Bilevel-medianscore method leads to a solution that is closest to the expert's solution.

To better visually compare the different solutions obtained with our methods, we show the [0,1]-scaled optimal values in Figure 17. We can see that there are differences between the optimal parameters obtained with the different methods, in particular, the results of the robust optimization method tend to be further away from the expert's solution for parameters

Table 23: Optimal parameter values for the A14 dataset obtained when using all observables in the optimization and the *cubic polynomial* approximation.

| ID | Parameter name | Expert | Bil.-meanscore | Bil.-medianscore | Bil.-portfolio | Robust opt | All-weights-equal |
|----|----------------|--------|----------------|------------------|----------------|------------|-------------------|
| 1 | `SigmaProcess:alphaSvalue` | 0.143 | 0.139 | 0.141 | 0.140 | 0.136 | 0.138 |
| 2 | `BeamRemnants:primordialKThard` | 1.904 | 1.867 | 1.884 | 1.866 | 1.826 | 1.862 |
| 3 | `SpaceShower:pT0Ref` | 1.643 | 1.632 | 1.735 | 1.651 | 1.395 | 1.603 |
| 4 | `SpaceShower:pTmaxFudge` | 0.908 | 0.939 | 0.904 | 0.988 | 0.933 | 0.944 |
| 5 | `SpaceShower:pTdampFudge` | 1.046 | 1.079 | 1.069 | 1.047 | 1.063 | 1.067 |
| 6 | `SpaceShower:alphaSvalue` | 0.123 | 0.129 | 0.130 | 0.130 | 0.128 | 0.129 |
| 7 | `TimeShower:alphaSvalue` | 0.128 | 0.123 | 0.124 | 0.121 | 0.136 | 0.124 |
| 8 | `MultipartonInteractions:pT0Ref` | 2.149 | 2.083 | 2.065 | 2.039 | 1.925 | 2.092 |
| 9 | `MultipartonInteractions:alphaSvalue` | 0.128 | 0.127 | 0.127 | 0.126 | 0.120 | 0.127 |
| 10 | `BeamRemnants:reconnectRange` | 1.792 | 1.531 | 1.405 | 1.591 | 2.567 | 1.636 |
| | Euclidean distance from the expert solution | | 0.246 | 0.235 | 0.428 | 0.451 | 0.259 |

Table 24: Optimal parameter values for A14 when using the *cubic polynomial* approximation with all methods after outlier detection to filter out observables that cannot be approximated well by the model.

| ID | Parameter name | Expert | Bilevel-meanscore | Bilevel-medianscore | Bilevel-portfolio | Robust opt | All-weights-equal |
|----|----------------|--------|-------------------|---------------------|-------------------|------------|-------------------|
| 1 | `SigmaProcess:alphaSvalue` | 0.143 | 0.136 | 0.141 | 0.137 | 0.136 | 0.137 |
| 2 | `BeamRemnants:primordialKThard` | 1.904 | 1.793 | 1.853 | 1.754 | 1.829 | 1.772 |
| 3 | `SpaceShower:pT0Ref` | 1.643 | 1.329 | 1.369 | 1.218 | 1.425 | 1.301 |
| 4 | `SpaceShower:pTmaxFudge` | 0.908 | 1.079 | 1.088 | 1.223 | 0.926 | 1.085 |
| 5 | `SpaceShower:pTdampFudge` | 1.046 | 1.069 | 1.053 | 1.101 | 1.065 | 1.074 |
| 6 | `SpaceShower:alphaSvalue` | 0.123 | 0.129 | 0.128 | 0.129 | 0.129 | 0.129 |
| 7 | `TimeShower:alphaSvalue` | 0.128 | 0.124 | 0.123 | 0.116 | 0.136 | 0.124 |
| 8 | `MultipartonInteractions:pT0Ref` | 2.149 | 1.971 | 2.098 | 1.870 | 1.971 | 1.983 |
| 9 | `MultipartonInteractions:alphaSvalue` | 0.128 | 0.122 | 0.126 | 0.120 | 0.121 | 0.123 |
| 10 | `BeamRemnants:reconnectRange` | 1.792 | 1.812 | 1.614 | 1.714 | 2.632 | 1.851 |
| | Euclidean distance from the expert solution | | 0.447 | 0.279 | 0.553 | 0.432 | 0.480 |

1, 2, 3, 7, 8, 9 and 10. The results of the portfolio optimization differ from the expert tune in particular for parameters 1, 2, 3, 4 and 7. The mean- and medianscore results are very similar to each other as well as to the expert's solution.

We conducted a similar analysis on the observable- and bin-filtered data. Table 24 shows the optimal parameter values that we obtain with the automated optimization methods after filtering out the 12 observables that the model cannot explain (see also Section 3.1). The expert solution is the same as before and based on all observables. We include it for easier comparison. With only a few exceptions, all parameters obtained with the automated optimizations change (as compared to using the full dataset). Figure 17 shows the optimal parameter values obtained with each method scaled to [0,1]. In comparison to when using the full dataset, we see that the results of the robust optimization now agree better with the expert's tune for parameters 3, 4, and 8, but less agreement is achieved for parameter 10. Of the three bilevel methods, the medianscore objective function leads to optimal parameters that are most similar to the expert tune.

In Table 25 and Figure 17 we show the optimal parameter values obtained with our methods after applying the bin-filtering approach described in Section 3.2 in the main document. In comparison to our results that do not use any filtering, we can see a much larger disagreement in the optimal parameters for all methods. In fact, all methods yield optimal parameters that are significantly further away from the expert's solution, except for parameters 7 and 10. The Euclidean distance between the optimal parameters obtained by our proposed methods and the expert solution shows that the bilevel-medianscore method leads to the most similar parameter values while all the other methods lead to very different tunes.

### A.12.3 Comparison of optimal weights for the A14 dataset with cubic polynomial approximation

In Table 26 we present the optimal weights assigned to each observable group by each method following the presentation style in [3]. The weights reported for our method are averages of

Table 25: Optimal parameter values obtained for A14 with the *cubic polynomial* approximation with all methods after using the bin-filtering approach that excludes individual bins from the optimization.

| ID | Parameter name | Expert | Bilevel-meanscore | Bilevel-medianscore | Bilevel-portfolio | Robust opt | All-weights-equal |
|---|---|---|---|---|---|---|---|
| 1 | SigmaProcess:alphaSvalue | 0.143 | 0.141 | 0.143 | 0.136 | 0.136 | 0.132 |
| 2 | BeamRemnants:primordialKThard | 1.904 | 1.919 | 1.918 | 1.575 | 1.794 | 1.716 |
| 3 | SpaceShower:pT0Ref | 1.643 | 1.802 | 2.284 | 2.300 | 1.355 | 2.123 |
| 4 | SpaceShower:pTmaxFudge | 0.908 | 0.968 | 1.014 | 0.920 | 0.856 | 0.843 |
| 5 | SpaceShower:pTdampFudge | 1.046 | 1.071 | 1.147 | 1.442 | 1.047 | 1.465 |
| 6 | SpaceShower:alphaSvalue | 0.123 | 0.130 | 0.130 | 0.144 | 0.132 | 0.143 |
| 7 | TimeShower:alphaSvalue | 0.128 | 0.129 | 0.127 | 0.131 | 0.138 | 0.130 |
| 8 | MultipartonInteractions:pT0Ref | 2.149 | 2.059 | 1.800 | 2.228 | 1.925 | 2.306 |
| 9 | MultipartonInteractions:alphaSvalue | 0.128 | 0.126 | 0.120 | 0.131 | 0.118 | 0.131 |
| 10 | BeamRemnants:reconnectRange | 1.792 | 1.860 | 1.922 | 1.807 | 2.340 | 1.622 |
| | Euclidean distance from the expert solution | | 0.376 | 0.354 | 0.848 | 0.525 | 1.111 |

the weights over all observables that belong to the same group. We scaled the weights such that they are on equal footing (all add up to 4580).

The largest differences between the expert-adjusted values and the values determined by our methods are for *Multijets*, $t\bar{t}$ *gap* and *Jet UE*, while for the remaining groups, the values are very similar. These results, together with our analysis above let us conclude that an automated method for adjusting the weights of observables for tuning parameters is a viable approach and can lead to better results than hand-tuning.

### A.12.4 Optimal parameter values for the SHERPA dataset with rational approximation

For a better visual comparison of the different solutions obtained with our methods, we show the [0,1]-scaled optimal values in Figure 18. Compared to the results for the A14 dataset, we see that there are significant differences between the optimal parameters obtained with the different methods.

### A.12.5 Optimal parameter values for the SHERPA dataset with the cubic polynomial approximation

The physics parameters **p** and their optimization ranges are shown in Table 16. Tables 27, 28 and 29 shows the optimal values for the physics parameters obtained by all methods when no filtering was applied before optimization, after using outlier detection to remove observables from the optimization, and after using the bin-filtering approach that excludes individual bins from the optimization, respectively. For an illustrative comparison, we show the [0,1]-scaled optimal parameter values in Figure 18. The default values lie right in the middle of the parameter range.

### A.12.6 Comparison metric outcomes for the SHERPA dataset with the cubic polynomial approximation

Tables 31 shows the comparison metrics of our experiments when using the cubic polynomial approximation for the full data, the observable-filtered data, and the bin-filtered data, respectively. Smaller numbers indicate better performance. The smallest number of each metric is bold for better visualization.

Based on these results, we can see that the all-weights-equal method (i.e. not adjusting any weights) has the best performance for the full dataset under the A- and D-optimality. The bilevel-portfolio method performs best under the A- and D-optimality for both the observable- and bin-filtered datasets. The robust optimization method performs best in all three cases under the Weighted $\chi^2$ criterion.

Table 26: Comparison of the optimal weights obtained by each method using the *cubic polynomial* approximation. The observable grouping corresponds to the same grouping used in [3].

| | expert | Bilevel-meanscore | Bilevel-medianscore | Bilevel-portfolio | robustopt |
|---|---|---|---|---|---|
| **Track jet properties** | | | | | |
| Charged jet multiplicity (50 distributions) | 10 | 10.74 | 14.98 | 10.64 | 19.38 |
| Charged jet $z$ (50 distributions) | 10 | 11.29 | 8.66 | 13.71 | 0.00 |
| Charged jet $p_T^{rel}$ (50 distributions) | 10 | 11.20 | 10.39 | 10.99 | 0.00 |
| Charged jet $\rho_{ch}(r)$ (50 distributions) | 10 | 11.57 | 10.58 | 12.55 | 0.00 |
| **Jet shapes** | | | | | |
| Jet shape $\rho$ (59 distributions) | 10 | 11.57 | 11.06 | 10.20 | 19.38 |
| **Dijet decorr** | | | | | |
| Decorrelation $\Delta\phi$ (Fit range: $\Delta\phi > 0.75$) (9 distributions) | 20 | 12.39 | 8.37 | 9.39 | 15.07 |
| **Multijets** | | | | | |
| 3-to-2 jet ratios (8 distributions) | 100 | 12.99 | 27.19 | 5.88 | 19.38 |
| **$p_T^Z$ (Fit range: $p_T^Z < 50$GeV)** | | | | | |
| Z-boson $p_T$ (20 distributions) | 10 | 12.78 | 14.53 | 6.71 | 19.38 |
| **Substructure** | | | | | |
| Jet mass, $\sqrt{d_{12}}$, $\sqrt{d_{23}}$, $\tau_{21}$, $\tau_{23}$ (36 distributions) | 5 | 10.55 | 9.91 | 9.74 | 15.61 |
| **$t\bar{t}$ gap** | | | | | |
| Gap fraction vs $Q_0$, $Q_{sum}$ for $|y| < 0.8$ | 100 | 0.18 | 2.10 | 3.88 | 19.38 |
| Gap fraction vs $Q_0$, $Q_{sum}$ for $0.8 < |y| < 1.5$ | 80 | 0.75 | 9.52 | 5.71 | 19.38 |
| Gap fraction vs $Q_0$, $Q_{sum}$ for $1.5 < |y| < 2.1$ | 40 | 7.93 | 8.31 | 39.20 | 19.38 |
| Gap fraction vs $Q_0$, $Q_{sum}$ for $|y| < 2.1$ | 10 | 18.19 | 13.43 | 11.05 | 19.38 |
| **Track-jet UE** | | | | | |
| Transverse region $N_{ch}$ profiles (5 distributions) | 10 | 15.87 | 13.45 | 13.53 | 19.38 |
| Transverse region mean $p_T$ profiles for $R = 0.2, 0.4, 0.6$ (3 distributions) | 10 | 7.56 | 11.72 | 10.30 | 19.38 |
| **$t\bar{t}$ jet shapes** | | | | | |
| Jet shapes $\rho(r)$, $\psi(r)$ (20 distributions) | 5 | 10.86 | 10.91 | 12.25 | 10.66 |
| **Jet UE** | | | | | |
| Transverse, trans-max, trans-min sum $p_T$ incl. profiles (3 distributions) | 20 | 12.76 | 22.51 | 9.65 | 19.38 |
| Transverse, trans-max, trans-min $N_{ch}$ incl. profiles (3 distributions) | 20 | 15.57 | 9.65 | 6.01 | 19.38 |
| Transverse sum $E_T$ incl. profiles (2 distributions) | 20 | 12.71 | 12.75 | 25.03 | 3.73 |
| Transverse sum $ET/$sum $p_T$ ratio incl., excl. profiles (2 distributions) | 5 | 7.53 | 18.29 | 28.35 | 19.38 |
| Transverse mean $p_T$ incl. profiles (2 distributions) | 10 | 7.65 | 7.45 | 13.34 | 19.38 |
| Transverse, trans-max, trans-min sum $p_T$ incl. distributions (15 distributions) | 1 | 9.39 | 5.50 | 11.04 | 19.38 |
| Transverse, trans-max, trans-min sum $N_{ch}$ incl. distributions (15 distributions) | 1 | 11.92 | 9.85 | 14.52 | 19.38 |

Table 27: Optimal parameter values for the SHERPA dataset obtained with all methods using the *cubic polynomial* approximation when no filtering was applied before optimization (88 observables).

| ID | Parameter name | Default | Bilevel-meanscore | Bilevel-medscore | Bilevel-portfolio | Robust opt | All-weights-equal |
|---|---|---|---|---|---|---|---|
| 1 | KT_0 | 1.00 | 0.850 | 0.837 | 0.903 | 0.870 | 0.853 |
| 2 | ALPHA_G | 1.25 | 0.626 | 0.626 | 0.626 | 1.874 | 0.626 |
| 3 | ALPHA_L | 2.50 | 3.634 | 2.022 | 3.108 | 1.252 | 3.749 |
| 4 | BETA_L | 0.10 | 0.150 | 0.069 | 0.050 | 0.150 | 0.150 |
| 5 | GAMMA_L | 0.50 | 0.250 | 0.353 | 0.750 | 0.619 | 0.286 |
| 6 | ALPHA_H | 2.50 | 3.455 | 2.047 | 1.251 | 2.712 | 3.454 |
| 7 | BETA_H | 0.75 | 0.736 | 0.610 | 0.657 | 0.573 | 0.922 |
| 8 | GAMMA_H | 0.10 | 0.144 | 0.124 | 0.050 | 0.150 | 0.140 |
| 9 | STRANGE_FRACTION | 0.50 | 0.531 | 0.521 | 0.529 | 0.514 | 0.497 |
| 10 | BARYON_FRACTION | 0.18 | 0.099 | 0.132 | 0.091 | 0.139 | 0.104 |
| 11 | P_QS_by_P_QQ_norm | 0.48 | 0.720 | 0.617 | 0.502 | 0.601 | 0.720 |
| 12 | P_SS_by_P_QQ_norm | 0.02 | 0.010 | 0.030 | 0.030 | 0.030 | 0.010 |
| 13 | P_QQ1_by_P_QQ0 | 1.00 | 1.499 | 1.499 | 1.349 | 1.164 | 1.499 |
| | Euclidean distance from the default solution | | 1.508 | 1.130 | 1.400 | 1.236 | 1.497 |

## A.13 Weights assigned by different fitting methods

Figure 19 shows the weights per observable obtained from the tune to SHERPA using the methods described in this paper.

## A.14 Coefficients of the approximation function of the toy model

In Table 32, we give the coefficients of the approximation $f_b(\mathbf{p})$ for each bin $b$, which is a linear function of the form $\mathbf{a}^T\mathbf{p} + c$ of the toy model from the closure test described in Section 4.8.

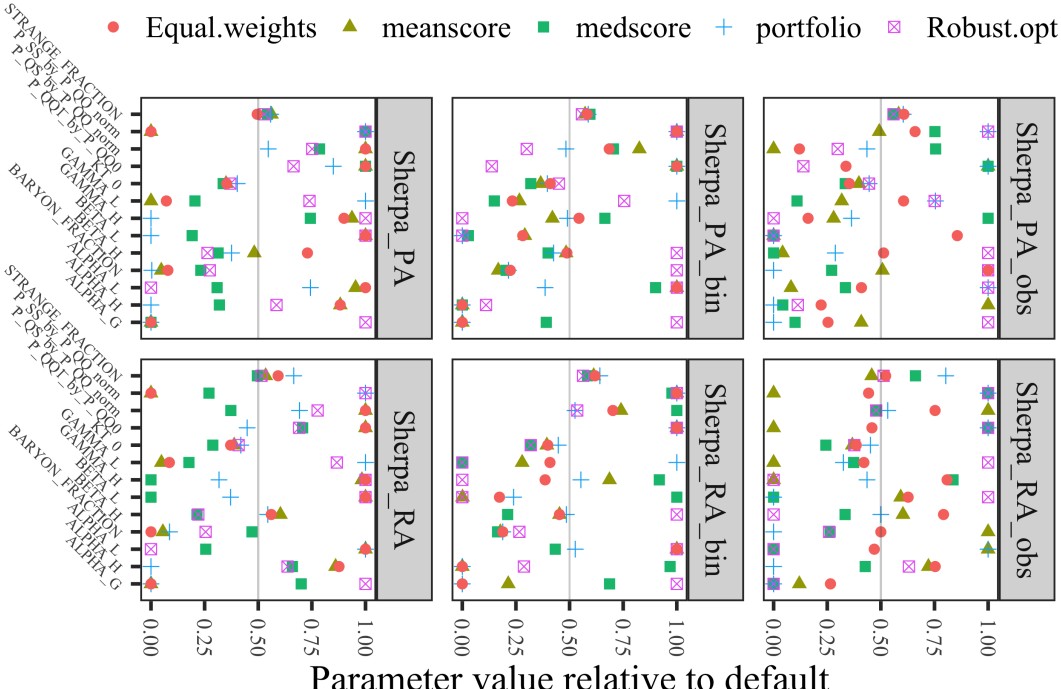

Figure 18: Comparison of the optimal parameter values for SHERPA obtained with the different optimization methods when no, observable, and bin data filtering was applied and the *rational* and polynomial approximation was used. Values are normalized to [0,1].

Table 28: Optimal parameter values for the SHERPA dataset obtained with all methods using the *cubic polynomial* approximation after using outlier detection to remove observables from the optimization (3 observables removed).

| ID | Parameter name | Default | Bilevel-meanscore | Bilevel-medscore | Bilevel-portfolio | Robust opt | All-weights-equal |
|----|----------------|---------|-------------------|------------------|-------------------|------------|-------------------|
| 1 | KT_0 | 1.00 | 0.898 | 0.834 | 0.946 | 0.945 | 0.853 |
| 2 | ALPHA_G | 1.25 | 1.136 | 0.751 | 0.626 | 1.874 | 0.942 |
| 3 | ALPHA_L | 2.50 | 1.454 | 2.088 | 3.749 | 3.749 | 2.275 |
| 4 | BETA_L | 0.10 | 0.050 | 0.050 | 0.050 | 0.050 | 0.136 |
| 5 | GAMMA_L | 0.50 | 0.409 | 0.305 | 0.627 | 0.626 | 0.553 |
| 6 | ALPHA_H | 2.50 | 3.748 | 1.358 | 1.251 | 1.533 | 1.804 |
| 7 | BETA_H | 0.75 | 0.406 | 0.375 | 0.591 | 1.125 | 0.760 |
| 8 | GAMMA_H | 0.10 | 0.078 | 0.150 | 0.086 | 0.050 | 0.066 |
| 9 | STRANGE_FRACTION | 0.50 | 0.541 | 0.528 | 0.552 | 0.529 | 0.553 |
| 10 | BARYON_FRACTION | 0.18 | 0.181 | 0.139 | 0.090 | 0.270 | 0.270 |
| 11 | P_QS_by_P_QQ_norm | 0.48 | 0.240 | 0.602 | 0.449 | 0.384 | 0.298 |
| 12 | P_SS_by_P_QQ_norm | 0.02 | 0.020 | 0.025 | 0.030 | 0.030 | 0.023 |
| 13 | P_QQ1_by_P_QQ0 | 1.00 | 1.499 | 1.499 | 1.499 | 0.639 | 0.837 |
| | Euclidean distance from the default solution | | 1.222 | 1.327 | 1.378 | 1.463 | 0.937 |

Table 29: Optimal parameter values for the SHERPA dataset obtained with all methods using the *cubic polynomial* approximation after using the bin-filtering approach that excludes individual bins from the optimization (204 bins out of 5246 total bins were removed).

| ID | Parameter name | Default | Bilevel-meanscore | Bilevel-medscore | Bilevel-portfolio | Robust opt | All-weights-equal |
|---|---|---|---|---|---|---|---|
| 1 | KT_0 | 1.00 | 0.866 | 0.820 | 0.897 | 0.950 | 0.911 |
| 2 | ALPHA_G | 1.25 | 0.626 | 1.114 | 0.626 | 1.874 | 0.626 |
| 3 | ALPHA_L | 2.50 | 3.749 | 3.502 | 2.216 | 3.749 | 3.749 |
| 4 | BETA_L | 0.10 | 0.079 | 0.053 | 0.050 | 0.050 | 0.078 |
| 5 | GAMMA_L | 0.50 | 0.383 | 0.325 | 0.750 | 0.627 | 0.367 |
| 6 | ALPHA_H | 2.50 | 1.251 | 1.251 | 1.251 | 1.527 | 1.251 |
| 7 | BETA_H | 0.75 | 0.738 | 0.675 | 0.694 | 1.125 | 0.741 |
| 8 | GAMMA_H | 0.10 | 0.092 | 0.116 | 0.099 | 0.050 | 0.104 |
| 9 | STRANGE_FRACTION | 0.50 | 0.536 | 0.547 | 0.543 | 0.529 | 0.541 |
| 10 | BARYON_FRACTION | 0.18 | 0.120 | 0.127 | 0.129 | 0.270 | 0.130 |
| 11 | P_QS_by_P_QQ_norm | 0.48 | 0.636 | 0.578 | 0.472 | 0.384 | 0.569 |
| 12 | P_SS_by_P_QQ_norm | 0.02 | 0.030 | 0.030 | 0.030 | 0.030 | 0.030 |
| 13 | P_QQ1_by_P_QQ0 | 1.00 | 1.499 | 1.499 | 1.499 | 0.637 | 1.499 |
| | Euclidean distance from the default solution | | 1.263 | 1.215 | 1.272 | 1.464 | 1.224 |

Table 30: Results for the comparison metrics for the full, observable-filtered and bin-filtered SHERPA dataset using the *cubic polynomial* approximation, calculated on the full dataset. The best results are in bold. In each dataset, W-$\chi^2$ refers to the Weighted $\chi^2$ metric, A-o refers to the A-opt metric, and l-D-o refers to the log D-opt metric. Note that we do not have an expert solution for this dataset.

| Data | full dataset | | | observable-filtered dataset | | | bin-filtered dataset | | |
|---|---|---|---|---|---|---|---|---|---|
| Method | W-$\chi^2$ | A-o | l-D-o | W-$\chi^2$ | A-o | l-D-o | W-$\chi^2$ | A-o | l-D-o |
| Bilevel-meanscore | 0.1777 | 9.0959 | -39.9863 | 0.4740 | 14.3374 | -35.2608 | 0.2504 | 17.2334 | -32.7683 |
| Bilevel-medscore | 0.2370 | 13.3943 | -37.1420 | 0.4786 | 13.6299 | -36.6594 | 0.1835 | 16.9248 | -32.1289 |
| Bilevel-portfolio | 0.3409 | 8.7863 | -39.6956 | 0.2139 | **10.4481** | **-36.8254** | 0.2906 | 13.3500 | -36.3598 |
| All-weights-equal | 0.2305 | **6.8732** | **-42.0678** | 0.4789 | 28.2419 | -28.1536 | 0.1928 | **10.4897** | **-37.0305** |
| Robust optimization | **0.0507** | 56.9168 | -21.9561 | **0.0093** | 94.7811 | -23.5723 | **0.0364** | 72.5601 | -26.8516 |

## A.15 Eigentunes for the results obtained with the cubic polynomial approximation

Tables 33 and 34 shows the eigentune results for the A14 and SHERPA datasets, respectively, when using the cubic polynomial approximation.

## A.16 Generator settings for PYTHIA and SHERPA

Typical run card for A14 studies using PYTHIAv8.186.

```
Tune:pp = 14
Tune:ee = 7

PDF:useLHAPDF = on
PDF:LHAPDFset = NNPDF23_lo_as_0130_qed
```

Table 31: Results for the comparison metrics for the full, observable-filtered and bin-filtered SHERPA dataset using the *cubic polynomial* approximation, calculated on the reduced dataset. The best results are in bold. In each dataset, W-$\chi^2$ refers to the Weighted $\chi^2$ metric, A-o refers to the A-opt metric, and l-D-o refers to the log D-opt metric. Note that we do not have an expert solution for this dataset.

| Data | full dataset | | | observable-filtered dataset | | | bin-filtered dataset | | |
|---|---|---|---|---|---|---|---|---|---|
| Method | W-$\chi^2$ | A-o | l-D-o | W-$\chi^2$ | A-o | l-D-o | W-$\chi^2$ | A-o | l-D-o |
| Bilevel-meanscore | 0.1777 | 9.0959 | -39.9863 | 0.4740 | 14.3374 | -35.2608 | 0.2526 | 17.5098 | -32.5916 |
| Bilevel-medscore | 0.2370 | 13.3943 | -37.1420 | 0.4786 | 13.6299 | -36.6594 | 0.1147 | 15.1990 | **-36.2567** |
| Bilevel-portfolio | 0.3409 | 8.7863 | -39.6956 | 0.2139 | **10.4481** | **-36.8254** | 0.2255 | 15.5833 | -34.9095 |
| All-weights-equal | 0.2305 | **6.8732** | **-42.0678** | 0.3922 | 28.9575 | -27.9246 | 0.1571 | **13.5814** | -34.7914 |
| Robust optimization | **0.0507** | 56.9168 | -21.9561 | **0.0093** | 94.7811 | -23.5723 | **0.0856** | 77.0710 | -26.2532 |

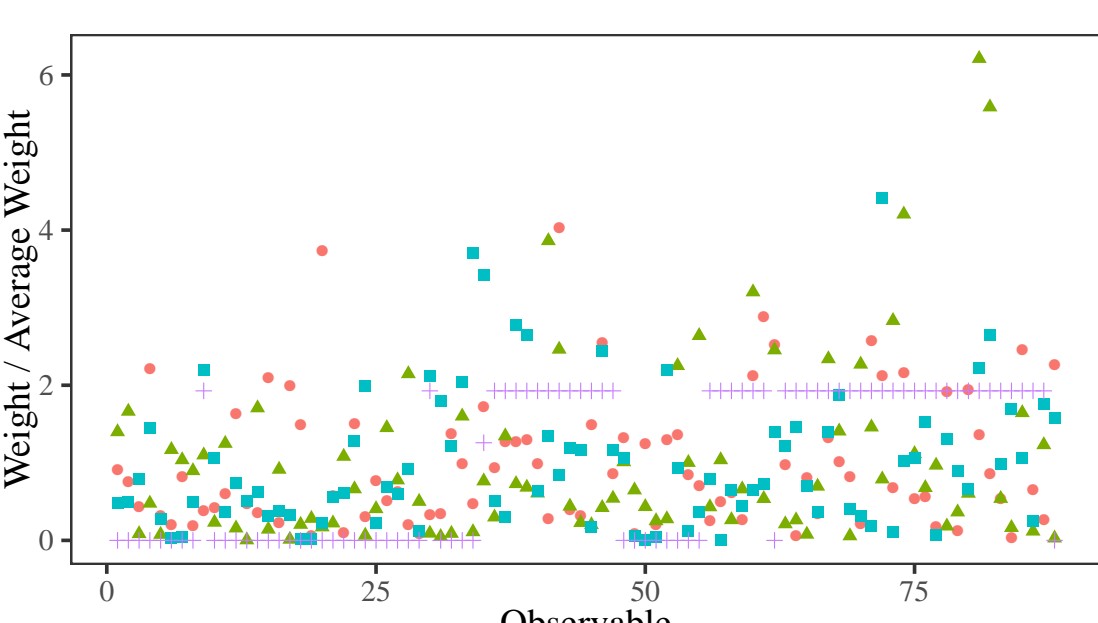

Figure 19: Distribution of weights assigned to observables for the different fitting methods described in the paper. Observables to the left are based on kinematic properties of events, while those to the right are particle multiplicities.

```
PDF:LHAPDFmember = 0
PDF:extrapolateLHAPDF = off

! 3) Beam parameter settings. Values below agree with default ones.
Beams:idA = 2212                    ! first beam, p = 2212, pbar = -2212
Beams:idB = 2212                    ! second beam, p = 2212, pbar = -2212
Beams:eCM = 7000.               ! CM energy of collision

# uncomment for QCD
PhaseSpace:pTHatMin = 10.0
HardQCD:all = on
```

Table 32: Coefficients of the approximation $f_b(\mathbf{p})$ for each bin $b$, which is a linear function of the form $\mathbf{a}^T\mathbf{p} + c$ of the toy model from the closure test described in Section 4.8.

| Observable | Bin | $\mathbf{a}^T$ | $c$ |
|---|---|---|---|
| Observable 1 | Bin 1 | (8.21, 8.22) | 17.65 |
| | Bin 2 | (8.13, 5.23) | 18.96 |
| | Bin 3 | (9.53, 5.54) | 18.37 |
| | Bin 4 | (8.08, 6.41) | 17.61 |
| | Bin 5 | (8.80, 8.75) | 17.07 |
| Observable 2 | Bin 1 | (6.01, 9.71) | 15.63 |
| | Bin 2 | (6.16, 7.12) | 16.71 |
| | Bin 3 | (7.96, 9.10) | 17.18 |
| | Bin 4 | (6.54, 8.98) | 16.74 |
| | Bin 5 | (8.95, 9.42) | 18.93 |
| Observable 3 | Bin 1 | (9.13, 7.66) | 18.23 |
| | Bin 2 | (7.79, 7.86) | 18.07 |
| | Bin 3 | (7.94, 9.14) | 13.81 |
| | Bin 4 | (7.16, 9.07) | 16.15 |
| | Bin 5 | (9.61, 7.97) | 17.49 |
| Observable 4 | Bin 1 | (8.21, 8.22) | 14.40 |
| | Bin 2 | (8.13, 5.23) | 16.98 |
| | Bin 3 | (9.53, 5.54) | 10.89 |
| | Bin 4 | (8.08, 6.41) | 19.48 |
| | Bin 5 | (8.80, 8.75) | 16.50 |

Table 33: Eigentune results for the A14 dataset using the optimal physics parameters $\mathbf{p}^*$ obtained with the different optimization methods when using the *cubic polynomial* approximation.

| Parameters | Expert | | Bilevel-meanscore | | Bilevel-medianscore | | Bilevel-portfolio | | Robust optimization | |
|---|---|---|---|---|---|---|---|---|---|---|
| | min | max | min | max | min | max | min | max | min | max |
| SigmaProcess:alphaSvalue | 0.072 | 0.196 | 0.071 | 0.197 | 0.079 | 0.190 | 0.076 | 0.191 | 0.079 | 0.187 |
| BeamRemnants:primordialKThard | 1.899 | 1.904 | 1.849 | 1.888 | 1.877 | 1.894 | 1.855 | 1.881 | 1.764 | 1.895 |
| SpaceShower:pT0Ref | 1.616 | 1.633 | 1.622 | 1.640 | 1.733 | 1.737 | 1.631 | 1.667 | 1.377 | 1.411 |
| SpaceShower:pTmaxFudge | 0.904 | 0.914 | 0.938 | 0.940 | 0.884 | 0.923 | 0.986 | 0.990 | 0.932 | 0.935 |
| SpaceShower:pTdampFudge | 1.039 | 1.047 | 1.059 | 1.102 | 1.053 | 1.085 | 1.045 | 1.049 | 1.061 | 1.064 |
| SpaceShower:alphaSvalue | 0.116 | 0.128 | 0.128 | 0.130 | 0.118 | 0.141 | 0.129 | 0.131 | 0.128 | 0.129 |
| TimeShower:alphaSvalue | 0.076 | 0.199 | 0.034 | 0.223 | 0.046 | 0.205 | 0.083 | 0.145 | 0.042 | 0.198 |
| MultipartonInteractions:pT0Ref | 1.749 | 2.666 | 1.533 | 2.707 | 1.536 | 2.621 | 1.989 | 2.116 | 1.866 | 1.965 |
| MultipartonInteractions:alphaSvalue | 0.045 | 0.186 | 0.095 | 0.154 | 0.114 | 0.140 | 0.044 | 0.180 | 0.100 | 0.133 |
| BeamRemnants:reconnectRange | 1.719 | 1.719 | 1.523 | 1.541 | 1.390 | 1.420 | 1.589 | 1.595 | 2.565 | 2.568 |

```
PhaseSpace:bias2Selection = on
PhaseSpace:bias2SelectionRef = 10.0
# uncomment for t-tbar
#Top:qqbar2ttbar = on
#Top:gg2ttbar = on
#SpaceShower:pTmaxMatch = 2
#SpaceShower:pTmaxFudge = 1
#SpaceShower:pTdampMatch = 1
# uncomment for Z
#WeakSingleBoson:ffbar2gmZ = On
#23:onMode = off
#23:onIfAny = 11 13 15 5 4 3
#SpaceShower:pTmaxMatch = 2
```

Table 34: Eigentune results for the SHERPA dataset using the optimal physics parameters $\mathbf{p}^*$ obtained with the different optimization methods when using the *cubic polynomial* approximation.

| Parameters | Bilevel-meanscore | | Bilevel-medianscore | | Bilevel-portfolio | | Robust optimization | |
|---|---|---|---|---|---|---|---|---|
| | min | max | min | max | min | max | min | max |
| KT_0 | 0.572 | 1.845 | 0.818 | 0.884 | 0.798 | 1.002 | 0.350 | 1.021 |
| ALPHA_G | 0.113 | 0.769 | 0.472 | 0.690 | 0.612 | 0.639 | 1.288 | 2.044 |
| ALPHA_L | 3.468 | 4.227 | 1.956 | 2.181 | 2.917 | 3.309 | 0 | 1.697 |
| BETA_L | 0 | 0.255 | 0 | 0.487 | 0 | 0.305 | 0 | 0.233 |
| GAMMA_L | 0.064 | 0.915 | 0.226 | 0.405 | 0.746 | 0.755 | 0.328 | 1.625 |
| ALPHA_H | 2.981 | 3.587 | 2.000 | 2.162 | 1.235 | 1.268 | 2.427 | 2.898 |
| BETA_H | 0.662 | 0.771 | 0.582 | 0.677 | 0.637 | 0.675 | 0 | 0.741 |
| GAMMA_H | 0.045 | 0.190 | 0.070 | 0.255 | 0 | 0.134 | 0 | 0.652 |
| STRANGE_FRACTION | 0.068 | 0.749 | 0.446 | 0.655 | 0.501 | 0.558 | 0.413 | 0.546 |
| BARYON_FRACTION | 0 | 0.335 | 0.117 | 0.166 | 0 | 0.186 | 0.030 | 0.516 |
| P_QS_by_P_QQ_norm | 0.669 | 0.828 | 0.576 | 0.715 | 0.458 | 0.549 | 0.537 | 0.619 |
| P_SS_by_P_QQ_norm | 0 | 0.087 | 0 | 0.105 | 0 | 0.076 | 0 | 0.050 |
| P_QQ1_by_P_QQ0 | 1.496 | 1.508 | 1.498 | 1.500 | 1.348 | 1.349 | 1.153 | 1.200 |

```
#SpaceShower:pTmaxFudge = 1
#SpaceShower:pTdampMatch = 1

# Example set of tuning parameters
SigmaProcess:alphaSvalue            0.1343
BeamRemnants:primordialKThard        1.711
SpaceShower:pT0Ref                   1.823
SpaceShower:pTmaxFudge               1.047
SpaceShower:pTdampFudge              1.492
SpaceShower:alphaSvalue             0.1302
TimeShower:alphaSvalue              0.1166
MultipartonInteractions:pT0Ref       2.953
MultipartonInteractions:alphaSvalue  0.127
BeamRemnants:reconnectRange           4.747

ParticleDecays:limitTau0 = on
ParticleDecays:tau0Max = 10
```

We used these settings to reproduce the original results when necessary and to make full predictions for parameters selected using the surrogate function. Some of the original data using in the A14 study was private at that time and was only made public later. In a relatively small number of cases, the public data was in a different form than that used for the original study, so we were unable to reproduce those predictions.

Typical run card for SHERPA studies using v3.0.0.

```
# general settings

SHOWER_GENERATOR: CSS
ANALYSIS: Rivet
FRAGMENTATION: Ahadic
INTEGRATION_ERROR: 0.02

# model parameters

ALPHAS(MZ): 0.1188
ORDER_ALPHAS: 2
```

```
# collider setup

BEAMS: [11, -11]
BEAM_ENERGIES: 45.6

# hadronization parameters
AHADIC:
 KT_0  : 0.9088969039427998
 ALPHA_G  : 1.8736652396525728
 ALPHA_L  : 1.2518697247467987
 BETA_L  : 0.14989272155179253
 GAMMA_L  : 0.6832145156132761
 ALPHA_H  : 2.840868263919124
 BETA_H  : 0.5404054759080933
 GAMMA_H  : 0.14984034099619253
 STRANGE_FRACTION  : 0.5075082631730515
 BARYON_FRACTION  : 0.1357479921139296
 P_QS_by_P_QQ_norm  : 0.612797404412154
 P_SS_by_P_QQ_norm  : 0.029994467832440565
 P_QQ1_by_P_QQ0  : 1.1896505751927051

PARTICLE_DATA:
  4: {Massive: true}
  5: {Massive: true}

PARTICLE_CONTAINER:
  1098: {Name: C, Flavours: [4, -4]}
  1099: {Name: B, Flavours: [5, -5]}

PROCESSES:
- 11 -11 -> 93 93:
    Order: {QCD: 0, EW: 2}
- 11 -11 -> 4 -4:
    Order: {QCD: 0, EW: 2}
- 11 -11 -> 5 -5:
    Order: {QCD: 0, EW: 2}

RIVET:
    ANALYSES:
        - SLD_2002_S4869273
        - DELPHI_1996_S3430090
        - JADE_OPAL_2000_S4300807
        - PDG_HADRON_MULTIPLICITIES
```

We used these settings to reproduce the data for our surrogate function and to make full predictions for parameters selected using the surrogate function.

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
