# Peer review of "BROOD: Bilevel and Robust Optimization and Outlier Detection for Efficient Tuning of High-Energy Physics Event Generators"

_SciPost Physics, doi:SciPost Phys. Core 5, 001 (2022)_

## Round 1 · Referee Report · Anonymous (Referee 1) · 2021-4-23

Report

My apologies for the delay in providing this report ... the paper is very long (!)

Here is my report:

This paper reports a study of multiple methods to automate the selection of bins, observables, and weights for parameter tuning for parton shower simulations in high energy physics. The paper is a serious study, it should certainly be published, and SciPost Physics is a reasonable venue for this. Before I can recommend publication, please see my comments and suggestions below.

Here are two overall impressions:

  • Parameter tuning seems to be a bit of an art and this paper feels like it is adding a lot of mathematical rigor to a problem which lacks mathematical rigor behind the scenes (this comes out in some of my specific comments below). I know this is how tuning is done now, but maybe this could be stated somewhere near the beginning and/or end?

  • There is quite a mix of rigor and non-rigor (for lack of a better word). It would be useful to harmonize this across the draft. There are some concrete suggestions in my comments below.

Detailed comments:

p5:

  • "The uncertainty on the MC simulation comes from the numerical methods used to calculate the predictions, and it typically scales as the inverse of the square root of the number of simulated events in a particular bin." -> Perhaps worth commenting that this is excluding non-statistical theory uncertainty? Some aspects of MCs are under theoretical control and thus have theory uncertainties (that I understand are usually ignored).

  • Since you are being clear with your terminology, it would be good to say that \Delta \mathcal{R}_b is the one-sigma uncertainty from the measurement and will be interpreted as the standard deviation of a Gaussian random variable (often, systematic uncertainties without any statistical origin dominate measurements so I think the word "interpreted" (or similar) is important to state).

  • "A "good" tune is one where the red line falls within the yellow band." -> If the yellow band really is interpreted as the 68% CI, then shouldn't a good tune be one that contains the red line 68% of the time (so ~1/3 of the time, it does not)? People like to look at plots and see all the points within error, but this is a sign of overfitting!

Fig 1: Are these real data? I know you don't want to confuse the reader at this point, but if the data and simulations are real, please say what they are (feel free to forward-reference to a later section).

  • "optimal set of physics parameters" -> Perhaps it would be good to be clear what you mean by "optimal". Since the title of this section is "mathematical formulation", it would be sensible to state mathematically what you mean by "optimal". Along these lines, it would be good to explicitly state somewhere around Eq. 1 that you are ignoring correlations between measurements.

p6:

  • More measurements are starting to provide proper covariance matrices, so you can at least get correlations between bins so going from Eq. 1 to Eq. 2 is a non-trivial approximation. You say this "implicitly assumes that each bin b is completely independent of all other bins." but I would have expected some statement about the impact on the results.

  • Eq. 3c: why is it \hat{p}_w \in (...) and not \hat{p}_w = (...) ?

  • p9: Isn't it redundant to write e(\hat{p}_w | w) since the w is already part of the symbol \hat{p}_w?

  • Eq. 5a, 5b, and 6: Something seems strange here; in optimal portfolio theory, the goal is to identify weights of each component asset. However, your function 6 only depends implicitly on these weights - they do not enter in the "expected return" (Eq. 5a) or the "return variance" (Eq. 5b). Am I missing something?

  • Sec. 2.1.2: can you please provide some intuition here instead of making the reader dig through [13] to find Eq. 27?

  • Sec. 2.2: I don't understand the notion of "uncertainty set" - can you please expand? The interval does not represent the 1\sigma interval or the maximum possible variation (which is infinite). The text after Eq. 10 suggests it has some meaning and is not just a definition for the symbol \mathcal{U}_b in Eq. 10.

p15:

  • "It would be non-physical to adjust the model parameters to explain these extremes." -> I agree, but then why don't you drop these bins from all histograms? If you don't, then you will tune away these effects in some cases because by chance the simulator happens to have region of parameter space that can explain it (physical or otherwise).

  • I believe the precise wording for the null hypothesis is that the mean of R_b is f_b(b) (?) ("appropriately described by" and "no significant difference between" are not precise). Same for the alternative.

  • What is the level that you actually pick?

  • Eq. 15: I think if you do this, then the chi^2 hypothesis is not true. If you are comparing many subsets, than something like an F-test would be more appropriate, or maybe a sentence that says that this is motivated by statistics but does not have a strict type I error at the set point (and then also probably good to remove all of the ultra pedagogical and likely not applicable explanation at the top of p16)

p19: What does "some of the simulation data were available to us" mean?

p21: I found it strange that you did not cite the original data papers that go into the A14 tune (sorry if I missed it!) I also see that Fig. 9, 10, 11 do not provide a reference for the data - please add it!

  • Table 5-7; 13-15: What should I take away from that fact that there is a huge spread in performance and the ranking from the different metrics is quite different? (in some cases, the worse in one metric is the best in another!)

  • Sec. 5: I was surprised that this comes after the results. It is a bit hard to compare your tunes to the "expert ones" if I don't have a sense for the "uncertainty". Can you maybe add the expert values to Table 21?

  • Sec. 7: You have compared many method variations - which one do you suggest as a baseline recommendation?

  • validity: -
  • significance: -
  • originality: -
  • clarity: -
  • formatting: -
  • grammar: -

Author:  Wenjing Wang  on 2021-07-19  [id 1582]

(in reply to Report 1 on 2021-04-23)

Please see the file attachment.

Attachment:

Response_to_Reviews_H0jK6th.pdf

---

## Round 1 · Referee Report · Tilman Plehn (Referee 2) · 2021-4-28

Report

The paper is very interesting, extremely relevant, and has great potential. However, it is missing physics aspects here and there, so I am asking the authors to add more physics discussions for the typical LHC physicists. All my comments are included in the attached pdf file (in red). Please feel free to ignore some of them, if they do not make any sense, but you will get the idea. What I am missing most is the final step, namely an application of the eigentunes to something new...

Attachment

  • validity: -
  • significance: -
  • originality: -
  • clarity: -
  • formatting: -
  • grammar: -

Author:  Wenjing Wang  on 2021-07-19  [id 1581]

(in reply to Report 2 by Tilman Plehn on 2021-04-28)

Please see the file attachment.

Attachment:

Response_to_Reviews_MocOyef.pdf

Tilman Plehn  on 2021-07-24  [id 1610]

(in reply to Wenjing Wang on 2021-07-19 [id 1581])

Thank you for going through all my comments. Looks great, so let's publish!

---

## Round 1 · Referee Report · Anonymous (Referee 3) · 2021-5-14

Strengths

New procedure for computing demanding task, that is reduced to a few hours run with the proposed methodology. In addition, overall quality improvement demonstrated with specific examples. A clear improvement over the state of the art alternatives.

Weaknesses

The procedure follows closely what is done normally ( a chisquare fit over histograms). So, as much as the accepted procedure, it makes little sense to me. Correlations are systematically neglected and the N-dim distribution of the observed quantities taking as input is simplified to a set of 1D distributions. In my mind, this can introduce biases in the problem.

Report

The paper discusses the task of MC tuning and proposes a new procedure to improve over current state of the art, by speeding up the computation and reaching a better agreement on a real-life example. This is done utilising a few new elements, including a better minimisation strategy (a few are proposed) and an outlier removal procedure. I have a few questions regarding the big picture proposed: 1) To which extent the outlier removal is sound? Even in absence of systematic issues, outliers will occur. Removing them might help to established the expectation value of the parameter one is fitting, but any sense of statistical interpretation of the uncertainty range is lost (i.e., coverage is broken). The paper offers no discussion of this point and how crucial this is. 2) The paper is TOO LONG. The same content can be delivered in 1/2 the length. Authors explain established concepts (I doubt that one needs to explain what an histogram is in a physics paper). Often, the authors repeat sentences twice, assuming that this comes as an explanation (e.g., page 34). I think that an effort should be taken to reduce the paper length and make the paper more readable. The text can be reduced with no impact on the amount of transmitted information. 3) I would expect that a paper of this kind would address what I consider the elephant in the room. MC tuning uses a set of correlated 1D quantities as uncorrelated quantities, instead of taking as input a N-dim distribution. To me, this is potentially dangerous. At least, I would expect a paper of this kind to discuss this as a potential problem and discuss the balance between what is right and what is doable with the existing information. It is true that authors discuss input weights to alleviate the issue. But this sounds to me as the survival of the "by hand" intervention that this paper aims to remove from the tuning procedure, since there is an arbitrariness in the weight setting strategy (it is not obvious to me that Eq.3 is the only choice). 4) The analysis of the accuracy benefit is left on a qualitative level. I would be interested to understand the coverage property of the fitting method, the validity of the quoted uncertainties, and the capability to converge to the correct minimum. 5) Related to the previous points, I have the impression that the paper would benefit from including a closure test on some toy dataset in which one knows the "right" answer and could demonstrate that the procedure would provide an unbiased result.

In view of the previous comments, I think that a revision is needed. Considering the last few comments. I also ask the authors to consider the list of proposed changes, given below.

Requested changes

Sec 1.1: I think one can live w/o the explanation of what a histogram is
Pag.6: Isn’t this replacement of the histogram by a surrogate model subject to a systematic uncertainty related to the choice of the functional form? I would expect an analysis of performance benefits vs accuracy costs.
Eq. 3.b. s.t. for "subject to" is a very uncommon notation for the physics literature I am used to. You might want to consider to re-format the three equations
Pag 13: Are the optimized parameters varied in a range? If so, a discrepancy could also be induced by too strict boundaries. This possibility is not considered but it might be relevant.
Pag 15: the offered procedure is effectively a single-hypothesis test. Only Fisher called this kind of test hypothesis test. He disagreed with the Neyman Pearson hypothesis test, which requires two hypotheses to be specified. My understanding is that Fisher's problem is ill posed, as decades of frequentist literature exposed problems related to this .In particular, no any claim of optimization in this context is typically an overstatement. This point should be discussed.
Line 451: "The main reason for this discrepancy is the fact that we use a better optimization routine" Certainly a different one. But better in which context? I think the paper should offer more evidence of this claim. My understanding is that most of the improvement comes from the surrogate model, which has pros and cons (see previous comment) that should be assessed.
Table 3: I assume the quoted uncertainties correspond to a “1-sigma” range because it’s related to the eigenvectors of the linearised problem. If so, it should be clarified. I have doubts about the fact that your outlier removal procedure is not altering the statistical interpretability of these uncertainties.
Line 544: Isn’t this obvious (and not necessarily right)?Outliers were removed, so I would expect exactly this. Am I too naive? I would like to see this procedure repeated on a toy data in which you know that the method can describe the model, i.e., in which you should not remove anything in principle. What happens? (see general comment in the report). I think this deserves some comment.
Line 559: I am concerned with the fact that 50% of the points come from an observable. Are your weigths acting as a regularization of this? My understanding was that their main purpose was to limit the redundancy among correlated quantities. Or does the number of bins/observable enter?
Fig. 6: The label on the bottom-right is cut.
Sec 4.8: why not running everything on one machine and make a one-to-one comparison?

  • validity: good
  • significance: good
  • originality: high
  • clarity: good
  • formatting: good
  • grammar: excellent

Author:  Wenjing Wang  on 2021-07-19  [id 1580]

(in reply to Report 3 on 2021-05-14)
Category:
answer to question
correction

Please see the file attachment.

Attachment:

Response_to_Reviews.pdf

---

## Round 2 · Referee Report · Anonymous (Referee 1) · 2021-7-24

Report

Thank you to the authors for their detailed responses to my comments and for integrating my suggestions into the manuscript. I am satisfied with the responses and I think the paper is now ready for publication.
  • validity: -
  • significance: -
  • originality: -
  • clarity: -
  • formatting: -
  • grammar: -

Author:  Wenjing Wang  on 2021-09-01  [id 1723]

(in reply to Report 1 on 2021-07-24)

Thank you for your positive review.

Attachment:

Response_to_Reviews_2_qKCzTu4.pdf

---

## Round 2 · Referee Report · Anonymous (Referee 3) · 2021-8-20

Strengths

Same as in previous review

Weaknesses

In this round of review, the authors made il clear that the improvement they obtain is practical (automatised workflow) but not conceptual. In this respect, my previous assessment on the interest of this paper should be reduced (still, the paper is interesting enough to meet publication requirements)

Report

I went through the new version of the manuscript and the provided answers to my comments. I am in general very satisfied with the effort made by the authors to improve the paper.

I still have a few follow up comments:

To comment iii) The authors insist that removing points is conservative. I don’t see it, sorry. Would be nice if they could show what they say. From my experience, removing outliers in a fit (particularly a chisq fit) has an effect on the fit (bias and uncertainty underestimate). I have the impression that what the authors say is true under the assumption that the model you use to fit the data is correct. But they comment at length that this is not the case (maybe because of the underlying assumptions, e.g., fixed-order perturbative calculations, soft-physics modeling, etc). So I am not very convinced of all this and I would like to see a more clear demonstration.

To comment e) I disagree. NP requires an alternative hypothesis. Not clear what that is, in your method. A quick google search for a paper that spells this out: https://www.ncbi.nlm.nih.gov/pmc/articles/PMC4347431/
Not a super relevant point. I would just avoid saying hypothesis test and would say null-hypothesis test or some such

To comment l) I acknowledge the fact that speed is not the main aspect. But I think the question stands. I find this choice quite odd, also because even if the time is not the main concern of the paper, it is the main concern of this section (see title of 4.9). At least some explanation of the reasoning behind this choice should be given
  • validity: -
  • significance: -
  • originality: -
  • clarity: -
  • formatting: -
  • grammar: -

Author:  Wenjing Wang  on 2021-09-01  [id 1722]

(in reply to Report 2 on 2021-08-20)

please find attached our response to your comments.

Attachment:

Response_to_Reviews_2.pdf

---

## Round 3 · Referee Report · Anonymous · 2021-9-27

Report

The new version clarified all my remaining questions. I agree with the paper being accepted for publication

  • validity: high
  • significance: good
  • originality: good
  • clarity: good
  • formatting: good
  • grammar: good

Author:  Wenjing Wang  on 2021-09-30  [id 1791]

(in reply to Report 1 on 2021-09-27)

Thank you for your positive comments! We appreciate your support of our paper.

---

## Editorial Decision

published